# Modeling seasonal and vertical habitats of planktonic foraminifera on a global scale

Kerstin Kretschmer[1], Lukas Jonkers[1], Michal Kucera[1], and Michael Schulz[1]

[1]MARUM - Center for Marine Environmental Sciences and Faculty of Geosciences, University of Bremen, Germany.

*Correspondence to:* Kerstin Kretschmer (kkretschmer@marum.de)

**Abstract.** Species of planktonic foraminifera exhibit specific seasonal production patterns and different preferred vertical habitats. The seasonal and vertical habitats are not constant throughout the range of the species and changes therein must be considered when interpreting paleoceanographic reconstructions based on fossil foraminifera. However, detecting the effect of changing vertical and seasonal habitat on foraminifera proxies requires independent evidence for either habitat or climate change. In practice, this renders accounting for habitat tracking from fossil evidence almost impossible. An alternative method that could reduce the bias in paleoceanographic reconstructions is to predict species-specific habitat shifts under climate change using an ecosystem modeling approach. To this end, we present a new version of a planktonic foraminifera model, PLAFOM2.0, embedded into the ocean component of the Community Earth System Model, version 1.2.2. This model predicts monthly global concentrations of the planktonic foraminiferal species: *Neogloboquadrina pachyderma*, *N. incompta*, *Globigerina bulloides*, *Globigerinoides ruber* (white), and *Trilobatus sacculifer* throughout the world ocean, resolved in 24 vertical layers to $250\,\mathrm{m}$ depth. The resolution along the vertical dimension has been implemented by applying the previously used spatial parameterization of carbon biomass as a function of temperature, light, nutrition, and competition on depth-resolved parameter fields. This approach alone results in the emergence of species-specific vertical habitats, which are spatially and temporally variable. Although an explicit parameterization of the vertical dimension has not been carried out, the seasonal and vertical distribution patterns predicted by the model are in good agreement with sediment trap data and plankton tow observations. In the simulation, the colder-water species *N. pachyderma*, *N. incompta*, and *G. bulloides* show a pronounced seasonal cycle in their depth habitat in the polar and subpolar regions, which appears to be controlled by food availability. During the warm season, these species preferably occur in the subsurface (below $50\,\mathrm{m}$ water depth), while towards the cold season they ascend through the water column and are found closer to the sea surface. The warm-water species *G. ruber* (white) and *T. sacculifer* exhibit a less variable shallow depth habitat with highest carbon biomass concentrations within the top $40\,\mathrm{m}$ of the water column. Nevertheless, even these species show vertical habitat variability and their seasonal occurrence outside the tropics is limited to the warm surface layer that develops at the end of the warm season. The emergence in PLAFOM2.0 of species-specific vertical habitats, which are consistent with observations, indicates that the population dynamics of planktonic foraminifera species may be driven by the same factors in time, space, and with depth, in which case the model can provide a reliable and robust tool to aid the interpretation of proxy records.

## 1 Introduction

Planktonic foraminifera are found throughout the open ocean, where they inhabit roughly the top $500\,\text{m}$ of the water column (Fairbanks et al., 1980, 1982; Kohfeld et al., 1996; Kemle-von Mücke and Oberhänsli, 1999; Mortyn and Charles, 2003;
Field, 2004; Kuroyanagi and Kawahata, 2004; Bergami et al., 2009; Wilke et al., 2009; Pados and Spielhagen, 2014; Iwasaki et al., 2017; Rebotim et al., 2017). Their calcareous shells, preserved in ocean sediments, are widely used to reconstruct past climate conditions. To do so, information about their habitat including their horizontal and vertical distribution are needed. It is known from observational data that the prevailing environmental conditions, such as temperature, stratification, light intensity, and food availability, affect the growth and distribution of the individual planktonic foraminifera (Fairbanks et al., 1980, 1982;
Bijma et al., 1990b; Watkins et al., 1996; Schiebel et al., 2001; Field, 2004; Kuroyanagi and Kawahata, 2004; Žarić et al., 2005; Salmon et al., 2015; Rebotim et al., 2017). Based on stratified plankton tow and sediment trap data the seasonal succession of planktonic foraminifera species has been assessed on a local/regional scale (e.g., Fairbanks and Wiebe, 1980; Kohfeld et al., 1996; Wilke et al., 2009; Jonkers et al., 2013; Jonkers and Kučera, 2015), whereas for a broader regional/global perspective, modeling approaches have been used to study the seasonal variations in the surface (mixed) layer of the ocean (Žarić et al.,
2006; Fraile et al., 2008, 2009a, b; Lombard et al., 2011; Kretschmer et al., 2016). Comparatively less is known about the depth habitat of planktonic foraminifera species and how it varies seasonally. Although previous studies identified different environmental and ontogenetic factors (i.a., temperature, chlorophyll *a* concentration, the lunar cycle and/or the structure of the water column), which influence the species-specific depth habitats including their mean living depth and vertical migration (e.g., Fairbanks and Wiebe, 1980; Fairbanks et al., 1982; Schiebel et al., 2001; Simstich et al., 2003; Field, 2004; Salmon et al.,
2015; Rebotim et al., 2017), the only attempt to model the vertical habitat is by Lombard et al. (2011).

It is well known that species-specific habitats vary seasonally and spatially depending on the prevailing climatic conditions (Mix, 1987; Mulitza et al., 1998; Ganssen and Kroon, 2000; Skinner and Elderfield, 2005; Jonkers and Kučera, 2015). Yet, despite this evidence for a variable habitat, it is often assumed in paleoceanographic studies that the habitat of planktonic foraminifera is constant, i.e., that it does not change in time and space, potentially leading to erroneous estimates of past
climate conditions. Jonkers and Kučera (2017) recently highlighted how foraminifera proxies are affected by habitat tracking and showed that by not accounting for this behavior, spatial and temporal trends in proxy records may be underestimated. Given the habitat variability in planktonic foraminifera, it is more than likely that a climate-dependent offset from mean annual sea surface conditions results not only from a seasonal, but also from depth habitat variability due to changes in ambient conditions. Such vertical habitat variability was shown by Rebotim et al. (2017), who investigated parameters controlling the
depth habitat of planktonic foraminifera in the subtropical eastern North Atlantic. In line with studies from other regions of the world ocean (e.g., Fairbanks et al., 1982; Bijma et al., 1990a; Ortiz et al., 1995; Schiebel et al., 2001; Field, 2004; Salmon et al., 2015), Rebotim et al. (2017) identified distinct species-specific depth habitats, but they also showed that the habitats vary on lunar and seasonal time scales and in response to temperature, chlorophyll *a*, and other environmental factors. Evidence for

variable depth habitats at least on a regional scale has emerged from studies in other regions (Watkins et al., 1998; Peeters and Brummer, 2002; Kuroyanagi and Kawahata, 2004).

These observations underline the necessity to consider species-specific habitats and their variability on a global scale to increase the reliability of paleoceanographic reconstructions. However, a global assessment of species-specific depth habitat
variability in time and space and the potential underlying control mechanisms is lacking. Since the observational data coverage of the global ocean is too sparse to provide in this regard a broad general estimate, we apply an ecosystem modeling approach to predict the vertical and seasonal distribution of planktonic foraminifera on a global scale.

## 2   Methods

### 2.1   Approach

To predict the seasonally varying global species-specific depth habitat of planktonic foraminifera, we modified the previously developed PLAFOM model (Fraile et al., 2008; Kretschmer et al., 2016), which is implemented as an off-line module into the ocean component of the Community Earth System Model, version 1.2.2 (CESM1.2; Hurrell et al., 2013), with active ocean biogeochemistry (which is denoted as CESM1.2(BGC) configuration). This model system simulates the monthly concentrations of five modern planktonic foraminiferal species, which are widely used in paleoceanographic reconstructions. The original
approach of Fraile et al. (2008) and Kretschmer et al. (2016) aimed to predict the distribution of planktonic foraminifera in the surface mixed layer on geological time scales. This model version has been successfully used to assess the effect of changing environmental conditions on species distributional patterns in time and space (Fraile et al., 2009a, b; Kretschmer et al., 2016) and to aid in interpreting paleoceanographic records regarding seasonal production shifts in the geological past (Kretschmer et al., 2016), but could not provide any information about depth. To implement the vertical dimension, we used an approach, in
which we first updated PLAFOM (hereafter referred to as PLAFOM2.0) by including light dependency for symbiont-bearing planktonic foraminifera and then applied the previously used spatial parameterization of carbon biomass as a function of temperature, nutrition, and competition, together with light, on depth-resolved parameter fields. By combining PLAFOM2.0 with the CESM1.2(BGC) configuration (hereafter referred to as CESM1.2(BGC+PLA) configuration), the vertical dimension can be resolved throughout the ocean, with 24 layers in the top $250\,\mathrm{m}$. Thus, PLAFOM2.0, as belonging to a suite of proxy system
models (e.g., Pollard and Schulz, 1994; Schmidt, 1999; Fraile et al., 2008; Evans et al., 2013; Dee et al., 2015; Völpel et al., 2017), will aid the interpretation of paleoclimate reconstructions. In addition, PLAFOM2.0 has the potential to be used in a paleoclimate data assimilation framework (see, e.g., Goosse et al., 2010; Steiger et al., 2014; Dee et al., 2016; Hakim et al., 2016).

### 2.2   CESM1.2(BGC) configuration

We used the CESM1.2(BGC) configuration (Moore et al., 2013; Lindsay et al., 2014) as code base. This configuration includes the Biogeochemical Elemental Cycling (BEC) model (Moore et al., 2004, 2006; Krishnamurthy et al., 2007; Moore

and Braucher, 2008), which is based on the upper ocean ecosystem model of Moore et al. (2002a, b) coupled to a biogeo-chemistry model based on the Ocean Carbon Model Intercomparison Project (OCMIP; Doney et al., 2006). The BEC model includes various potentially growth-limiting nutrients (nitrate, ammonium, phosphate, dissolved iron, and silicate), three explicit phytoplankton functional types (diatoms, diazotrophs, pico/nano phytoplankton), a partial calcifier class (representing

coccolithophores), a single adaptive zooplankton class, dissolved organic matter, sinking particulate detritus, and full carbonate system thermodynamics (Moore et al., 2004, 2013). Phytoplankton growth rates are controlled by temperature, light, and available nutrients (Moore et al., 2002b, 2004). The single zooplankton pool grazes on all phytoplankton types, whereby the routing of grazed material varies depending on the type of prey (Moore et al., 2004, 2013). For further details, we refer to Moore et al. (2002b, 2004, 2013).

The BEC model has been embedded into the ocean component of CESM, version 1.2.2. CESM1.2 is a fully coupled climate model consisting of several components including the atmosphere, ocean, land, and sea ice (Hurrell et al., 2013), whereby the geophysical fluxes among the components are exchanged through a central coupler (Craig et al., 2012). Here we performed an ocean-ice-only simulation with active ocean biogeochemistry, whereby the ocean model is coupled to both the sea ice model and data models for the atmosphere, land, and river routing, which provide the required input data for the simulation.

The CESM1.2 ocean component is the Parallel Ocean Program, version 2 (POP2; Smith et al., 2010; Danabasoglu et al., 2012), with a zonal resolution of $1°$ and an increased meridional resolution of $0.27°$ near the equator. POP2 employs a non-uniform dipolar grid with the North Pole being displaced into Greenland. With a total number of 60 vertical levels, the grid spacing is fine near the surface (10 levels in the top $100\,\mathrm{m}$) and increases with depth to $250\,\mathrm{m}$ near the bottom. The sea ice component of CESM1.2 is the Community Ice Code, version 4 (CICE4; Hunke and Lipscomb, 2008; Holland et al., 2012),

which uses the same horizontal grid as the ocean model.

## 2.3 PLAFOM2.0

This new model version, PLAFOM2.0, considers the polar species *Neogloboquadrina pachyderma*, which is supplemented by the subpolar species *N. incompta* (sensu Darling et al., 2006) and *Globigerina bulloides* as well as by the warm-water algal symbiont-bearing species *Globigerinoides ruber* (white) and *Trilobatus sacculifer* (sensu Spezzaferri et al., 2015). Those

species have been chosen as they can be considered to represent a large portion of the planktonic foraminiferal biomass in the surface ocean (for further details see Kretschmer et al., 2016). The different planktonic foraminifera species were added to the ocean component of CESM1.2 as optional passive tracers with the requirement that the BEC model is active.

PLAFOM2.0 is driven by temperature, the available food sources (including zooplankton, diatoms, small phytoplankton, and organic detritus), and also light availability, whereby the latter only matters with regard to the growth of the two algal

symbiont-bearing species (Erez, 1983; Jørgensen et al., 1985; Gastrich, 1987; Gastrich and Bartha, 1988) and *G. bulloides*, which according to the latest findings hosts the picocyanobacterium *Synechococcus* as a photosynthesizing endobiont (Bird et al., 2017). *Synechococcus* is known to be important for cyanobacterial photosynthesis in marine and freshwater ecosystems (Ting et al., 2002; Jodłowska and Śliwińska, 2014).

The food preferences and temperature tolerance limits for each species have been derived from sediment trap data and culturing experiments (see Fraile et al., 2008, for details). Changes in the foraminifera carbon concentration for each species are determined as follows:

$$\frac{dF}{dt} = (GGE \cdot TG) - ML \tag{1}$$

where $F$ is the foraminifera carbon concentration (in $\mathrm{mmol\,C\,m^{-3}}$), $GGE$ (gross growth efficiency) is the portion of grazed matter that is incorporated into foraminiferal carbon biomass, $TG$ represents total grazing (i.e., the growth rate in $\mathrm{mmol\,C\,m^{-3}s^{-1}}$), and $ML$ denotes mass loss (i.e., the mortality rate in $\mathrm{mmol\,C\,m^{-3}s^{-1}}$). To properly simulate the vertical distribution of each considered planktonic foraminifera, we included light dependency and modified parts of the parameterizations of the foraminiferal species concentration. Therefore, we extended the growth rate equation by not only considering food availabil-

ity and temperature sensitivity, but also light intensity to define growth. Additionally, we adjusted parts of the mortality rate equation to improve the model accuracy. In the following, the performed modifications are described in detail in regard to the growth and mortality rates. The modifications compared to the earlier model version are summarized in Table 1.

### 2.3.1 Growth rate

The growth rate depends on the available food and temperature sensitivity of each foraminiferal species as well as on light for

the species with algal symbionts and/or cyanobacterial endobionts. To account for the light dependence with depth influencing the growth of *G. bulloides* and of the spinose species *G. ruber* (white) and *T. sacculifer*, we included a photosynthetic growth rate. As a first-order estimate, we applied a similar approach as Doney et al. (1996) and Geider et al. (1998), who determined phytoplankton growth rates based on available light and nutrient conditions, which have been accordingly used in the BEC model (Moore et al., 2002b, 2004). We are aware that a phytoplankton response to light is not directly transferable to planktonic

foraminifera, but we argue that as a first approximation this is a valid approach.

Photosynthesis depends on light availability and temperature. This co-dependency can be expressed as follows:

$$P_{F,photo} = P_{F,max} \cdot \left[ 1 - \exp\left( \frac{-\alpha_{PI} \cdot I_{PAR}}{P_{F,max}} \right) \right]$$

where $P_{F,photo}$ is the foraminiferal specific rate of photosynthesis (in $\mathrm{s^{-1}}$) and $P_{F,max}$ is the maximum value of $P_{F,photo}$ at temperature $T$ (in $\mathrm{s^{-1}}$), calculated as:

$P_{F,max} = P_{F,0} \cdot T_{func}$

$\alpha_{PI}$ is the initial slope of the photosynthesis-light curve (in $\mathrm{m^2\,W^{-1}s^{-1}}$) (Table 1), $I_{PAR}$ is the average irradiance over the mixed layer depth provided by the ecosystem model (in $\mathrm{W\,m^{-2}}$), $P_{F,0}$ represents the maximum foraminiferal growth rate at a specific temperature $T_0$ (in $\mathrm{s^{-1}}$) (Table 1), and $T_{func}$ is the temperature response function (dimensionless). The temperature function is defined as:

$T_{func} = q_{10}^{\frac{T-T_0}{10}} \tag{2}$

with a $q_{10}$ value of 1.5 (Sherman et al., 2016) and $T$ being the ambient ocean temperature (in K) and $T_0$ the reference temperature of 303.15 K.

The photosynthetic growth rate, $P_F$ (in $\mathrm{mmol\,C\,m^{-3}s^{-1}}$), can finally be determined as follows:

$$P_F = P_{F,photo} \cdot F \cdot p_\%$$

where $p_\%$ represents the fraction of photosynthesis contributing to growth (see Table 1).

### 2.3.2    Mortality rate

The mortality rate is determined by respiration loss, predation by higher trophic levels, and competition among species. To improve the seasonal patterns in the foraminiferal carbon biomass for low temperatures, we followed Moore et al. (2004) and adjusted the temperature dependence of the predation term ($ML_{pred}$ in $\mathrm{mmol\,C\,m^{-3}s^{-1}}$):

$$ML_{pred} = f_{mort2} \cdot T_{func} \cdot F_p^2$$

where $f_{mort2}$ represents the quadratic mortality rate (in $\mathrm{s^{-1}(mmol\,C\,m^{-3})^{-1}}$), $T_{func}$ is the temperature response function (dimensionless) used for scaling, and $F_p$ (in $\mathrm{mmol\,C\,m^{-3}}$) is used to limit the planktonic foraminifera mortality at very low carbon biomass levels. Compared to Fraile et al. (2008), here predation is scaled by Eq. (2), a temperature function using a $q_{10}$ value of 1.5 (Sherman et al., 2016).

Additionally, we included a stronger competitive behavior of *G. bulloides* by adjusting the free parameters in the competition term. In PLAFOM2.0, competition ($ML_{comp}$ in $\mathrm{mmol\,C\,m^{-3}s^{-1}}$) is defined as follows:

$$ML_{comp} = \sum_i \left[ F_p \cdot \frac{cl_{ij} \cdot F_i \cdot d}{F_i \cdot d + 0.1} \right]$$

with $F_i$ being the concentration of the foraminiferal species exerting competition, $cl_{ij}$ the maximum competition pressure of species $i$ upon species $j$, and $d$ the constant controlling the steepness of the Michaelis-Menten relationship for competition. In

comparison with Kretschmer et al. (2016), we only modified the parameter $cl_{ij}$ for *N. incompta*, *G. bulloides*, and *G. ruber* (white) (Table 1).

We added the present implementation of PLAFOM2.0 to the code trunk of POP2 as a separate module. Additionally, the food sources for the planktonic foraminifera species are computed in the ecosystem model and instantly passed to PLAFOM2.0 to calculate the foraminifera carbon concentration. A parameter sensitivity assessment for PLAFOM was carried out by Fraile

et al. (2008) and since PLAFOM2.0 is based on the same underlying formulation, we consider an extensive new sensitivity assessment not essential at this stage. For a more detailed description of the planktonic foraminifera model and its behavior on a regional/global scale in the surface mixed layer, we refer to Fraile et al. (2008) and Kretschmer et al. (2016).

### 2.4    Model simulation

To test the model, we performed a preindustrial-control experiment. Therefore, we derived the initial ocean and sea ice states

from an ocean-ice-only simulation, which did not include the BEC ocean biogeochemistry. This model integration was spun-up

from rest for 300 years to approach a quasi-steady state by using a climatological forcing (based on atmospheric observations and reanalysis data) as repeated normal year forcing. Heat, freshwater, and momentum fluxes at the sea surface are based on the atmospheric data sets developed by Large and Yeager (2004, 2009) and implemented following the CORE-II-protocol (Coordinated Ocean-ice Reference Experiment) suggested by Griffies et al. (2009).

The oceanic and sea ice tracer fields (such as potential temperature, salinity, and ice area) resulting from the end of this 300-year-long spin-up run were used to initialize the CESM1.2(BGC+PLA) preindustrial-control simulation. The biogeochemical tracer fields (such as nutrients) were, i.a., initialized from climatologies. For instance, initial nutrient (phosphate, nitrate, silicate) distributions were taken from the World Ocean Atlas 2009 (WOA09; Garcia et al., 2010), initial values for dissolved inorganic carbon and alkalinity are from the Global Ocean Data Analysis Project (GLODAP; Key et al., 2004), whereas zoo-

plankton, phytoplankton pools, and dissolved organic matter have been initialized uniformly at low values (Moore et al., 2004). Additionally, each planktonic foraminiferal species was also initialized uniformly at low values assuming the same (vertical) distribution as the zooplankton component of the BEC model. Furthermore, the atmospheric deposition of iron and dust is based on the climatology of Luo et al. (2003).

The CESM1.2(BGC+PLA) preindustrial-control simulation was integrated for 300 years to reach stable conditions in the

ocean biogeochemistry in the upper $500\,\mathrm{m}$ of the water column (see Figure S1 in the Supplement). Since this simulation has been forced and/or initialized based on climatologies, inter-annual variability and forcing trends can be excluded and, therefore, we focus our analysis on the model output of only one year, here of year 300.

## 2.5  Comparison to observations

To validate the model performance, we compare the simulated spatial and temporal distributions of the considered planktonic

foraminiferal species with data from core-tops, sediment traps, and plankton tows (Figure 1). Based on data availability, we focus our analysis on distinct regions distributed over the world ocean covering all climate zones from the poles to the tropics.

### 2.5.1  Core-top data

To examine the spatial pattern of the five considered planktonic foraminiferal species, we compared the model predictions with fossil data by using in total 2896 core-top samples distributed over all oceans (Figure 1a). We combined the Brown

University Foraminiferal Database (Prell et al., 1999) with the data assembled by the MARGO project (Kucera et al., 2005), and the data sets provided by Pflaumann et al. (1996, 2003). For the comparison, we recalculated the relative abundances of the faunal assemblages by only considering those five species used in PLAFOM2.0. Similarity between the simulated and observed abundances was quantified using the Bray-Curtis index of similarity ($b_{jk}$ in %) between the relative abundances of the core-top data and the modeled data at the respective sample locations:

$$b_{jk} = \left(1 - \frac{1}{2} \cdot \sum_{i=1}^{5} |x_{ji} - x_{ki}|\right) \cdot 100\%$$

Here $x_{ji}$ and $x_{ki}$ are the modeled and observed relative abundances (with values between 0 and 1) of each species $i$ at the given core-top locations, respectively. Note that for the calculation of the modeled relative abundances, we accounted for the different

sizes of each individual species by multiplying the modeled annual mean concentration of each species with an estimate of their relative sizes (Table 2).

### 2.5.2 Sediment trap data

To compare modeled and observed seasonal production patterns, several sediment traps (Table S1 in the Supplement, Figure 1b) have been examined. Those can provide foraminiferal shell fluxes continuously collected over several months or even years. However, some sediment traps comprise only of a few months (i.e., less than a year) and might have just recorded local short-term processes of a particular season/year and can, thus, not provide a long-term/climatological mean.

Here we use the same approach as in Jonkers and Kučera (2015) and present the observed fluxes for multiple years from every location on a $\log_{10}$ scale versus day of year, whereby the zero fluxes have been replaced by half of the observed minimum flux to be able to visualize the results. In this way, we can directly compare the peak timings of the measured fluxes at each location with the model, whereby we assume that the flux through the water column (in $\#\,\mathrm{m}^{-2}\,\mathrm{day}^{-1}$) is proportional to the volume integrated model concentrations (in $\mathrm{mmol}\,\mathrm{C}\,\mathrm{m}^{-3}$).

### 2.5.3 Plankton tow data

To analyze the vertical distribution, plankton net hauls from different sites distributed across the world ocean (Table S2 in the Supplement, Figure 1b) have been used for a comparison with the simulated vertical distributions. Plankton tow samples have been collected by means of a multiple opening-closing net with a vertical resolution differing between 5 depth levels (one haul) and up to 13 depth levels (two or more consecutive hauls) resolving the upper 100s of meters of the water column. Since the plankton tow data has been collected during a particular time (i.e., a specific day/month) (Table S2), the same month has been considered for the simulated vertical planktonic foraminifera profile for the model-data comparison.

Here we followed the same approach as Rebotim et al. (2017) and calculated an average living depth (ALD) and the vertical dispersion (VD) around the ALD to provide a direct comparison with the modeled depth profile. The ALD (in m) is defined as follows:

$$ALD = \frac{\sum_i C_i \cdot D_i}{\sum_i C_i}$$

with $C_i$ being the foraminiferal species concentration (in $\#\,\mathrm{m}^{-3}$) in the depth interval $D_i$ and VD (in m) is calculated as:

$$VD = \frac{\sum_i(|ALD - D_i| \cdot C_i)}{\sum_i C_i}$$

For further information, we refer to Rebotim et al. (2017).

## 3 Results

### 3.1 Modeled horizontal distribution patterns

The modeled global spatial distribution patterns based on the depth integrated annual mean relative abundances of the five considered foraminiferal species (Figure 2) correspond to the five major provinces of the modern ocean (i.e., polar, subpolar, transitional, subtropical, and tropical) known to be inhabited by those species (Bradshaw, 1959; Bé and Tolderlund, 1971; Hemleben et al., 1989; Kucera, 2007). Note that since the core-top data used for comparison provide information neither on the depth habitat of the planktonic foraminiferal species nor on their life cycle, the modeled annual mean relative abundances have been obtained by integrating the individual foraminiferal concentrations over the whole water column and by subsequently calculating the percentage of each species relative to the modeled total foraminiferal carbon biomass, whereby we also accounted for the different sizes of each species (Table 2).

For a direct comparison of the observed (i.e., the core-top data) and modeled foraminiferal community composition the Bray-Curtis index of similarity was used. The comparison reveals generally a good fit between the simulated and sedimentary assemblage composition with median Bray-Curtis similarity of $\sim 68\%$. The fit is particularly good in the high latitudes and in the tropics (Bray-Curtis similarity $> 80\%$) and only a few regions (off South America and southern Africa, in the equatorial and North Pacific, and in the eastern North Atlantic) reveal a poorer agreement with similarities of $< 50\%$ (Figure 2a).

In the simulation, the cold-water species *N. pachyderma* is confined to the high latitudes dominating the polar waters of both hemispheres. *Neogloboquadrina pachyderma* shows the highest modeled annual mean relative abundances ($> 90\%$) north of the Arctic Circle and south of the Antarctic Convergence, whereas toward the subtropics the species' occurrence in the model reduces gradually (Figure 2b). *Neogloboquadrina incompta* occurs mainly in the subpolar to transitional water masses of the world ocean in the simulation. This species shows highest modeled annual mean relative abundances in the latitudinal belt at around $45°N$ and/or $45°S$ (Figure 2c). *Globigerina bulloides* also occurs in the subpolar to transitional waters of the world oceans with the highest modeled annual mean relative abundances ($> 60\%$) occurring in the Southern Ocean and in the subpolar gyres (Figure 2d). In the upwelling region of the equatorial Pacific and in the coastal upwelling systems associated with the cold eastern boundary currents of the Atlantic and Pacific Oceans, *G. bulloides* is found with modeled annual mean relative abundances of $< 40\%$. In the simulation, the warm-water species *G. ruber* (white) is mostly confined to the subtropical and tropical regions of both hemispheres, whereby the highest modeled annual mean relative abundances of up to 60% are reached in the subtropical gyres (Figure 2e). Lowest modeled annual mean relative abundances can be found in the ocean's upwelling areas, especially in the equatorial Pacific cold tongue, where *G. ruber* (white) appears to be almost absent. The modeled distribution pattern of *T. sacculifer* is limited to the warm waters of the subtropics and tropics and is similar to the one of *G. ruber* (white). *Trilobatus sacculifer* shows highest modeled annual mean relative abundances ($> 60\%$) in the equatorial Pacific between $15°N$ and $15°S$ and exhibits low modeled annual mean relative abundances ($< 30\%$) in the coastal upwelling regions of the ocean basins (Figure 2f).

## 3.2 Modeled seasonal distribution

For each foraminiferal species, the month of modeled maximum production changes on average with temperature and consequently with latitude (Figure 3, Figure S2 in the Supplement). In the simulation, there is a general tendency for the maximum production peak of the cold-water species *N. pachyderma* to occur later in the year (i.e., during summer) for lower annual mean temperatures (Figures 3a and S2a). With increasing mean annual temperatures, however, the modeled peak timing occurs earlier in the year (i.e., during spring) (Figure 3a). For *N. incompta*, modeled maximum production is reached during late summer in the midlatitudes at lower temperatures and is shifted towards spring/early summer when temperatures increase (Figure S2b). In the low latitudes at high temperatures, however, *N. incompta* exhibits a constant flux pattern throughout the year (Figure 3b). The modeled peak timing of *G. bulloides* is similar to the modeled peak timing of *N. incompta*, where the highest modeled fluxes are reached later (earlier) in the year in the midlatitudes at lower (higher) temperatures (Figure S2c). In the warm waters (of the tropics), *G. bulloides* exhibits year-round a rather uniform flux pattern (Figure 3c). In the model, both *N. incompta* and *G. bulloides* show indications of a double peak in their timing that is shifted towards the first half of the year when temperatures rise (Figures 3b and 3c). This earlier-when-warmer pattern is also indicated in the modeled peak timing of *N. pachyderma* (Figure 3a). *Globigerinoides ruber* (white) shows a uniform flux pattern all year round in the warm waters of the world ocean in the subtropical/tropical regions (Figure S2d). In colder waters (e.g., towards higher latitudes), modeled peak fluxes of *G. ruber* (white) are reached in late summer/fall (Figure 3d). A similar seasonal pattern in the modeled peak timing is evident for the tropical species *T. sacculifer* with constant fluxes occurring year-round at high temperatures in the low latitudes (Figure S2e). At lower ambient temperatures, modeled peak fluxes of *T. sacculifer* occur during fall (Figure 3e). For both *G. ruber* (white) and *T. sacculifer*, the modeled peak timing is shifted to later in the year when the surroundings become colder (Figures 3d and 3e).

To allow for a global comparison of the modeled and observed flux seasonality, we standardized peak amplitudes for each foraminiferal species, i.e., the species' maximum concentration divided by its annual mean. This reveals that the timing of the modeled foraminiferal peak abundances varies with temperature, but all five species exhibit an almost constant peak amplitude in their preferred thermal habitat. Outside their preferred living conditions, modeled peak amplitudes considerably increase for most of the species (Figure 3), thus, the species experience a strong deviation from their annual mean living conditions and likely occur only at times when the ambient conditions are (close to) their optima. For the warm-water species *G. ruber* (white) and *T. sacculifer*, peak amplitudes rise when the ambient temperatures fall below $20\,°C$ (Figures 3d and 3e). The peak amplitude of *G. bulloides* increases noticeably with mean annual temperatures falling below $10\,°C$ (Figure 3c). By contrast, when ambient temperatures exceed $25\,°C$, the peak amplitude of *N. incompta* increases (Figure 3b). For the cold-water species *N. pachyderma*, the relation between peak amplitudes and mean annual temperatures is more complex (Figure 3a).

## 3.3 Modeled vertical distribution

Among the three major ocean basins the modeled vertical distribution of each considered planktonic foraminiferal species shows similar patterns in the annual mean (Figure 4). The temperate/cold-water species (i.e., *G. bulloides*, *N. incompta*, and *N.*

*pachyderma*) occur from the surface down to about $200\,\text{m}$ water depth (Figures 4a, 4b and 4c). *Neogloboquadrina pachyderma* is consistently present in the top few $100\,\text{m}$ of the water column in the high latitudes and absent in the subtropical/tropical regions. In the polar waters of the three ocean basins, modeled maximum annual mean concentrations are found at the surface and deeper toward lower latitudes. The highest modeled annual mean concentrations of *N. pachyderma* are, however, located in the subpolar gyres between 0 and $75\,\text{m}$ water depth (Figure 4a). *Neogloboquadrina incompta* is in general present between $60\,°\text{N}$ and $60\,°\text{S}$ with the modeled annual mean concentration reaching its maximum at around $100\,\text{m}$ water depth. In the mid- to higher latitudes, *N. incompta* is found from the surface to $\sim 200\,\text{m}$ water depth in the Atlantic, Indian, and Pacific Oceans, but seems to be rarely present in the respective uppermost water layers (i.e., between 0 and $\sim 75\,\text{m}$) of the tropics. However, the modeled annual mean concentration increases with depth especially from the subpolar regions toward the equator (Figure 4b). As for *N. incompta*, *G. bulloides* has been consistently found from the surface to $\sim 200\,\text{m}$ water depth between about $60\,°\text{N}$ and $60\,°\text{S}$ (Figure 4c). Depending on the ocean basin, modeled maximum annual mean concentrations of *G. bulloides* are either mainly reached at the surface (i.e., in the Indian and Pacific Oceans) or at depth (i.e., in the Atlantic Ocean), but also at around $100\,\text{m}$ water depth in the subpolar regions of the three chosen transects. Both, *N. incompta* and *G. bulloides*, show highest modeled annual mean concentrations between $30\,°$ and $60\,°$ latitude (Figures 4b and 4c).

The warm-water species, *G. ruber* (white) and *T. sacculifer*, are found between the surface of each ocean basin and $\sim 100\,\text{m}$ water depth, thus occurring in a shallower depth range compared to *N. pachyderma*, *N. incompta*, and *G. bulloides* (Figures 4d and 4e). Among all five planktonic foraminiferal species, *G. ruber* (white) exhibits on average the highest modeled annual mean concentrations along the transects (Figure 4). This species is confined to the subtropical/tropical regions of the ocean basins with the highest modeled annual mean concentrations occurring between $\sim 15\,°$ and $30\,°$ latitude and the lowest around the equator (Figure 4d). Along the three chosen transects, modeled maximum annual mean concentrations of *G. ruber* (white) are almost consistently reached at the surface in the low latitudes and at around $60\,\text{m}$ water depth in those areas, where the highest modeled abundance of this species occurs. *Trilobatus sacculifer* also occurs predominantly between $30\,°\text{N}$ and $30\,°\text{S}$ with modeled annual mean concentrations gradually decreasing with depth. Compared to the other planktonic foraminiferal species, *T. sacculifer* exhibits a rather uniform distribution pattern along the different transects (Figure 4e) with modeled maximum annual mean concentrations being primarily located at the surface.

### 3.4 Modeled seasonal variability of habitat depth

In the model, the depth of maximum production of each considered planktonic foraminifera changes over the course of a year (Figure 5). Towards higher latitudes, *N. incompta* and *N. pachyderma* show maximum abundances at lower depth levels compared to low and midlatitudes. In the polar regions, *N. pachyderma* occurs close to the surface during winter and descends through the water column from spring to summer with modeled maximum abundances being reached at $\sim 40\,\text{m}$ water depth in summer. In the subpolar regions, *N. pachyderma* is generally found between 50 and $100\,\text{m}$ water depth for almost the entire year except for the winter season, where highest modeled concentrations are reached close to the surface (Figure 5a). The modeled depth habitat of *N. incompta* increases from spring to summer and is shallower in winter in the subpolar regions (Figure 5b).

In the subtropics and tropics, however, *N. incompta* shows year-round highest modeled concentrations consistently below 90 m water depth.

*Globigerina bulloides* exhibits a relatively shallow habitat (i.e., up to $\sim 50$ m water depth) along the equator throughout the year (Figure 5c). In the subpolar regions, the depth of modeled maximum production of *G. bulloides* varies seasonally and, similar to *N. incompta*, is shallower during winter and deepest during summer. The modeled depth habitat of *G. ruber* (white) is mostly confined to the top 60 m of the water column and seems to be less variable compared to the temperate and cold-water species (Figure 5). In the midlatitudes and near the equator, highest modeled concentrations of *G. ruber* (white) occur close to the surface during almost the entire year, whereas in the subtropical/tropical regions, this species is most abundant below 20 m and shows a weak seasonal cycle, occurring deeper in late summer/early fall (Figure 5d). *Trilobatus sacculifer* exhibits the least variable depth habitat in the simulation among the five considered species and is consistently found close to the surface above 20 m water depth throughout the year (Figure 5e).

## 4 Discussion

### 4.1 Large-scale patterns

#### 4.1.1 Geographical range of planktonic foraminifera species

The predicted global distribution patterns of the five considered planktonic foraminiferal species are in good agreement with the core-top data (Figure 2a). This is remarkable, considering the simplifications that had to be used to facilitate the comparison, such as the use of a constant biomass to size scaling within a species and a constant size scaling among the species.

*Neogloboquadrina pachyderma* is most abundant in the polar-subpolar waters of the northern and southern hemispheres both in the model and in the core-top samples (Figure 2b). This cold-water species dominates the waters north of the Arctic Circle and south of the Antarctic Convergence with relative abundances exceeding 90% and is very rarely found in subtropical/tropical waters, which is also seen in the model output. Bé (1969), Bé and Tolderlund (1971), and Bé and Hutson (1977) showed that *N. pachyderma* mainly occurs in regions with sea surface temperatures (SSTs) below 10 °C, but is also present in the cold-temperate waters of, e.g., the subpolar gyres with relative abundances being reduced to 30-50%. Thus, in areas, which are influenced by warmer waters the abundance of this species decreases gradually. This is especially evident in the eastern North Atlantic Ocean, where the abundance of *N. pachyderma* is reduced to about 50% due to the influence of the warm Atlantic Water, which is transported northward by the North Atlantic Current (NAC) (Husum and Hald, 2012). In line with the observations, the modeled annual mean relative abundances of *N. pachyderma* also decrease with decreasing latitude and, hence, get reduced towards warmer surface waters (Figure 2b). Additionally, PLAFOM2.0 is able to reproduce the observed species' abundance pattern in the North Atlantic with a reduced relative abundance of <30% in the area, which is influenced by the NAC. Similar to PLAFOM (see Fraile et al., 2008) a slight deviation between the simulated and observed relative abundances of *N. pachyderma* at the edge of the species' distribution pattern is observed in the northern hemisphere. It has been shown that distinct genotypes discovered within this morphologically defined species exhibit different ecological preferences

(Darling et al., 2006; Morard et al., 2013). Thus the above mentioned minor discrepancy might partly arise due to the underlying model parameterizations, which are mainly based on the environmental preferences (i.e., temperature tolerance limits) of the *N. pachyderma* genotypes found in the Southern Ocean (for more details see Fraile et al., 2008), which differ genetically from the genotypes found in the North Atlantic and North Pacific Oceans (Darling et al., 2004, 2006, 2007).

The modeled global distribution patterns of *N. incompta* and *G. bulloides* agree to a broad extent with the observations (Figures 2c and 2d). Both species are predominantly found in the subarctic/-antarctic and transitional waters of the world oceans (with relative abundances >50%), where SSTs range between $10°$ and $18°$C (Bé and Tolderlund, 1971; Bé and Hutson, 1977). They are also highly abundant in the cool eastern boundary currents off Africa and South America (e.g., Bé and Tolderlund, 1971; Giraudeau, 1993; Darling et al., 2006) as well as in the eastern North Atlantic and occur continuously in a subantarctic

belt between $30°$S and the Antarctic Convergence (Bé, 1969; Bé and Tolderlund, 1971; Boltovskoy et al., 1996). In addition, high abundances (>40%) of *N. incompta* have been observed in the equatorial Pacific upwelling system and of *G. bulloides* in the Arabian Sea. In the model, *N. incompta* is confined to the subpolar belts at around $45°$ latitude, which matches the general distribution pattern seen in the core-top data, but the relative abundance is underestimated (here *N. incompta* accounts for <20% of the modeled assemblage compared to up to 50% in the observations; Figure 2c). The model prediction for *G. bulloides* shows

in accordance with the core-top samples higher abundances in the subantarctic belt (here the species accounts for up to 80% of the modeled assemblage) and in the (coastal) upwelling regions of the Atlantic and Pacific Oceans (Figure 2d). PLAFOM2.0, however, fails to fully capture the relative abundances in those areas, where the assemblages are usually dominated by *N. incompta* and *G. bulloides* (Figures 2c and 2d). For instance, in the Benguela upwelling system, *N. incompta* and *G. bulloides* together account locally for >60% of the total planktonic foraminifera population (Bé and Tolderlund, 1971; Giraudeau, 1993),

whereas in the model, both species account for $< 40\%$ of the assemblage. In fact, *N. incompta* is almost absent in the model simulation outside of the subpolar belts. Furthermore, in the western Arabian Sea, the modeled annual mean relative abundance of *G. bulloides* ranges between 10 and 20%, which corresponds to the lower end of the observed range varying between 20 and $\sim 50\%$ (Naidu and Malmgren, 1996). Additionally, it is evident that the model slightly overestimates the relative abundance of *G. bulloides* in the central subtropical/tropical waters of the ocean basins (Figure 2d). The apparent discrepancies between

the observations and PLAFOM2.0 arise, firstly, due to an overestimation of the modeled annual mean relative abundances of *G. bulloides*, in particular in the subpolar belt at around $45°$N, and of *G. ruber* (white) and *T. sacculifer* especially in the upwelling regions, and/or due to the overall underestimation of the occurrence of *N. incompta*, outside the subpolar belts. Secondly, since the model parameterizations are performed on a global scale, distinct genotypes (possibly having different environmental preferences) of *N. incompta* and especially *G. bulloides* (e.g., Kucera and Darling, 2002; Morard et al., 2013)

cannot be included in detail in the model, potentially resulting in the model-data-mismatch.

      The simulated global distribution patterns of *G. ruber* (white) and *T. sacculifer* compare favorably with the core-top samples (Figures 2e and 2f). Both species dominate the subtropical and tropical waters of the global ocean, together accounting for 75-100% of the total planktonic foraminiferal fauna (Bé and Tolderlund, 1971; Bé and Hutson, 1977). *Globigerinoides ruber* (white) is the most abundant species in the subtropical areas, where SSTs range between $21°$ and $29°$C, whereas *T. sacculifer*

shows highest relative abundances (>50%) in the tropics with SSTs between $24°$ and $30°$C (Bé and Hutson, 1977). Addi-

tionally, *G. ruber* (white) is also highly abundant (>50%) compared to *T. sacculifer* along the continental margins of the low latitudes (Figures 2e and 2f). However, in the coastal upwelling regions, *G. ruber* (white) and *T. sacculifer* are rarely found as cooler water masses influence their usual habitat (e.g., Thiede, 1975). Since both species thrive in warmer waters, their (relative) abundance gradually diminishes when transported towards the higher latitudes, thus being absent in the subpolar/polar

regions of the ocean basins. The model predictions for *G. ruber* (white) and *T. sacculifer* show in general similar patterns as the observations with higher loadings in the subtropical and tropical regions and a gradual decrease in the occurrence toward the poles (Figures 2e and 2f). PLAFOM2.0 is also able to reproduce the dominance of *G. ruber* (white) in the subtropics and of *T. sacculifer* around the equator; and together both species account for >70% of the modeled assemblage in the warm waters of the world ocean. Additionally, the reduction in the (relative) abundances in the upwelling regions (i.e., along the equatorial

Pacific and the coasts of South America and Africa) is likewise captured by the model. However, in those provinces dominated by *G. ruber* (white) and *T. sacculifer*, the relative abundances are underestimated in the model, whereas in the coastal upwelling regions, the species' abundances are slightly overestimated compared to the observations. Such deviations may result from the over- and/or underestimation of *G. bulloides* and *N. incompta* in the tropical/subtropical or upwelling regions (Figures 2c and 2d) or from the $1°$ model resolution leading to an inadequate representation of the coastal upwelling regions.

Thus, we consider that part of the model-data-mismatch may arise from uncertainty in the conversion of biomass to (relative) abundance, which is based on constant offsets approximated from sparse data (cf. Schmidt et al., 2004). Likely an even larger part of the discrepancies between the model and core-top data stems from the underlying model parameterizations applied on a global scale, which do not distinguish between distinct genotypes of the different species with potentially varying ecological preferences. Theoretically, this problem could be solved by parameterizing all known genotypes individually and approximat-

ing the total morphospecies abundance as the sum of its constituent genotypes. This would allow a comparison with sediment data, but not a diagnosis, since the sediment data provide no information on which genotypes are contained in the assemblages. Interestingly, the generally fair fit between the model and observations suggests that ecological differences between cryptic species are likely limited and that the model provides a useful first-order approximation of global species distribution.

### 4.1.2   Seasonality of planktonic foraminifera species

The meta-analysis of Jonkers and Kučera (2015), which is based on sediment trap data, revealed that the (spatially varying) seasonality of individual planktonic foraminifera is predominantly related to either temperature or the timing of primary productivity. For the temperate and cold-water species, such as *G. bulloides*, *N. incompta*, and *N. pachyderma*, one or two flux maxima have been observed, which occur earlier in the year at higher temperatures. This seasonal pattern is also to a large degree evident in the model results (Figures 3a-c and S2a-c). At lower temperatures (below $5°C$), the modeled season of

maximum production for the cold-water species *N. pachyderma* is predominantly reached in (late) summer, whereas in the comparatively warmer subpolar and transitional waters, the modeled peak season is shifted towards spring (Figures 3a and S2a). A similar pattern can be observed for *N. incompta* and *G. bulloides*. In line with Jonkers and Kučera (2015), none of the three species shows a clear dependency of the peak amplitude with temperature (Figure 3a-c). In the model, the temperate and cold-water species exhibit a shift in their peak timing, but do not considerably change their peak amplitude (except for *G.*

*bulloides* when temperatures fall below $5\,^{\circ}$C). Hence, the observed and predicted earlier-when-warmer pattern can most likely be sought to a large extent in the timing of the primary productivity rather than in a temperature dependence. Several studies showed that the seasonality of the temperate and cold-water planktonic foraminiferal species is closely tied to phytoplankton bloom events leading to an increased food supply (e.g., Fairbanks and Wiebe, 1980; Donner and Wefer, 1994; Wolfteich, 1994;

Kohfeld et al., 1996; Mohiuddin et al., 2002, 2004, 2005; Northcote and Neil, 2005; Asahi and Takahashi, 2007; Storz et al., 2009; Wilke et al., 2009; Jonkers and Kučera, 2015). In particular, the flux of *G. bulloides* reaches highest values in response to an increased food supply to a large extent associated with open ocean and/or coastal upwelling (e.g., Thiede, 1975; Curry et al., 1992; Wolfteich, 1994; Naidu and Malmgren, 1996; Kincaid et al., 2000; Mohiuddin et al., 2004, 2005; Storz et al., 2009). The warm-water species *G. ruber* (white) and *T. sacculifer* exhibit relatively uniform annual flux patterns with almost no seasonal

peak in the subtropical/tropical regions of the ocean basins (e.g., Deuser et al., 1981; Jonkers and Kučera, 2015). Similar to observations, the modeled timing of the low-amplitude peaks is random during the year in warm waters (Figures 3d-e and S2d-e). However, in colder waters, peak fluxes are concentrated towards fall and peak amplitudes increase considerably both in the observations and in the model (Figures 3d-e and S2d-e). This shift in the seasonality can most likely be linked to temperature. In the low latitudes, optimum temperatures prevail all year round, whereas further north-/southward those optimum thermal

conditions occur only during a short period later in the year. Thus, those species focus their flux into the warm season in colder waters (Figure 3d-e). This emerging behavior is consistent with observations from sediment traps (Jonkers and Kučera, 2015) and suggests that the seasonality of the warm-water species is driven by temperature rather than food availability, which is in agreement with observational studies (e.g., Wolfteich, 1994; Eguchi et al., 1999, 2003; Kincaid et al., 2000; Kuroyanagi et al., 2002; Mohiuddin et al., 2002, 2004; Storz et al., 2009; Jonkers and Kučera, 2015).

### 4.1.3   Spatial and temporal variability of depth habitats of planktonic foraminifera species

The modeled depth habitats of *N. pachyderma*, *N. incompta*, *G. bulloides*, *G. ruber* (white), and *T. sacculifer* differ and show (distinct) spatial and temporal variability in response to different environmental conditions (Figures 4 and 5). Plankton tow studies have shown that the vertical distribution of planktonic foraminifera is mostly affected by temperature, primary productivity, light availability, and thermal/density stratification of the upper water column (e.g., Fairbanks et al., 1982; Ortiz

et al., 1995; Schiebel et al., 2001; Field, 2004; Kuroyanagi and Kawahata, 2004; Salmon et al., 2015; Rebotim et al., 2017).

In line with the observations, the modeled depth distribution patterns indicate that the warm-water species *G. ruber* (white) and *T. sacculifer* occur at shallower depths compared to the temperate and cold-water species *G. bulloides*, *N. incompta*, and *N. pachyderma* (see Figures 4 and 5). In the model, both *G. ruber* (white) and *T. sacculifer* have been consistently found from the surface to $\sim 100\,\mathrm{m}$ water depth in the subtropical/tropical regions of the ocean basins (Figure 4d-e). In the tropics, they

are most abundant close to the surface, which agrees well with the observations. In the Arabian Sea and in the central tropical Pacific Ocean, both species have been mostly found in the upper $60\,\mathrm{m}$ (Peeters and Brummer, 2002; Watkins et al., 1996, 1998). In the transitional and subtropical waters, however, PLAFOM2.0 slightly underestimates the depth habitat of *G. ruber* (white) and *T. sacculifer* (Figures 4d-e and 5d-e) as they inhabit the upper $125\,\mathrm{m}$ in the western North Atlantic (Fairbanks et al., 1980) and/or consistently occur from 0 to $200\,\mathrm{m}$ water depth in the subtropical eastern North Atlantic (Rebotim et al.,

2017) or in the seas surrounding Japan (Kuroyanagi and Kawahata, 2004). Nevertheless, both species typically live close to the surface (above 100 m) (e.g., Bé and Hamlin, 1967; Fairbanks et al., 1982; Kemle-von Mücke and Oberhänsli, 1999; Schiebel et al., 2002; Wilke et al., 2009; Rippert et al., 2016), thus being associated with a shallow depth habitat, which is reproduced by the model. Since *T. sacculifer* and *G. ruber* (white) are algal symbiont-bearing species, they are most abundant in the photic

zone, where light intensities are highest, but also chlorophyll *a* concentrations and temperature control their habitat. Light intensity is especially important for the growth of *T. sacculifer* (Caron et al., 1982, 1987; Jørgensen et al., 1985; Bijma et al., 1990b; Watkins et al., 1998), whereas *G. ruber* (white) seems to be more affected by food availability (Peeters and Brummer, 2002; Field, 2004; Kuroyanagi and Kawahata, 2004; Wilke et al., 2009) rather than light. This is to some degree also indicated in our results, as on average the highest modeled concentrations of *T. sacculifer* occur at shallower depths compared to *G.*

*ruber* (white) (see Figures 4d-e and 5d-e). However, at some locations both model and observations show the reverse (see Figure S4 and, e.g., Rippert et al., 2016; Rebotim et al., 2017), indicating that this depth ranking is not globally valid. In comparison with the temperate and cold-water species, *G. ruber* (white) and *T. sacculifer* are most abundant in the model in waters with temperatures above 22 °C and absent, where temperature values drop below 15 °C (see Figure 4), reflecting the different temperature tolerance limits of the two species.

*Neogloboquadrina pachyderma*, *N. incompta*, and *G. bulloides* generally thrive in cold to temperate waters. In the model, the depth habitat of those species decreases with increasing latitude (Figure 4a-c), indicating a preferred habitat in the subsurface (see Figure 5a-c). This is consistent with the observations from several locations, where the three species have typically been found between 50 and 200 m water depth (e.g., Kohfeld et al., 1996; Mortyn and Charles, 2003; Kuroyanagi and Kawahata, 2004; Bergami et al., 2009; Wilke et al., 2009; Pados and Spielhagen, 2014; Iwasaki et al., 2017; Rebotim et al., 2017). In the

subtropical to subpolar regions, the highest modeled concentrations of *G. bulloides* occur, however, between 60 and 100 m, whereas in the tropics, maxima are reached close to the surface (Figures 4c and 5c). This agrees well with the observations: *G. bulloides* has been found to be tightly linked to phytoplankton bloom events occurring either at deeper depth layers associated with a deep chlorophyll maximum (DCM) (Fairbanks and Wiebe, 1980; Mortyn and Charles, 2003; Wilke et al., 2009; Iwasaki et al., 2017) or in the coastal and equatorial upwelling regions, where a shoaling of the species' habitat towards the

near-surface can also be related to high chlorophyll *a* concentrations (Ortiz et al., 1995; Watkins et al., 1998; Peeters and Brummer, 2002; Field, 2004; Kuroyanagi and Kawahata, 2004). *Neogloboquadrina incompta* is also highly abundant, where chlorophyll *a* concentrations are high, but, nevertheless has most often been observed at mid-depth (Ortiz et al., 1995; Mortyn and Charles, 2003; Field, 2004; Kuroyanagi and Kawahata, 2004; Iwasaki et al., 2017; Rebotim et al., 2017). In the model, *N. incompta* shows also highest concentrations between 30 and 120 m (Figures 4b and 5b), clearly inhabiting the subsurface. This

is especially evident in the tropics, where *N. incompta* is virtually absent in the near-surface layers, but present, albeit in low numbers, around 100 m water depth. The predictions show, in general, that *N. incompta* prefers warmer waters compared to *N. pachyderma* and, where the species co-exist, *N. incompta* inhabits for this reason shallower depths (Figures 4a-b and 5a-b). This agrees with the observations from the subarctic Pacific and the seas around Japan (Iwasaki et al., 2017; Kuroyanagi and Kawahata, 2004). *Neogloboquadrina pachyderma* is confined to the high latitudes with peak abundances occurring in the upper

100 m of the water column (Kohfeld et al., 1996; Stangeew, 2001; Mortyn and Charles, 2003; Kuroyanagi and Kawahata, 2004;

Bergami et al., 2009; Pados and Spielhagen, 2014) (partly associated with high chlorophyll *a* concentrations), which agrees well with the model results. Although *N. pachyderma* has been classified as a "deep dweller" in different studies (Bé, 1960; Boltovskoy, 1971; Hemleben et al., 1989; Simstich et al., 2003), this species appears to be more surface-restricted at higher latitudes (Carstens and Wefer, 1992; Kohfeld et al., 1996; Mortyn and Charles, 2003), which is also evident in the model results (Figures 4a and 5a).

Several studies showed that the depth habitat of planktonic foraminifera varies throughout the year in response to changing environmental conditions. Rebotim et al. (2017) identified an annual cycle in the habitat of *T. sacculifer* and *N. incompta* in the subtropical eastern North Atlantic. Both species appear to descend in the water column from winter to spring and reach their deepest habitat in spring to summer before ascending again to a shallower depth towards winter (Rebotim et al., 2017). It has been associated that *N. incompta* is affected by chlorophyll *a* concentrations, hence, the seasonal shift in its habitat depth could be related to food availability as a DCM develops in the summer months. In the Canary Islands region, *G. ruber* (white) and *G. bulloides* have been found at lower depth levels during winter, and during summer/fall, shell concentrations were highest at depth associated with the DCM (Wilke et al., 2009). However, *G. ruber* (white) did occur at moderate abundance levels throughout the year, whereas *G. bulloides* was only present in low numbers during wintertime in the study area of Wilke et al. (2009). Peeters and Brummer (2002) investigated the influence of a changing hydrography on the habitat of living planktonic foraminifera in the northwest Arabian Sea. During the southwest monsoon (occurring in summer), strong coastal upwelling associated with low SSTs and a near-surface chlorophyll maximum leads to high abundances of *G. bulloides* dominating the species assemblage in the uppermost part of the water column (Peeters and Brummer, 2002). In comparison, during the northeast monsoon (occurring in winter), a relatively warm nutrient-depleted surface mixed layer as well as a DCM develop resulting in high concentrations of *G. ruber* (white) and *T. sacculifer* near the surface, whereas the concentrations of *G. bulloides* are low and show a subsurface maximum between the DCM and the thermocline (Peeters and Brummer, 2002). Based on their findings, Peeters and Brummer (2002) conclude that the habitat depth of individual foraminifera strongly depends on the local hydrography controlling, i.a., the food availability. Watkins et al. (1998) also found high abundances of *G. bulloides* in the equatorial surface waters of the Pacific Ocean associated with higher primary productivity due to an intensified upwelling, but also with the zonal advection by the South Equatorial Current during La Niña conditions. In contrast, during El Niño conditions, *G. bulloides* has been absent in the central tropical Pacific (Watkins et al., 1996) due to unfavorable living conditions.

The change in the depth of modeled maximum production of each considered planktonic foraminifera throughout a year (Figure 5) agrees to a large extent with the observations. *Neogloboquadrina pachyderma* is almost constantly found below $50\,\mathrm{m}$ except during winter, where highest modeled concentrations occur close to the surface (Figure 5a). The shift in the simulated habitat depth most likely indicates that *N. pachyderma* is highly dependent on food availability (cf. Figure 5a), which coincides with observational studies, where this species has been extensively found at mid-depth during summer associated with the chlorophyll maximum (Kohfeld et al., 1996; Mortyn and Charles, 2003; Bergami et al., 2009; Pados and Spielhagen, 2014). The simulated change from a deeper to a shallower depth habitat of *N. incompta* in the subpolar regions over the course of a year could be strongly affected by the food supply by potentially following the seasonal distribution of phytoplankton. In

the low latitudes, modeled maximum concentrations of *N. incompta* are constantly reached below 90 m water depth, which might be attributed to the presence of a permanent DCM (Figure 5b), being a characteristic feature throughout the low latitudes (Mann and Lazier, 1996). *Globigerina bulloides*, however, is found year-round close to the surface along the equator in the model (Figure 5c), which, in line with the observations, can be associated with equatorial upwelling, but also the inclusion of the photosynthetic growth rate in the model could explain the occurrence of modeled maximum concentration values at lower depth levels due to higher light requirements compared to *N. incompta*. In the subpolar regions, the simulated depth habitat of *G. bulloides* varies seasonally, most likely following the chlorophyll maximum (Figure 5c). The model simulation indicates that the seasonal occurrence of both *G. ruber* (white) and *T. sacculifer* in colder regions, where they face suboptimal environmental conditions, is limited to the warm surface layer during the warm season (Figure 5d-e). Even in the low latitudes, both species exhibit a weak seasonal cycle in their simulated depth habitat, which is more pronounced for *G. ruber* (white) (Figure 5d), indicating some influence of primary productivity, which also agrees with the observations (Peeters and Brummer, 2002; Field, 2004; Kuroyanagi and Kawahata, 2004; Wilke et al., 2009). In line with Kuroyanagi and Kawahata (2004), our results suggest that *T. sacculifer* seems to prefer living in warmer waters than *G. ruber* (white) year-round (Figure 5e) and is most abundant at shallow depths, where the light intensity is highest. Our results, thus, confirm the observations by Jonkers and Kučera (2015) that both *G. ruber* (white) and *T. sacculifer* adapt to changing environmental conditions by adjusting their seasonal and vertical habitat to local circumstances. This emerging behavior can have important implications for paleoceanographic reconstructions (Jonkers and Kučera, 2017).

We find that the modeled depth habitats of the five considered foraminiferal species are in agreement with the relative ranking of their apparent calcification depths, but the inferred absolute values of calcification depth are often deeper or show a broader range of depths (e.g., Carstens and Wefer, 1992; Kohfeld et al., 1996; Ortiz et al., 1996; Bauch et al., 1997; Schiebel et al., 1997; Ganssen and Kroon, 2000; Peeters and Brummer, 2002; Anand et al., 2003; Simstich et al., 2003; Nyland et al., 2006; Jonkers et al., 2010, 2013; van Raden et al., 2011). This is not surprising, because PLAFOM2.0 does not model species' ontogeny and cannot capture processes related to ontogenetic depth migration (e.g., Fairbanks et al., 1980; Duplessy et al., 1981). The same limitation applies to estimates of living depth derived from plankton tow data, which often appears to deviate from apparent calcification depths (e.g., Duplessy et al., 1981; Rebotim et al., 2017). Nevertheless, as a first essential step in understanding the variability in calcification depths, PLAFOM2.0 provides a powerful tool that can aid the interpretation of proxy records.

## 4.2 Detailed comparison with observations

The emergence of seasonal and vertical habitat patterns consistent with observational data provides important support for our modeling approach. Nevertheless, a more detailed comparison with observations is warranted to gain further insight into the model behavior. However, when comparing observational data and model output, one has to bear in mind several caveats. These can be broadly categorized into four groups: i) model resolution, ii) model parameterization, iii) model hierarchy, and iv) analytical constraints on the observations.

i) The model resolution has limits on temporal and spatial scales when compared to sediment trap and plankton tow data. Most sediment trap time series span at most a few years and hence represent short time series that are potentially aliased/biased by inter-annual, seasonal, and/or monthly variability. Similarly, plankton tow samples represent snapshots (of one particular day) and the prevailing environmental conditions during their actual sampling time cannot be fully captured by the model. In fact, the model is forced using climatological data, thus representing a long-term average response that ignores such short-term variability. Additionally, because of the employed $1°$ model resolution, only the nearest model grid points rather than the exact locations of the sediment traps and plankton tows (especially along the coast lines) can be considered. This potentially results in different environmental conditions influencing the seasonality and depth habitat of planktonic foraminifera compared to the observations. The observational records are, additionally, affected by sub-grid phenomena (such as mesoscale eddies and/or steep gradients). For instance, Gulf Stream cold core rings transport large planktonic foraminiferal assemblages into the generally nutrient-poor Sargasso Sea (Fairbanks et al., 1980). In addition, Beckmann et al. (1987) found that an increase in zooplankton (including planktonic foraminifera) productivity coincided with an increase in phytoplankton biomass in a cold-core eddy in the eastern North Atlantic. The $1°$ resolution of the underlying model configuration leads to an inadequate representation of such sub-grid processes and, thus, their impact cannot be fully reflected by the CESM1.2(BGC+PLA) configuration.

ii) The underlying model parameterizations used in PLAFOM2.0 are limited in regard to taxonomic resolution and species' ontogeny. Different genotypes of one species could exhibit different habitat preferences (e.g., Kuroyanagi and Kawahata, 2004), which are not captured by PLAFOM2.0 since the model parameterizations do not resolve the different known genotypes of the considered planktonic foraminiferal species. Several studies from different areas also showed that the main habitat depth of some species increases from the surface to deeper water layers during shell growth (Peeters and Brummer, 2002; Field, 2004; Iwasaki et al., 2017). This vertical migration of planktonic foraminifera during ontogeny cannot be reproduced by PLAFOM2.0 as the model parameterizations do not include the individual species' life cycles.

iii) The underlying complex model configuration consists of three major model components (i.e., the POP2 ocean model, the BEC ecosystem model, and PLAFOM2.0), which follow a certain model hierarchy by interacting differently with each other. Both the BEC model and PLAFOM2.0 run within POP2 (see Moore et al., 2013; Lindsay et al., 2014; this study), which provides the temperature distribution used to determine, i.a., the phytoplankton, zooplankton, and/or foraminifera carbon concentrations. It was shown that POP2 exhibits several temperature biases (e.g., Danabasoglu et al., 2012, 2014). These include large warm SST biases originating in the coastal upwelling regions of North and South America and of South Africa, colder-than-observed subthermocline waters in the equatorial Pacific as well as cold temperature biases of up to $7°C$ in the North Atlantic emerging throughout the water column (see Figure S5 and Danabasoglu et al., 2012, 2014). These temperature biases influence the foraminiferal distributions directly and indirectly by affecting the distributions of their food sources in the BEC model. In addition, the BEC model also exhibits several biases, such as higher-than-observed (lower-than-observed) surface nutrient and chlorophyll concentrations at low (high) latitudes (Moore et al., 2013), implying potential misrepresentations of the modeled phytoplankton and zooplankton distributions,

likely influencing the foraminiferal carbon concentrations. The inferred importance of temperature and food availability (estimated by POP2 and/or the BEC model) in PLAFOM (see Fraile et al., 2008; Kretschmer et al., 2016) on the distribution of planktonic foraminifera implies that each model component is important for an accurate representation of the foraminifera distribution. Therefore, it is difficult to unequivocally differentiate between the different model components of the CESM1.2(BGC+PLA) model configuration and their individual share likely leading to the model-data-mismatch.

iv) The analytical constraints regarding the observational records include drift due to (sub-grid) ocean processes, distinction between live and dead specimens, collection depths, and taxonomic agreement among different studies. For instance, a few sediment trap samples might be compromised due to the collection of sinking particles derived from different regions of the surface ocean, being transported through eddies and/or ocean currents (Mohiuddin et al., 2004). Strong current velocities sometimes associated with eddies could lead to a tilt in the moored sediment trap, resulting in fewer material being collected by the trap (Yu et al., 2001). The impact of eddies might, thus, hamper the observed season of maximum production of planktonic foraminifera as well as their average living depth. A further uncertainty in the plankton tow data arises from the identification of living cells, because dead cells with cytoplasm collected at depth still appear as living and lead to a shift in the average living depth to greater depth (Rebotim et al., 2017). Uneven sampling intervals of the tows also result in a bias in the observed depth habitat (cf. Figure S4). Additionally, a taxonomic consistency within the observational data is assumed, which cannot be guaranteed as different researchers have been responsible for the data collection (see Tables S1 and S2).

With these caveats in mind, we compare the results of PLAFOM2.0 with 26 sediment trap records and 45 plankton tow samples from all oceans (Figure 1b, Tables S1 and S2). Note that the results of the point-by-point comparative analysis for each site and species are given in the Supplement (see Figures S3 and S4).

The peak season of the temperate and cold-water species (*G. bulloides*, *N. incompta*, and *N. pachyderma*) is shifted from late summer in the higher latitudes towards spring at the more equatorward directed locations in the subpolar and transitional water masses both in the model and in the sediment trap records (Figure 6a, Table S3a). The modeled peak amplitudes of those species remain almost constant at rather low values independent of the considered region. In the sediment traps, however, the peak amplitude values are higher and more diverse and also no clear pattern is evident neither for the species nor for the provinces changing with latitude (Figure 6b, Table S3b). In line with the plankton tow samples, *N. pachyderma*, *N. incompta*, and *G. bulloides* occur to a large extent below $50\,\mathrm{m}$ water depth from the cold high latitudes to the warmer provinces. However, the modeled ALDs (ranging between 20 and $100\,\mathrm{m}$) are considerably lower than the observed ALDs, which spread over $250\,\mathrm{m}$ (Figure 6c, Table S4). The warm-water species *G. ruber* (white) and *T. sacculifer* occur year-round in the subtropical/tropical regions with no distinct preference for a particular season both in the observations and in the model simulation (Figure 6a, Table S3a). In the transitional waters, however, their peak fluxes are consistently concentrated into fall, leading to higher peak amplitude values at least in the model (Figure 6b, Table S3b). Throughout the tropics and subtropics, the modeled peak amplitudes remain constant at low values. In the sediment trap records, however, the peak amplitudes are higher (compared with PLAFOM2.0) and vary within both species and within each province (Figure 6b). In the tropics, *G. ruber* (white) and *T.*

*sacculifer* occur primarily close to the surface with ALDs below 50 m both in the model simulation and in the plankton tow records (Figure 6c, Table S4). In fact, the predicted ALD values (consistently ranging between the surface and 55 m) are lower in comparison with the observations in the transitional and subtropical waters and, accordingly, do not exhibit a similar value range as the plankton tow records.

In general, the point-by-point comparison between the observations and the model simulation reveals that the peak seasons are well predicted by PLAFOM2.0. The predicted peak amplitudes and average living depths also show realistic trends, but the model tends to underestimate the magnitude of these trends (cf. Figure 6). Additionally, some sediment trap flux time series of the temperate and cold-water planktonic foraminiferal species show two seasonal peaks a year (cf. Jonkers and Kučera, 2015) (see Figures S3 and 7a). PLAFOM2.0 is, however, not always able to faithfully reproduce this bimodal pattern (cf. Figures S3

and 7a). In the following, we try to identify the causes of discrepancies between the observations and predictions by comparing the model output with exemplarily chosen sediment trap records and/or plankton tow samples of three different locations in each case (Figure 7).

The timing of flux pulse(s) of the temperate and cold-water species has, in general, been linked to the timing of the peak in primary productivity (e.g., Fairbanks and Wiebe, 1980; Donner and Wefer, 1994; Wolfteich, 1994; Kohfeld et al., 1996;

Mohiuddin et al., 2002, 2004, 2005; Northcote and Neil, 2005; Asahi and Takahashi, 2007; Storz et al., 2009; Wilke et al., 2009; Jonkers and Kučera, 2015). It is known from studies of the North Atlantic Ocean that phytoplankton seasonality changes with latitude, featuring a single spring bloom in the polar and subpolar Atlantic, a bimodal pattern (one large peak in spring, one smaller peak in fall) in the temperate North Atlantic, a single fall/winter bloom in the subtropical Atlantic, and no prominent seasonal cycle in the tropical Atlantic (e.g., Colebrook, 1979, 1982; Taboada and Anadón, 2014; Friedland et al., 2016).

The ecosystem model (providing the food information for PLAFOM2.0), however, does not faithfully reproduce the observed seasonal cycle in the primary productivity (cf. Figure 4 in Moore et al., 2002b). The simulated (depth integrated) chlorophyll concentration, used as an indicator for productivity, does only in parts show two cycles per year (Figure 7a-c). Nevertheless, the peak timings of the (depth integrated) foraminifera concentration follow the maxima in the primary productivity. For instance, the modeled maximum production peak of *N. pachyderma* at site PAPA coincides with a peak in the diatom concentration

(Figure 7a), *N. incompta* reaches its maximum in the simulation more likely at depth at site CP following a DCM (Figure 7b), and the predicted spring and/or fall peak of *G. bulloides* at site WAST occurs slightly after the peak in the main food concentration or the maximum in the chlorophyll concentration (Figure 7c). However, the ecosystem model seems to underestimate the seasonality in the primary productivity, which most likely leads to the model-data-mismatch in the seasonal pattern of the planktonic foraminifera concentration. Additionally, the variability of planktonic foraminifera carbon biomass produced by

PLAFOM2.0 is in general too low compared to the observations. This mismatch can either be explained by misrepresentations of the foraminiferal carbon biomass or of the foraminifera response (to the environmental forcing) in the model parameterizations or by an underestimation of the driving factors (i.e., especially the main food sources as outlined above). The depth habitat of planktonic foraminifera depends on several environmental and ontogenetic factors (e.g., Fairbanks and Wiebe, 1980; Fairbanks et al., 1982; Schiebel et al., 2001; Simstich et al., 2003; Field, 2004; Salmon et al., 2015; Rebotim et al., 2017). The

simulated vertical distribution patterns can also be related to food availability and temperature (Figure 7d-f). For instance, at

station PS55-063, peak abundances of *N. pachyderma* are reached in the top 50 m in the model corresponding to the highest diatom concentrations (Figure 7d). At station MOC1-28, the predicted depth profile of *G. ruber* (white) coincides with the vertical distribution pattern of zooplankton with maximum concentrations occurring over the top 50 m of the water column (Figure 7e). At station SO225-21-3, the modeled species' concentration of *T. sacculifer* decreases gradually with depth following the zooplankton distribution, but also temperature (Figure 7f). However, the simulated depth profiles differ from the observations, which is also indicated by the differences in the ALDs. In PLAFOM2.0, the foraminiferal species do not occur below 200 m water depth (cf. Figures 4 and 7d-f) most likely being restricted through food availability and the ambient temperatures. Thus, depending on the vertical resolution of the sampling intervals of a plankton tow sample the predicted ALD is very likely lower by several meters than the observed ALD. In summary, PLAFOM2.0 is able to reproduce the observed species' behavior with regard to time and depth on a local scale, but is strongly dependent on the input variables (e.g., temperature and the different food sources) provided by both the ocean and the ecosystem model and is, thus, limited in its capability to match the observations.

Keeping the caveats regarding the model resolution, model parameterizations, model hierarchy, and analytical constraints on the observations in mind, the model-data-mismatch might, however, be reduced by a higher model resolution (in time and space), which would in turn increase the computational costs. A higher taxonomic resolution of the considered species (resulting in an increased number of passive tracers and likewise degrees of freedom) and by explicitly parameterizing the ontogeny of each individual planktonic foraminifera, thus, by considering the changes in the species' life cycles with depth (e.g., Bijma et al., 1990a; Bijma, 1991; Bijma and Hemleben, 1994; Bijma et al., 1994; Hemleben and Bijma, 1994; Schiebel et al., 1997), could considerably improve the model. The discrepancies between the model and the observations could, additionally, be minimized by including better ecological constraints on planktonic foraminifera species and their habitat, e.g., by introducing more phytoplankton and zooplankton functional groups in the ecosystem model to better resolve species' food preferences, which would, however, result in an increased computational cost. Nevertheless, additional knowledge about the factors controlling the habitat of planktonic foraminifera in time and space based on culturing experiments and field studies are needed for an optimization and better validation of the current model version. In addition, due to the model complexity it is not trivial to determine which model component (i.e., POP2, BEC or PLAFOM2.0) contributes to what extent to the model-data-mismatch. Determining this would require a suite of sensitivity experiments with each model component, which should be considered for future work. The model produces, nonetheless, seasonal and vertical abundance patterns that are consistent with our current understanding and which emerge from the model without an explicit parameterization of abundance in time and space. PLAFOM2.0, thus, represents a major step forward from the previous model version and can be used to assess paleoclimate information in a better way.

## 5    Conclusions

A new version of the dynamic planktonic foraminifera model PLAFOM (PLAFOM2.0) has been developed and combined with the CESM1.2(BGC) model configuration to simulate species-specific seasonal and depth habitats for *N. pachyderma*, *N.*

*incompta*, *G. bulloides*, *G. ruber* (white), and *T. sacculifer* on a global scale. In comparison with the original approach, where only species' concentrations in the surface mixed layer were predicted, PLAFOM2.0 includes a vertical component and, thus, predicts species' distribution patterns in space and time more realistically.

PLAFOM2.0 produces spatially and temporally coherent abundance patterns, which agree well with available observations. The model configuration faithfully reproduces the areal extent of the species. In line with core-top data, the modeled global distribution of each foraminifera changes with latitude. Additionally, PLAFOM2.0 successfully predicts the patterns in the timing of peak fluxes of planktonic foraminiferal species on a global scale. The earlier-when-warmer pattern for the temperate and cold-water species and the flux focusing at low temperatures of warm-water species, as inferred from observations by Jonkers and Kučera (2015), have emerged from the model.

Although an explicit parameterization of the vertical dimension is lacking, the model successfully predicts the preferred habitat depth of the individual planktonic foraminiferal species as well as the spatial and temporal variability in the vertical abundance. In accordance with the available observations, the warm-water species *G. ruber* (white) and *T. sacculifer* consistently occur close to the sea surface year-round in the tropics/subtropics, whereas the depth habitat of the colder-water species *N. pachyderma*, *N. incompta*, and *G. bulloides* changes seasonally in the polar/subpolar regions. During the cold season these species occur near-surface, while during the warmer season they descend in the water column to be found up to $120\,\mathrm{m}$ water depth or even below most likely following the chlorophyll maximum.

In general, paleoceanographic reconstructions based on planktonic foraminifera are hampered by the fact that the environmental signal preserved in their shells is the result of both habitat and climate change. The two effects are difficult to separate without independent data. PLAFOM2.0 presents a powerful tool to address this issue and can contribute to more meaningful comparisons of climate model results and paleoclimate reconstructions, ultimately aiding to the understanding of mechanisms of climate change.

*Code and data availability.* All model data can be obtained from the PANGAEA database (www.pangaea.de). The model code is available upon request from the corresponding author (Kerstin Kretschmer, kkretschmer@marum.de).

*Competing interests.* The authors declare that they have no conflict of interest.

*Acknowledgements.* This paper has benefited from the constructive comments and suggestions of Inge van Dijk, Jelle Bijma and two anonymous reviewers as well as of the handling associated editor Lennart de Nooijer. We are grateful to Graham Mortyn for providing the plankton tow data from the Atlantic sector of the Southern Ocean. We would like to thank Gerlinde Jung and Jeroen Groeneveld for their helpful advice. The model integration has been performed at the North-German Supercomputing Alliance (HLRN). This project was supported by the DFG (Deutsche Forschungsgemeinschaft) through the International Research Training Group IRTG 1904 ArcTrain and the DFG Research Center/Cluster of Excellence "The Ocean in the Earth System".

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

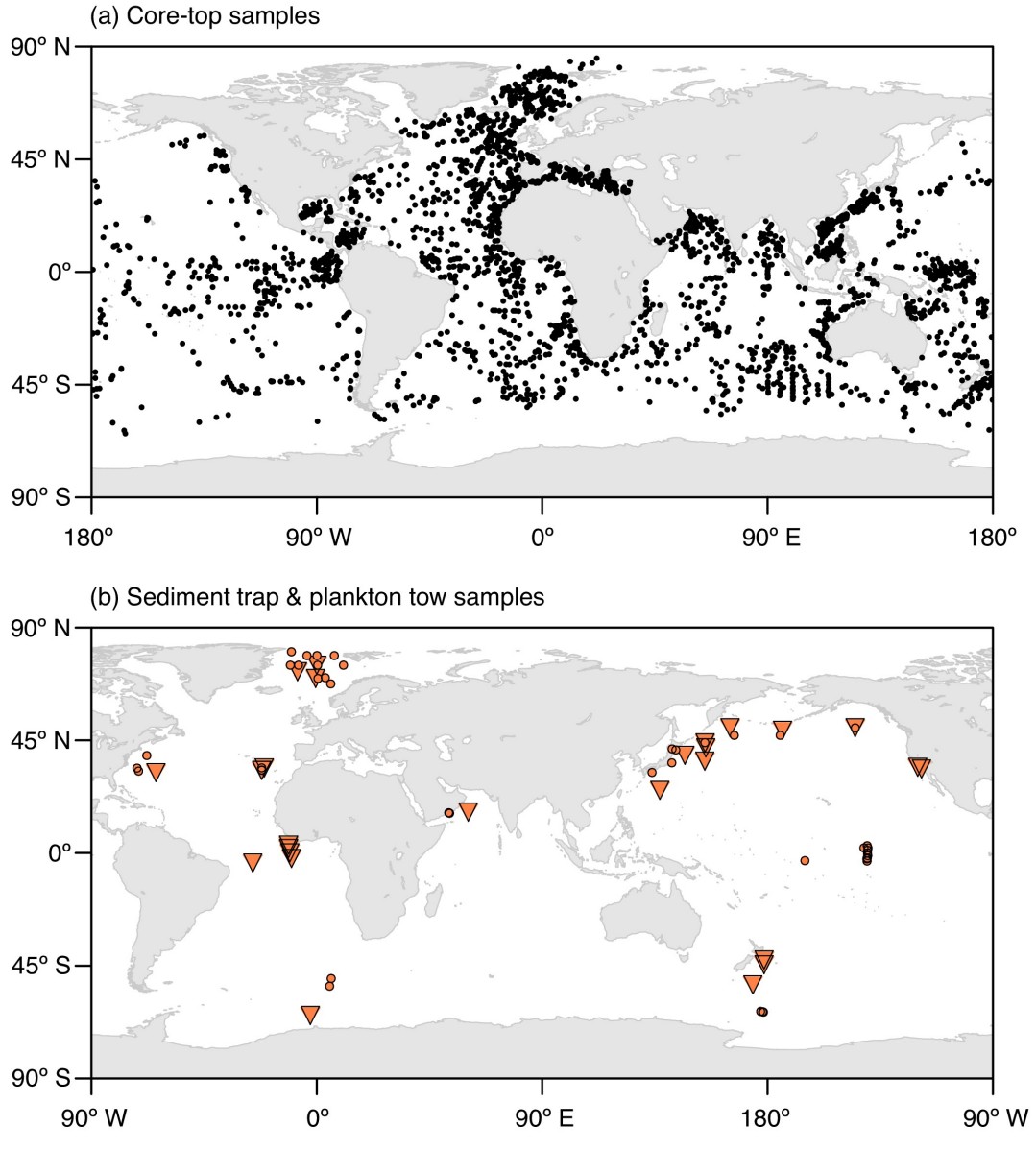

**Figure 1.** Locations of (a) the core-top samples with planktonic foraminifera counts and (b) the plankton tow (orange circles) and sediment trap (orange triangles) samples used for the model validation. The map in Figure 1a shows a combination of the data sets of Prell et al. (1999), Pflaumann et al. (1996, 2003), and Kucera et al. (2005). The respective information on the sediment trap and plankton tow data shown in Figure 1b is given in Tables S1 and S2 in the Supplement.

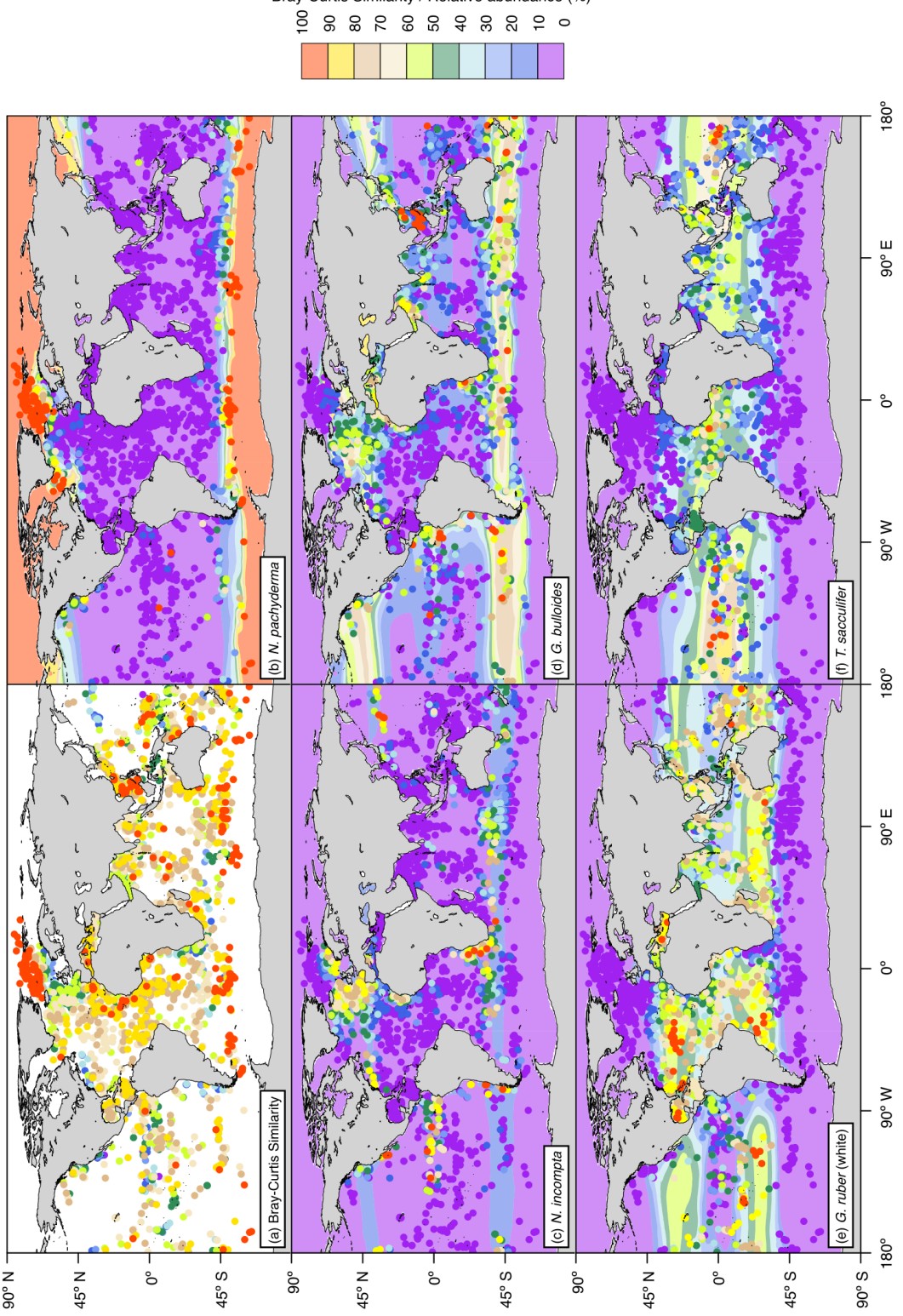

**Figure 2.** (a) Bray-Curtis index of similarity (in %) between the relative abundances of the modeled and core-top data as well as relative abundances of the depth integrated modeled annual mean concentration (pale-colored contours; in % carbon biomass) and of the core-top samples (circles; in % individuals) for (b) *N. pachyderma*, (c) *N. incompta*, (d) *G. bulloides*, (e) *G. ruber* (white), and (f) *T. sacculifer*. The relative abundances consider only the five foraminiferal species included in PLAFOM2.0. In addition, to account for the different sizes of each foraminiferal species, we multiplied the modeled annual mean concentration of each species with their relative size (Table 2) and subsequently calculated the depth integrated species' annual mean concentrations relative to the total modeled foraminiferal carbon biomass. Note that we are aware that for a small number of core-top samples the relative abundances of the individual planktonic foraminiferal species are overestimated due to the recalculations by only considering *N. pachyderma, N. incompta, G. bulloides, G. ruber* (white), and *T. sacculifer* rather than the whole assemblage. However, the overall general pattern does not change and can, thus, be used for the model-data comparison.

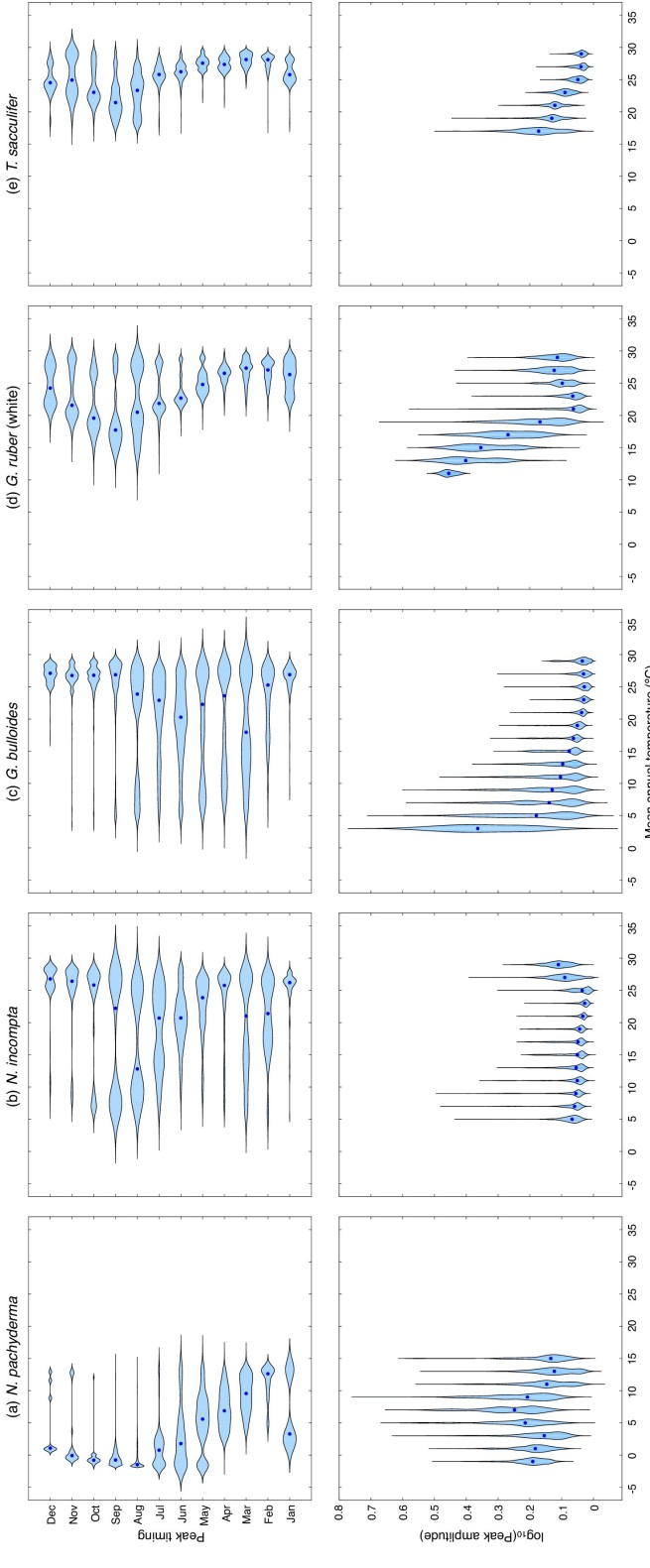

**Figure 3.** Violin plots showing the dependence of the modeled peak timing (top row) and/or the modeled peak amplitude (bottom row) on the annual mean temperature (in °C) averaged over the top 55 m of the water column for (a) *N. pachyderma*, (b) *N. incompta*, (c) *G. bulloides*, (d) *G. ruber* (white), and (e) *T. sacculifer*. The modeled peak timing is given in months and the modeled peak amplitude has been log-transformed. The blue dots represent the respective median values. Note that the peak timings of each species from the southern hemisphere have been transformed to northern hemisphere equivalents by adding or subtracting 6 months.

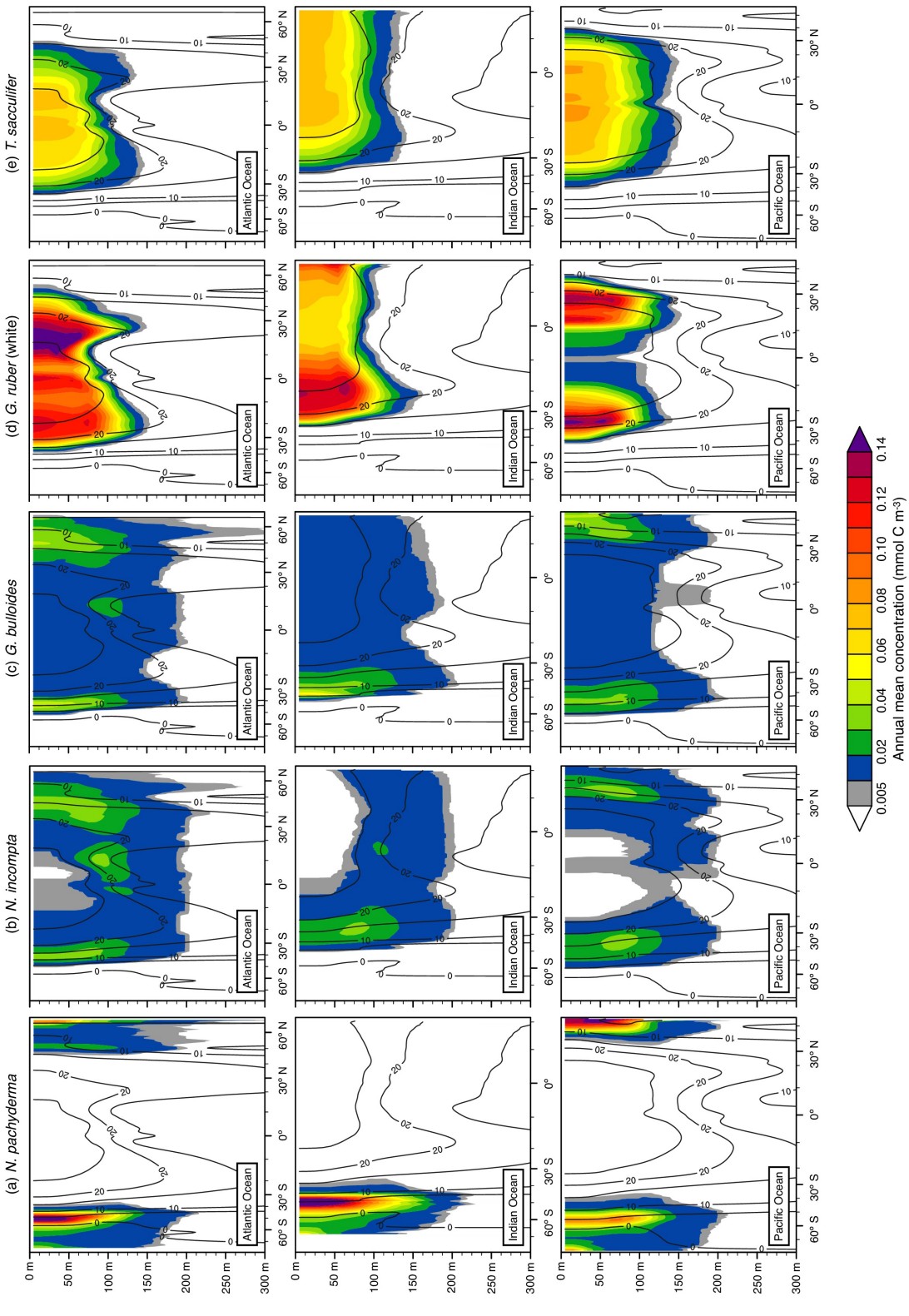

**Figure 4.** Depth transects of the modeled annual mean concentration (in $mmol\,C\,m^{-3}$) along $\sim 27\,°W$ in the Atlantic Ocean (top row), $\sim 71\,°E$ in the Indian Ocean (middle row), and $\sim 162\,°W$ in the Pacific Ocean (bottom row) over the top $300\,m$ for (a) *N. pachyderma*, (b) *N. incompta*, (c) *G. bulloides*, (d) *G. ruber* (white), and (e) *T. sacculifer*. The black contour lines indicate the annual mean temperature estimates (in $°C$). The blank areas denote where a species is absent.

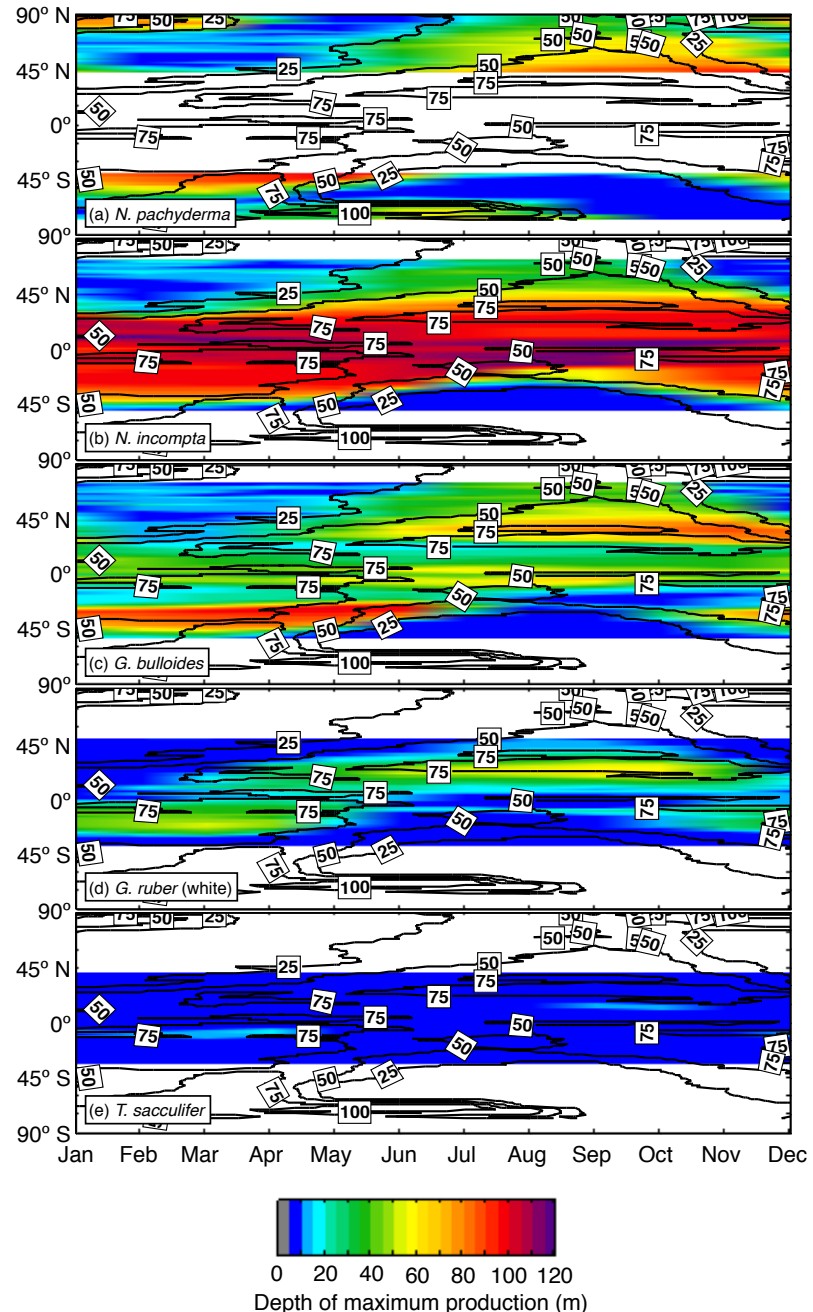

**Figure 5.** Zonal average of the depth (in m) at which the modeled maximum production of (a) *N. pachyderma*, (b) *N. incompta*, (c) *G. bulloides*, (d) *G. ruber* (white), and (e) *T. sacculifer* occurs over time. The black contour lines indicate the zonal average of the (seasonally varying) depth of the chlorophyll maximum (in m). The blank areas denote where a species is absent.

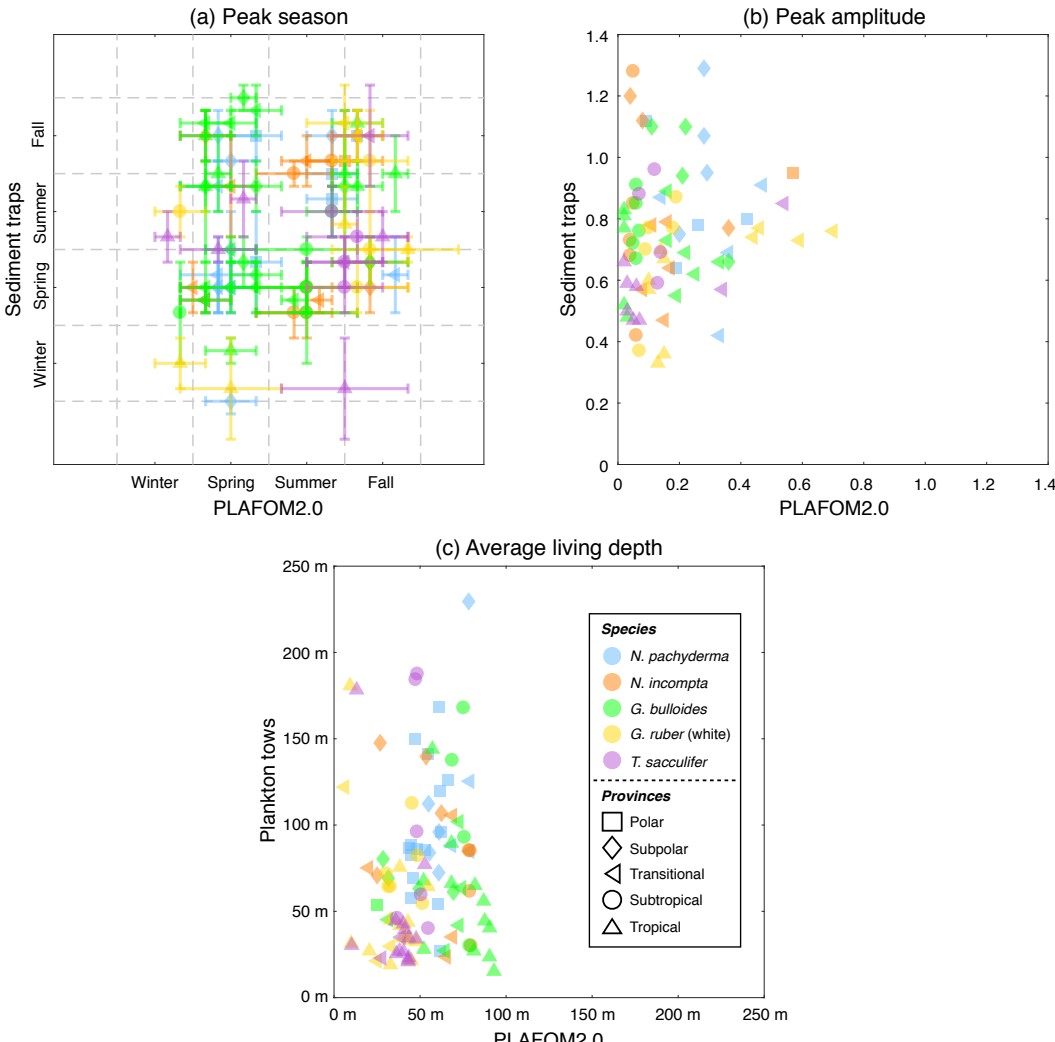

**Figure 6.** (a) Peak seasons (i.e., caloric season of the maximum production), (b) peak amplitudes (i.e., maximum production divided by annual mean), and (c) average living depths (in m) for *N. pachyderma* (light blue), *N. incompta* (orange), *G. bulloides* (green), *G. ruber* (white) (gold), and *T. sacculifer* (orchid) based on either the sediment trap data (given in Table S3) or the plankton tow data (given in Table S4) vs. PLAFOM2.0. The symbols represent the polar (squares), subpolar (diamonds), transitional (left-pointing triangles), subtropical (circles), and tropical (upward-pointing triangles) provinces of the ocean, respectively. The symbols in (a) indicate the month corresponding to the mid-season and the error bars refer to the overall time frame given in Table S3a. Note that the observed and modeled peak amplitudes in (b) have been log-transformed.

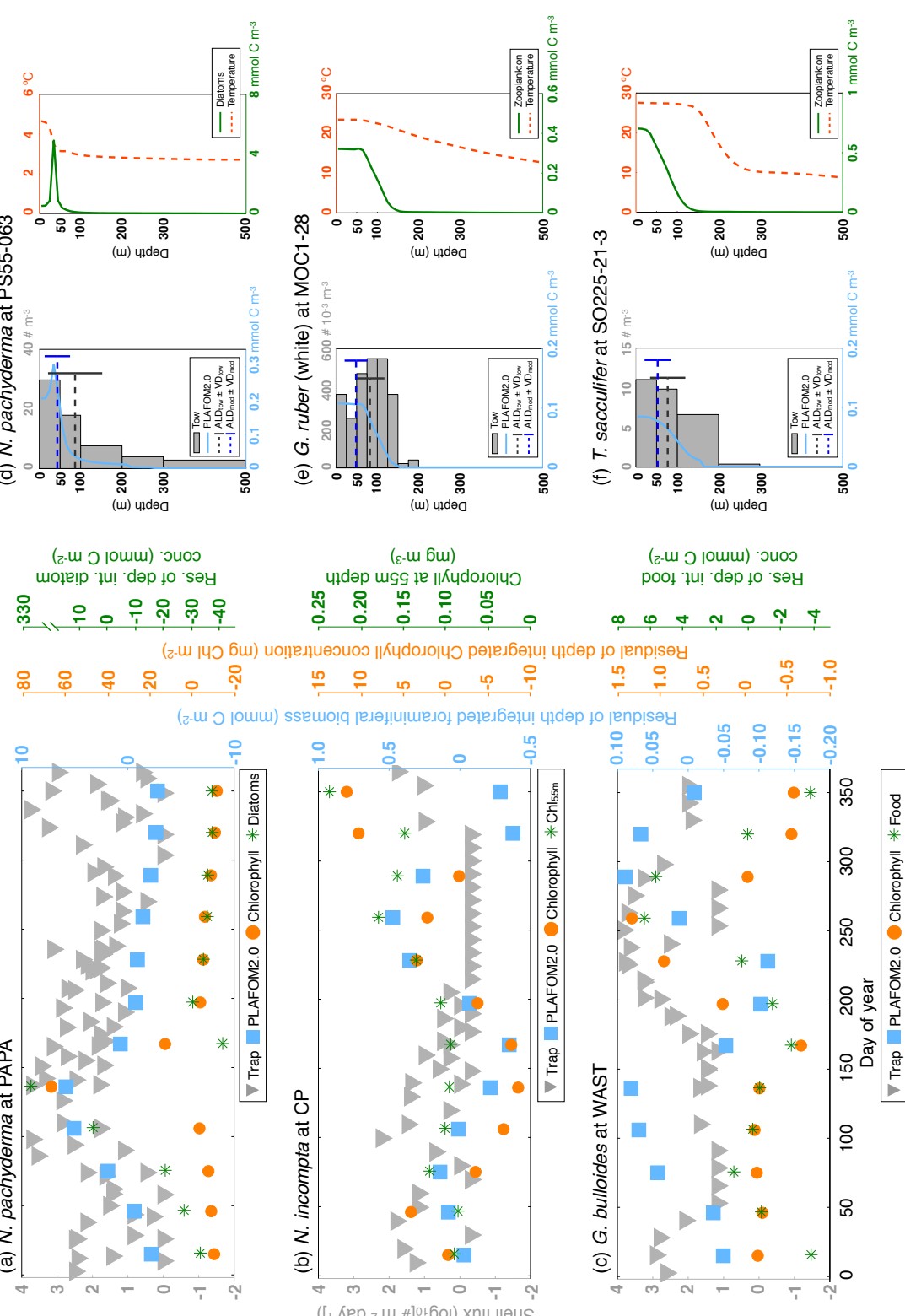

**Figure 7.** (a–c) Comparison of export planktonic foraminiferal shell fluxes in sediment traps (grey triangles) with the residuals (i.e., the deviation from the mean) of the depth integrated modeled foraminiferal carbon biomass (light blue squares). Note that the difference in the units between sediment trap data (in $\log_{10}[\# \, m^{-2} \, day^{-1}]$) and model output (in $mmol \, C \, m^{-2}$) does not affect the assessment of peak timing. The orange circles denote the residuals of the depth integrated modeled chlorophyll concentration (in $mg \, Chl \, m^{-2}$), and the dark green asterisks indicate in (a) the residuals of the depth integrated modeled diatom concentration (in $mmol \, C \, m^{-2}$), in (b) the modeled chlorophyll concentration (in $mg \, m^{-3}$) at 55 m water depth, and/or in (c) the residuals of the sum of the depth integrated modeled diatom and large detritus (i.e., main food) concentrations (in $mmol \, C \, m^{-2}$). (d–f) Comparison of the vertical distribution of live specimens in plankton tows (in $\# \, m^{-3}$; grey bars) with the modeled foraminiferal concentration over depth (in $mmol \, C \, m^{-3}$; light blue profiles). The dashed dark grey and blue lines indicate the average living depth (in m) and vertical dispersion calculated for the plankton tows ($ALD_{tow} \pm VD_{tow}$) and PLAFOM2.0 ($ALD_{mod} \pm VD_{mod}$), respectively. The dashed red lines denote the predicted temperature profiles (in °C), whereas the dark green lines correspond to the modeled vertical distribution of (d) diatoms (in $mmol \, C \, m^{-3}$) and/or (e–f) zooplankton (in $mmol \, C \, m^{-3}$). Data series of (a) *N. pachyderma* at site PAPA, (b) *N. incompta* at site CP, and (c) *G. bulloides* at site WAST. Depth profiles of (d) *N. pachyderma* at station PS55-063, (e) *G. ruber* (white) at station MOC1-28, and (f) *T. sacculifer* at station SO225-21-3. The locations of each sediment trap and plankton tow sample are given in Tables S1 and S2, respectively.

**Table 1.** Model parameter and their modifications relative to Fraile et al. (2008) and/or Kretschmer et al. (2016). The original value is given in parentheses.

| Species | N. pachyderma | N. incompta | G. bulloides | G. ruber (white) | T. sacculifer |
|---|---|---|---|---|---|
| $P_{F,0}$ | - (-) | - (-) | 2.6 (-) | 2.6 (-) | 2.6 (-) |
| $\alpha_{PI}$ | - (-) | - (-) | 0.012 (-) | 0.01 (-) | 0.07 (-) |
| $p_{\%}$ | - (-) | - (-) | 0.3 (-) | 0.3 (-) | 0.4 (-) |
| $T_{thres}$ | 18.0 (24.0) | 3.0 (-0.3) | 3.0 (-0.3) | 10.0 (5.0) | 15.0 (15.0) |
| $cl_{N.pachyderma,j}$ | - (-) | 0.2 (0.2) | 0 (0) | 0 (0) | 0 (0) |
| $cl_{N.incompta,j}$ | - (-) | - (-) | 0.1 (0.1) | 0.2 (0.8) | 0 (0) |
| $cl_{G.bulloides,j}$ | - (-) | 0.8 (0.5) | - (-) | 0.8 (0.8) | 0.8 (0.8) |
| $cl_{G.ruber(white),j}$ | - (-) | 0.2 (0.8) | 0.1 (0.5) | - (-) | 0.2 (0.2) |
| $cl_{T.sacculifer,j}$ | - (-) | 0 (0) | 0.1 (0.5) | 0.2 (0.2) | - (-) |

$P_{F,0}$ – maximum foraminiferal growth rate (in $\mathrm{day}^{-1}$) at $30\,^{\circ}$C (derived from the maximum zooplankton growth rate at $20\,^{\circ}$C given by Doney et al. (1996)).

$\alpha_{PI}$ – initial slope of the photosynthesis-light (PI) curve (in $\mathrm{m}^2\,\mathrm{W}^{-1}\mathrm{day}^{-1}$) (derived from PI-curve of *Synechococcus* given in Jodłowska and Śliwińska (2014) for *G. bulloides* and of endosymbiotic dinoflagellates given in Jørgensen et al. (1985) for *T. sacculifer*).

$p_{\%}$ – fraction of photosynthesis contributing to foraminiferal growth rate.

$T_{thres}$ – minimum (for *N. pachyderma*) or maximum (for all other species) threshold temperature at which foraminiferal species can thrive (in $^{\circ}$C).

$cl_{ij}$ – competition pressure of species $i$ upon species $j$.

**Table 2.** Relative sizes of the analyzed planktonic foraminifera species based on estimates of species size ranges from Schmidt et al. (2004) averaged over the sample locations in that study.

| Species | Size (in $\mu$m) |
|---|---|
| N. pachyderma | 321.50 |
| N. incompta | 321.50 |
| G. bulloides | 553.14 |
| G. ruber (white) | 541.00 |
| T. sacculifer | 661.44 |