# Peer review of "Modeling seasonal and vertical habitats of planktonic foraminifera on a global scale"

_Biogeosciences, 2017_

## Referee Comment (RC1) · I. van Dijk (Referee) · 11 Dec 2017

I have carefully read the manuscript 'Modeling seasonal and vertical habitats of planktonic foraminifera on a global scale' by Kretschmer and coauthors, which presents a model to predict global concentrations of five species of planktonic foraminifera and their depth habitat. This model could aid paleoclimatologists to correct for habitat depth when using shells of planktonic foraminifera to reconstruct ocean conditions. I need to remark that I have no experience using PLAFOM, or any practical experience with either the BEC model or CESM1.2(BGC) configuration. Therefore, my comments are rather general and an experienced user should review e.g. the use of model parameters and choice of configuration. I only have a couple of remarks that mainly focus on the usability and applicability of the model to reconstruct past depth habitats.

[Figure]

General comments

In general the authors should avoid certain 'model jargon', if they want to convince the broad foraminiferal society to use and apply this model. It is sometimes difficult to follow which steps are taken and assumptions were made to test or simulate certain scenarios (e.g. page 6, lines 23-25).

Even though habitat tracking is very important when using shells of planktonic foraminifera to reconstruct ocean conditions, it is still (more?) crucial to pinpoint the actual calcification depth within the depth habitat, since this is where the calcite is formed. Even though the model can reasonably well predict (globally) the vertical distribution, this does not mean that at this specific depth the environmental signal was 'logged' into the shell. Please include somewhere a couple of sentences on the reconstructed depth habitat compared to the actual calcification depth. Could this be the next step for PLAFOM3.0?

Section 2.3.1. What about other ocean parameters that vary over geological timescales which might influence growth rates? Like [PO3−4] (Aldridge et al., 2012, BG) on SNW or the effect of carbonate chemistry on calcification rates? For instance Lombard et al., 2010 found lower growth rates of several species with lowered [CO32-] conditions and Davis et al., 2017 (Sci. Rep.) observed lower calcification rates with decreasing pH. Why are these parameters not taken into account in the model? Are these effect minor compared to temperature and food availability?

Section 2.5.2. and 2.5.3. The authors use the sediment trap/plankton tow samples to test the accuracy of the model in predicting seasonality & depth habitats. However, the amount of data used for this comparison is not covering the total range of oceanic settings, since big parts of the ocean are underrepresented. Is it possible to extend this database by adding other published sediment trap data? This way you can show your model can predict depth habitat in a wider range of ocean conditions, which will make it more robust for application in deep time. Just some quick suggestions: Mediterranean

Sea: Mallo et al., 2017 BG; SW Atlantic: Venancio, et al., 2016 Marine Micropaleontology; Mozambique channel: Steinhardt et al., 2014 Marine Micropaleontology; Panama basin: Thunell et al., 1983 EPSL; Indian Ocean: Guptha et al., 1997 JFR.

Figure 2. Is it possible to add an 'offset map', in which you correlate e.g. the coretop data with the model data, to see where the model exactly over-/underestimates the data? This way you would be able to perform some (correlation) statistics, and this would clearly show the areas where the model didnot predict the correct distribution. I understand you are trying to capture the global signal (as stated several times in the manuscript), but paleooceanographers are more interested in specific areas when correction for e.g. depth habitat, and these are often also in more complicated oceanic settings (for example coastal/upwelling/river run off areas).

Page 11, line 27-31 and page 12, line 20-21. The authors state that part of the mismatch between the model and coretop data might stem from different genotypes having varying ecological preferences, and therefore their own unique model parameters. If so, does did not create a major bias for the whole model, especially when reconstructing depth habitats in deep time? For geological samples it is not possible to distinguish between genotypes, and therefore certain species might respond different in terms of depth habitat than the model will predict? Also, could it be that certain ecological preferences have changed over time? Can the authors predict how far in geological time you could still use this model to obtain reliable data on global distribution and depth habitat?

Minor comments

Page 2, line 18, 32; Page 6, line 16; page 11, line 23: Some problem with bracketing, e.g. double backeting etc.

Page 6, line 24: quasi-steady

Page 7, line 15: space missing between '(Figure 1a).' and 'We'

Page 8, line 5 and page 11, line 17: Arctic Circle

Page 12, line 10-14. Can you explain the underestimation of the model in scenarios were assemblages are dominated by two species?

Page 12, line 21: change or remove 'see'

―――――――――――――――

---

## Referee Comment (RC2) · Anonymous Referee #2 · 20 Dec 2017

The authors use existing sediment trap and plankton tow data to add seasonal and depth habitat information to the PLAFOM2.0 model. The authors then compare model results to modern data, concluding that they find a reasonable agreement between simulated and observed results for species-specific flux timing and depth habitat. The manuscript is well written, and the discussion of global trends in depth habitat is fantastic and alone an important contribution to the literature. Moreover, in light of an increasing understanding of the consequences of foraminifera habitat tracking for proxy data interpretation, the development of such a modeling tool is potentially quite useful.

The manuscript is successful in modeling modern depth preferences from unfortunately sparse observational data. While the model seems to reproduce broad trends (spinose species in near-surface waters) and earlier-when-warmer seasonality in some environments, figures 6-7 and the supplemental figures often show a strikingly poor fit between modeled and observed timing and depth preferences at specific sites. As the authors point out, the model tends to underestimate both amplitude of seasonal changes and potentially depth stratification. The authors should consider explicitly discussing why the model might be insensitive in replicating observed variability and how this would be likely to effect modeling of different climate inputs.

When the authors discuss relative abundance of species, are they referring to relative abundance with respect to just modeled species or all foraminifera? Is this consistent throughout? It might be worth clarifying this point

Why have the authors chosen not to include sediment trap based habitat depth based assessments?

p8/l23 (and throughout) – Do the authors really mean differences in biomass as opposed to species abundances? If so, is the biomass different in different species and how is this accounted for? And how does this metric compare to species abundances, as presumably used in the modern data to which the model is compared? p9/l18 (and throughout this section) – I'm not sure it makes sense for "maximum production" to be "year-round." Could you clarify? section 3.3 – might be helpful to define what you mean by "surface" and "subsurface" as these are pretty general terms but are being used as if the authors have a fairly specific depth range in mind p12/l30 –"prefer thriving" -> "thrive" p12/l33 – delete "largely" p14/l4 – delete "among each other" p14/l11 - delete "preferably" p14/l31 – "cold to transitional" compares a temperature to a zonation p15/l22 – a -> the p17/l2 – might be better to describe these as short time series as compared to plankton tows which really are "snapshots" p17/l18 – or genotypes or phenotypes? p17/l26 "a few"?

Figure 6 is extremely difficult to read given the mix of opacity and multiple symbols and colors. Is there a better way to present this data?

Figures 6 and 7 (a-c) suggest a quite poor fit of modeled data to sediment trap observations. i.e. 7c shows the model completing missing the flux timing of bulloides in JGOFS34. The authors include an overview or why there might be some data-model mismatch, but I think a wider discussion of why and how this could impact or limit interpretation of model results is warranted

———————————————————

---

## Referee Comment (RC3) · J. Bijma (Referee) · 22 Dec 2017

Scientific significance: Excellent

The manuscript by Kretschmer et al. represents a substantial contribution to scientific progress within the scope of Biogeosciences. It is the latest one in a series of "foram-flux modelling" papers from the Bremen group. In 2006, Zaric et al. developed the first empirical model that described globally the fluxes of planktonic foraminifera at species level in dependence of sea-surface temperature, mixed-layer depth and export production. Over the years, the foram model itself, its parameterization, and its implementation and coupling to other models has evolved (e.g. Fraile et al., 2008; 2009; Kretschmer et al., 2016). The aim of all of these papers has always been to project the

effect of changing environmental conditions on species distributional patterns in time and space. The current paper adds a vertical dimension to the existing foram model by applying the previously used spatial parameterization of biomass as a function of temperature, light, nutrition, and competition on depth-resolved parameter fields.

Scientific quality: good

The scientific approach and methods are valid. The results are discussed appropriately but the discussion lacks a critical analysis of the model-data comparison beyond the caveats mentioned in section 4.2 "Comparison with local observations".

The authors write on p. 17 line 22-23: "This vertical migration of planktonic foraminifera during their ontogeny cannot be reproduced by PLAFOM2.0 as the model parameterizations do not include the individual species' life cycles.". It is quite understandable that implementing true reproduction cycles of cohorts of foraminifera, including "real" population dynamics and ontogenetic migration is beyond the present manuscript. Hence, the model does not calculate absolute or relative numbers of a certain species within a certain ontogenetic size class based on reproductive success and size specific growth- and mortality-rates, but rather calculates changes in species specific carbon concentration (in mmol C m$-3$), which can be converted to numbers afterwards.

There is nothing wrong with this approach but it means that the parameterization of PLAFOM2.0 is based on practical "sum" or "composite" parameters. These are then used to tune the model outcome to the overall data. For instance, growth of all species is approximated using a modified form of Michaelis-Menton kinetics in dependence of species specific food availability and temperature sensitivity (Fraile et al., 2008). To account for the light dependence with depth, influencing the growth of only symbiont bearing foraminifera, the authors included a "photosynthetic growth rate". They use "......a similar approach as Doney et al. (1996) and Geider et al. (1998), who determined phytoplankton growth rates by available light and nutrients….. (p.5 line 15-17)". Such a parameterization is normally used for phytoplankton, that has orders of magnitude higher densities and cell division rates that respond very fast (within a day) and directly to light and nutrients. The symbiont bearing forams in this manuscript obey a (semi) lunar reproduction cycle and occur in densities that are very much lower, such that a "phytoplankton" kind of response cannot be expected. The authors use it as an additional tuning parameter for symbiont bearing forams next to food preference and temperature to develop species specific depth (light/nutrient) habitat preferences. Although it is a valid approach, the authors should clearly state that it is artificial.

Growth is balanced by mortality, which is not a formulation for "real" mortality but another tuning parameter: "we adjusted parts of the mortality rate equation to improve the model accuracy (p. 5 line 8-9).".

Overall, there are many factors that allow tuning, e.g. "p% represents the fraction of photosynthesis contributing to growth (p.5 line 31)". Interestingly, the authors have a higher p% for T. sacculifer (0.4) than for G. ruber (0.3), where I would have done it the other way around (see my comments on these species further below).

Another tuning factor is the temperature dependence of the predation term: ".......we followed Moore et al. (2004) and adjusted the temperature dependence of the predation term (MLpred in mmolCm$-3$s$-1$) (p.6 line 3-4). Also "....we included a stronger competitive behavior of G. bulloides by adjusting the free parameters in the competition term. (p.6 line 10-11). Having collected planktonic foraminifera by SCUBA diving for many, many years and looking at average typical blue water densities of ca. 10 specimens per m3 per species, and 3 dominant species in an assemblage, it is hard to believe that they compete with each other for resources as each of them occupies a space of only a few mm3 and they are stationary in the water column.

Certain boundary conditions also correct model misfits, e.g. "...zero fluxes have been replaced by half of the observed minimum flux. (p.7 line 25-26)".

All of these parameters were introduced to allow a good fit between model output and data but maybe not for the right reason. As such, we do not know how realistic this

parameterization represents real planktonic foraminiferal population dynamics which is more complex (including lunar based reproduction cycles, ontogenetic migration, etc.).

Winter mixing, thermocline shoaling and annual irradiation changes are probably important parameters controlling foram population dynamics just as certain density layers may be important for gamete fusion in real foram life. I'm not sure how well these features are implemented in the models.

The bottom line is that, even though I appreciate the model and the manuscript a lot, I would like to see a discussion on these issues and if possible a statistical verification of the model performance. The description of the results and the discussion on modeled geographical ranges, seasonal and vertical distribution, as well as on the modeled seasonal variability of depth habitat, lacks a statistical treatment of the data. How good is the model performance and how sensitive is it to each of the model parameters?

I would appreciate a more quantitative treatment of the model performance instead of statements like "The predicted global distribution patterns of the five considered planktonic foraminiferal species are in good agreement with the core-top data (Figure 2) (p. 11 line 14-15)?

The discussion on the global distribution patterns is mostly related to temperature. What about the other parameters: food, nutrients, productivity, light, etc.? How does it compare to the "Longhurst Biogeographical Provinces". He partitioned the world oceans into provinces ("Ecological Geography of the Sea") based on the prevailing physical factors as a regulator of phytoplankton distribution, including temperature, photic depth, mixed layer depth etc. (e.g. Longhurst 1995; 1998).

Having "fixed" model parameters simulates so called "habitat tracking" of the forams through the seasons (but also on timescales of climate change or on glacial/interglacial cycles). This is a very important aspect to verify and would call for a section/paragraph by itself (see also Rebotim et al., 2017). For instance, on p15 line 23-25 you write "Rebotim et al. (2017) identified an annual cycle in the habitat of T. sacculifer and N.

[Figure]

incompta in the subtropical eastern North Atlantic. Both species appear to descend in the water column from winter to spring and reach their deepest habitat in spring to summer before ascending again to a shallower depth towards winter (Rebotim et al., 2017).". How does this fit the "habitat tracking" picture? The authors could probably use observations on G. ruber and T. sacculifer for that as well. I may be wrong but I always thought that G. ruber lives closer to the surface than T. sacculifer (see also table 3 in Rebotim et al., 2017)? From laboratory experiments I know that T. sacculifer can handle living prey such as copepods much better than G. ruber while the latter seems to rely more on symbiont carbon, i.e. shows a more "autotrophic" lifestyle. Is it possible to see this in the data based on a more rigorous model-data comparison?

The results of the point-by-point comparative analysis for each site and species as provided in the Supplement (Figures S3 and S4) are very helpful but also show that the model is far from perfect and sometimes there is a complete mismatch. I would have appreciated a sensitivity study to determine the hierarchy of factors for the different species controlling the shell export fluxes regional and seasonal (including e.g. bimodal patterns) as well as the vertical distribution (including ALD). This would probably be a paper by itself but in my view a very important one.

Presentation quality: good/fair

Although the scientific results and conclusions are presented in a relatively clear and well-structured way it is not easy to grasp why the model underestimates e.g peak amplitude. What would happen if growth in the equation is increased or mortality is decreased? I sometimes wondered why the authors didn't play more with the model or used statistical techniques to quantify data-model mismatch (this is the reason for the "fair" mark). The number and quality of figures/tables is good and the supplementary material is very appropriate. The English language is very good.

Minor corrections:

On page 2 line 18-20: ".............the lunar cycle and/or the structure of the water column), which influence the species-specific depth habitats (including their mean living depth and vertical migration) (e.g., Fairbanks and Wiebe, 1980; Fairbanks et al., 1982; Schiebel et al., 2001; Simstich et al., 2003; Field, 2004; Salmon et al., 2015; Rebotim et al., 2017), the only attempt to model the vertical habitat is by Lombard et al. (2011).", and on page 17 line 20-23: "Several studies from different areas also showed that the main habitat depth of some species increases from the surface to deeper water layers during shell growth (Peeters and Brummer, 2002; Field, 2004; Iwasaki et al., 2017). Although I appreciate all the references that you list for ontogenetic migration and lunar cycle, there are only a few papers that specifically deal with very detailed population dynamics, lunar cyclicity and ontogenetic migration of planktonic forams that could/should be mentioned here (it was one of the first topics I studied when starting to work on planktonic foraminifera): Bijma et al., 1990; Bijma, 1991; Bijma and Hemleben, 1994; Bijma et al., 1994; Hemleben and Bijma, 1994; Schiebel et al., 1997. In my opinion, these references would fit best on p. 19 line 32-34: "………and by explicitly parameterizing the ontogeny of each individual planktonic foraminifera, thus, by considering the changes in the species' life cycles with depth, could considerably improve the model.".

P. 9 line 27-30: "Although seasonal changes in the modeled foraminiferal peak fluxes with temperature are evident, all five species exhibit an almost constant peak amplitude (i.e., the maximum concentration divided by the annual mean) in their preferred habitat, which is, i.a., limited by temperature. Outside their preferred living conditions the peak amplitudes increase for most of the species considerably (Figure 3).". It has not become clear to me what it means when "peak amplitude" is large or small in terms of real population dynamics ("bloom"?) and what it means in terms of model performance?

P. 14 line 26-28: "This would explain why the highest modeled concentrations of T. sacculifer occur at shallower depths compared to G. ruber (white) (see Figures 4d-e and 5d-e).". Striktly speaking this doesn't explain it because this is what you put into

the model in the first place (see my comments above)

P. 16 line 18: "G. bulloides, however, is found year-round close to the surface along the.. ...". Write the genus name full at the beginning of a sentence.

References:

Bijma, J., Erez, J. and Hemleben, C. (1990) Lunar and semi-lunar reproductive cycles in some spinose planktonic foraminifers. Journal of Foraminiferal Research 20, 117-127.

Bijma, J. (1991) Lunar pulses of carbonate output by spinose planktonic Foraminifera, in: Reid, P.C., Turley, C.M., Burkill, P.H. (Eds.), Protozoa and Their Role in Marine Processes. NATO ASI Series G: Ecological Sciences. Elsevier, Plymouth, pp. 353-354. Bijma, J. and Hemleben, C. (1994) Population dynamics of the planktic foraminifer Globigerinoides sacculifer (Brady) from the central red sea. Deep-sea research part I: oceanographic research papers 41, 485-510.

Bijma, J., Hemleben, C. and Wellnitz, K. (1994) Lunar-influenced carbonate flux of the planktic foraminifer Globigerinoides sacculifer (Brady) from the central red sea. Deep-sea research part I: oceanographic research papers 41, 511-530.

Hemleben, C. and Bijma, J. (1994) Foraminiferal population dynamics and stable carbon isotopes., in: Zahn, R., Pedersen, T.F., Kaminski, M., Labeyrie, L. (Eds.), Carbon Cycling in the Glacial Ocean: Constraints on the Ocean's Role in Global Change. Elsevier, Fellhorst, pp. 145-166.

Longhurst, A. (1995) Seasonal cycles of pelagic production and consumption. Progress in Oceanography 36, 77-167.

Longhurst, A. (1998) Ecological Geography of the Sea ACADEMIC PRESS

Schiebel, R., Bijma, J. and Hemleben, C. (1997) Population dynamics of the planktic foraminifer Globigerina bulloides from the North Atlantic. Deep Sea Research 44,

1701-1713.

---

## Referee Comment (RC4) · Anonymous Referee #4 · 3 Jan 2018

Review for Authors Modeling seasonal and vertical habitats of planktonic foraminifera on a global scale Kretschmer et al.

This paper builds upon preexisting work modeling planktonic foram distributions in the global oceans via a coupling to CESM's ocean model. The goal is to better understand how the vertical distribution of foramnifera species varies seasonally and throughout larger climatic changes in the ocean. The paper is generally well written, clear, and broadly does a fine job demonstrating the usefulness of the model. It is also very thorough in its examination of the model's performance against available data. The methods seem robust and I can recommend that with some minor revisions (mostly grammar and clarity) the paper be published in Biogeosciences.

I must acknowledge that I am not an expert on the biogeochemistry of planktonic

forams in any way and hope the other reviewers can address the methods and paramterizaitons employed in this paper in particular. I can instead comment on the benefit of this work and the need for such proxy system models for the robust interpretation of paleoceanographic records via the use of PLAFOM2.0 + CESM1.2. To that end, my first major comment is that the authors can focus more in the introduction and conclusion on the body of literature developing forward models, or proxy system models, for understanding paleoclimate proxies and introduce this work as a part of this group of literature. A major effort has been underway to build proxy system models, link them with GCMs, and make these models publicly available, and this paper is absolutely in this category and should make as much clear.

See for example:

>Evans, Michael N., et al. "Applications of proxy system modeling in high resolution paleoclimatology."ÂăQuaternary Science ReviewsÂă76 (2013): 16-28.

>Dee, S., et al. "PRYSM: An open‐source framework for PRoxY System Modeling, with applications to oxygen‐isotope systems."ÂăJournal of Advances in Modeling Earth SystemsÂă7.3 (2015): 1220-1247.

>Schmidt, Gavin A. "Forward modeling of carbonate proxy data from planktonic foraminifera using oxygen isotope tracers in a global ocean model."ÂăPaleoceanographyÂă14.4 (1999): 482-497.

You might also consider mentioning (in the intro or discussion) the potential for PLAFOM to assist in data assimilation exercises for periods extending back further than the last millennium, for example. A number of papers look at the impacts of using process-based models in the DA framework and this is another application of your model. See for example:

>(e.g. Goosse, Hugues, et al. "Reconstructing surface temperature changes over the past 600 years using climate model simulations with data assimilation."ÂăJournal

of Geophysical Research: AtmospheresÂă115.D9 (2010)), as well as: >Steiger, Nathan J., et al. "Assimilation of time-averaged pseudoproxies for climate reconstruction."ÂăJournal of Climate27.1 (2014): 426-441. >Dee, Sylvia G., et al. "On the utility of proxy system models for estimating climate states over the common era."ÂăJournal of Advances in Modeling Earth SystemsÂă8.3 (2016): >Hakim, Gregory J., et al. "The last millennium climate reanalysis project: Framework and first results."ÂăJournal of Geophysical Research: AtmospheresÂă121.12 (2016): 6745-6764

In Section 4, it would be nice if the authors could provide a more quantitative data-model comparison technique—you identify areas where the model does not well simulate the observations and Figure 2 summarizes this to some extent, but perhaps you could include an additional table or figure or even compute something like the RMSE for each oceanic province? Or the mean RMSE for each species over all of the locations where core-top data exist?

Finally, in the discussion, you assert (correctly) that your new model is a powerful tool for separating the independent influences of habitat and climate on foram reconstructions. I think this paper would be greatly strengthened by a demonstration of this. Can you take a well-known and vetted reconstruction and apply this model in a meaningful way to reassess the climatic interpretation? I think this would show the power of forward modeling in this field to make more robust assessments of uncertainties in oceanic climate changes. . . And I think having this demonstration would add weight to the assertions you make in your Discussion section.

Minor / Line by Line comments: (Page-Line)

2-10 awkward paragraph break, consider revising 2-13 comma after perspective, 2-20 Have you investigated/reviewed Schmidt et al., 1998, 1999? These papers I believe address vertical migration of foram species in the water column—worth checking/citing if appropriate.

@article{Schmidt1998, Author = {Schmidt, Gavin A}, Title = {{Oxygen-18 variations in
a global ocean model Relationships between $\delta^{18}O$ and S}}, Volume = {25}, Year = {1998}}

@article{Schmidt1999, Author = {Schmidt, Gavin a.}, Journal = {Paleoceanography}, Number = {4}, Pages = {482}, Title = {{Forward modeling of carbonate proxy data from planktonic foraminifera using oxygen isotope tracers in a global ocean model}}, Url = {http://www.agu.org/pubs/crossref/1999/1999PA900025.shtml}, Volume = {14}, Year = {1999},

2-26 need comma after behavior. Check for needed commas and small grammatical errors throughout text.

3-6 comma after estimate, 3-13 this phrase is awkward, revise ("with the biogeochemical model being enabled") 3-15 change "aimed for" to 'aimed to' 3-16 change "at geologic timescales" to "ON geologic timescales" Check for similar awkward language throughout. 3-23 comma after configuration, 3-30 no paragraph break. 4-9 what do you mean by 'data models' for the atmosphere, etc.? Are you not using the fully coupled simulations and using some kind of statistical representation of the other components? Heading 2.4 consider changing this to "Coupled GCM Setup" ? 7-15 missing space before new sentence. 8-21 comma after 'life cycle,' Throughout section 3, be extremely clear about whether you are referring to observations vs. the model simulation of foram distributions/abundances etc. The reader gets a bit lost in the data-model comparison here unless that's super clear. 16-29 no comma after 'data' 16-30 this is a run-on sentence—consider shortening/rewriting I appreciate the thorough discussion of the model – data comparison limitations on page 17.

Figure 5 has some strange cropping issues along top margin.

Please also note the supplement to this comment:
https://www.biogeosciences-discuss.net/bg-2017-429/bg-2017-429-RC4-supplement.pdf

---

## Author Comment (AC1) · 29 Mar 2018

**Response to Referee #1:**

**Ref.:  Ms. No. bg-2017-429**

**Title: Modeling seasonal and vertical habitats of planktonic foraminifera on a global scale**

We would like to thank reviewer Inge van Dijk for her constructive comments and suggestions, which will help us to greatly improve our manuscript. Based on the comments of all four reviewers we will prepare a new version of our manuscript as outlined below.

However, during the review process, we discovered an error in the underlying ocean model. Unfortunately, the ocean circulation is not correctly represented in the used coarse resolution (i.e., ~3°) model configuration. For a correct representation of the ocean and to yield scientifically consistent results, we had to perform a new model run with a higher horizontal resolution (i.e., ~1°) on a supercomputing system. This model run takes ca. 5 weeks and is currently in the final production phase. At a first glance, the new results will not differ that much from our previous results as the representation of the upper ocean, where the analyzed foraminiferal species live, was actually reasonably well simulated in the coarse resolution model configuration compared to, e.g., the World Ocean Atlas 2013. We expect that the distribution of only a few species might be affected, when using the higher resolution model configuration with a more realistic representation of the ocean physics. Since we have not yet obtained the final results, we were not always able to provide detailed answers to your comments and had to keep our responses rather general.

Please find, in the following, the original comments in black and our responses in light blue; the indicated page and line numbers refer to the previously submitted manuscript.

Referee #1 comments:

I have carefully read the manuscript 'Modeling seasonal and vertical habitats of planktonic foraminifera on a global scale' by Kretschmer and coauthors, which presents a model to predict global concentrations of five species of planktonic foraminifera and their depth habitat. This model could aid paleoclimatologists to correct for habitat depth when using shells of planktonic foraminifera to reconstruct ocean conditions. I need to remark that I have no experience using PLAFOM, or any practical experience with either the BEC model or CESM1.2(BGC) configuration. Therefore, my comments are rather general and an experienced user should review e.g. the use of model parameters and choice of configuration. I only have a couple of remarks that mainly focus on the usability and applicability of the model to reconstruct past depth habitats.

General comments

In general the authors should avoid certain 'model jargon', if they want to convince the broad

foraminiferal society to use and apply this model. It is sometimes difficult to follow which steps are taken and assumptions were made to test or simulate certain scenarios (e.g. page 6, lines 23-25).

Thank you for pointing this out. We will change parts of the method section, also according to the higher resolution model configuration, and will include more or delete redundant information, when appropriate, for a better understanding. However, to ensure reproducibility of our study, we cannot avoid using a certain 'model jargon' to explain the applied modeling approach and the used model setup. We already tried to use as little model jargon as possible and provided in all conscience a comprehensible model description.

Even though habitat tracking is very important when using shells of planktonic foraminifera to reconstruct ocean conditions, it is still (more?) crucial to pinpoint the actual calcification depth within the depth habitat, since this is where the calcite is formed. Even though the model can reasonably well predict (globally) the vertical distribution, this does not mean that at this specific depth the environmental signal was 'logged' into the shell. Please include somewhere a couple of sentences on the reconstructed depth habitat compared to the actual calcification depth. Could this be the next step for PLAFOM3.0?

This is a valid point and in a next step, we would like to combine PLAFOM2.0 with a module, which specifically takes this into account and calculates species-specific isotope compositions of the modeled foraminiferal species, such that we could directly infer information about the calcification depth of each species. However, without any information about the species-specific habitats, it is difficult to provide a statement regarding the calcification depths of the individual foraminiferal species. Therefore, we at first intended to simulate realistic species-specific habitat depths and next we plan on obtaining realistic calcification depths. We will include a paragraph in the discussion (section 4.1.3) regarding a comparison of the reconstructed depth habitat with the actual calcification depth of the individual species after evaluating the new results.

Section 2.3.1. What about other ocean parameters that vary over geological timescales which might influence growth rates? Like $[PO_4^{3-}]$ (Aldridge et al., 2012, BG) on SNW or the effect of carbonate chemistry on calcification rates? For instance Lombard et al., 2010 found lower growth rates of several species with lowered $[CO_3^{2-}]$ conditions and Davis et al., 2017 (Sci. Rep.) observed lower calcification rates with decreasing pH. Why are these parameters not taken into account in the model? Are these effect minor compared to temperature and food availability?

This is a valid point again, but we are not attempting to model species-specific growth rates (as opposed to Lombard et al., 2011). Rather we aim to more directly estimate foraminifera abundance, which can be compared to the sediment record more directly. The relationship

between growth rate and abundance is far from straightforward (cf. Lombard et al., 2011) and we are not aware of studies that have investigated the effect of those parameters on the abundance of planktonic foraminifera. We are aware that other ocean parameters might influence species-specific growth rates. The aim of this study, however, was to test if the existing planktonic foraminifera model is able to reproduce species-specific habitats when combined with a model configuration that resolves the vertical. One has to bear in mind that a model is only a simplification of reality and including more parameters would likely introduce more degrees of freedom and could lead to more model uncertainty and could additionally increase the computational costs. However, for a future model development it is worth considering those parameters. Here it is beyond the scope of this study to include more parameters to determine growth rates.

Section 2.5.2. and 2.5.3. The authors use the sediment trap/plankton tow samples to test the accuracy of the model in predicting seasonality & depth habitats. However, the amount of data used for this comparison is not covering the total range of oceanic settings, since big parts of the ocean are underrepresented. Is it possible to extend this database by adding other published sediment trap data? This way you can show your model can predict depth habitat in a wider range of ocean conditions, which will make it more robust for application in deep time. Just some quick suggestions: Mediterranean Sea: Mallo et al., 2017 BG; SW Atlantic: Venancio et al., 2016 Marine Micropaleontology; Mozambique channel: Steinhardt et al., 2014 Marine Micropaleontology; Panama basin: Thunell et al., 1983 EPSL; Indian Ocean: Guptha et al., 1997 JFR.

The reviewer rightly points out that our data compilation is not comprehensive. However, we pursued the strategy to acquire sediment trap and plankton tow data at more or less the same region to guarantee a consistent model-data-comparison throughout the manuscript when analyzing species-specific seasonal and vertical habitat patterns (see Figure 1b). We agree that this prerequisite limits the number of studies that can be used to evaluate the model, but the underlying data base covers all provinces and provides good estimates of the different species-specific habitats and their variability on a global scale that is sufficient to show the strength and weaknesses of our model.

Figure 2. Is it possible to add an 'offset map', in which you correlate e.g. the coretop data with the model data, to see where the model exactly over-/underestimates the data? This way you would be able to perform some (correlation) statistics, and this would clearly show the areas where the model did not predict the correct distribution. I understand you are trying to capture the global signal (as stated several times in the manuscript), but paleooceanographers are more interested in

specific areas when correction for e.g. depth habitat, and these are often also in more complicated oceanic settings (for example coastal/upwelling/river run off areas).

We will include an additional map in Figure 2 that provides a more thorough comparison between modeled and observed assemblages. Therefore, we will calculate the Bray-Curtis index of similarity between the model data and the core-top data, such that we can provide a measure of confidence. Note for the calculation, we will account for the different sizes of each species by using a mean size for each species based on the results of Schmidt et al. (2004) and recalculate the modeled relative abundances accordingly. We will add this analysis to the manuscript (i.e., to sections 3.1 and 4.1.1) to provide a thorough model-data-comparison. Nevertheless, the used model configuration consisting of three different models (i.e., POP2, BEC, PLAFOM2.0) could hamper a thorough statistical analysis as it is not unequivocally possible to differentiate which component might actually lead to a possible over-/underestimation of the data. Even the now used higher model resolution could likely lead to misrepresentations of small-scale processes, oceanic fronts, river runoff areas, and coastal upwelling regions, and could, thus, account for the model-data-mismatch. In addition, it is not possible to correlate the core-top data with the model data directly, because PLAFOM2.0 calculates foraminiferal concentrations via carbon biomass (i.e., in mmol $C/m^3$) and the core-top samples provide foraminiferal concentrations via number of specimens.

Page 11, line 27-31 and page 12, line 20-21. The authors state that part of the mismatch between the model and coretop data might stem from different genotypes having varying ecological preferences, and therefore their own unique model parameters. If so, does did not create a major bias for the whole model, especially when reconstructing depth habitats in deep time? For geological samples it is not possible to distinguish between genotypes, and therefore certain species might respond different in terms of depth habitat than the model will predict? Also, could it be that certain ecological preferences have changed over time? Can the authors predict how far in geological time you could still use this model to obtain reliable data on global distribution and depth habitat?

The reviewer points out two important considerations: i) cryptic species with different ecological preferences and ii) the question of stationarity. We would argue that both hold for all attempts to use planktonic foraminifera to reconstruct the past ocean. The assumption of stationarity of any proxy is fundamental to all paleoclimate reconstructions. The model can of course only be used for the time that the species have been present and for as long as we have indications that their ecology remained constant (cf. Huber et al., 2000 for *N. pachyderma*). The primary intended use of the model is to apply it to climate conditions covering the Last Glacial Maximum and/or the last couple of glacial-interglacial cycles, but not to deep time, when different species existed or extant

species may have had different ecological preferences.

With respect to cryptic species the reviewer is right to point out that this forms an important caveat. However, as the reviewer also mentions, it is often impossible to distinguish between cryptic species in the fossil record, so this caveat applies to any reconstruction using planktonic foraminifera. This is exactly the reason why ecological preferences of cryptic species need to be resolved, so that reconstructions and modeling efforts can be improved. To clarify this point, we will add this issue to the end of section 4.1.1:

*"[...] In addition, the discrepancies between the model and core-top data might also partly stem from the underlying model parameterizations applied on a global scale, which do not distinguish between distinct genotypes of the different species with potentially varying ecological preferences. However, the recognition of cryptic species remains challenging, if not impossible, and these species are therefore rarely separated in sediment samples. This complicates a direct comparison between geological samples and model data, even if the ecological preferences were perfectly constrained. However, the model skill/performance suggests that ecological differences between cryptic species are limited and that the model provides a useful first-order approximation of global species distribution."*

Minor comments

Page 2, line 18, 32; Page 6, line 16; page 11, line 23: Some problem with bracketing, e.g. double bracketing etc.

We checked for the double bracketing and, where possible, we will delete the unnecessary brackets. However, for some cases (i.e., Page 6, line 16; page 11, line 23) we will not change the bracketing as this would potentially cause a misunderstanding with the referencing.

Page 6, line 24: quasi-steady

Done.

Page 7, line 15: space missing between '(Figure 1a).' and 'We'

Done.

Page 8, line 5 and page 11, line 17: Arctic Circle

Done.

Page 12, line 10-14. Can you explain the underestimation of the model in scenarios were assemblages are dominated by two species?

Here, we actually meant that the model is not able to capture the full extent of the observed relative abundances in certain areas where a dominance of some species is actually expected. We will change this sentence accordingly after evaluating the new results.

Page 12, line 21: change or remove 'see'
Done.

References:

- Huber, R., H. Meggers, K.-H. Baumann, M. E. Raymo, and R. Henrich (2000), Shell size variation of the planktonic foraminifer *Neogloboquadrina pachyderma* sin. in the Norwegian-Greenland Sea during the last 1.3 Myrs: implications for paleoceanographic reconstructions, *Palaeogeography, Palaeoclimatology, Palaeoecology*, *160*, 193-212.
- Lombard, F., L. Labeyrie, E. Michel, L. Bopp, E. Cortijo, S. Retailleau, H. Howa, and F. Jorissen (2011), Modeling planktic foraminifer growth and distribution using an ecophysiological multi-species approach, *Biogeosciences*, *8*, 853-873.
- Schmidt, D. N., S. Renaud, J. Bollmann, R. Schiebel, and H. R. Thierstein (2004), Size distribution of Holocene planktic foraminifer assemblages: biogeography, ecology and adaptation, *Marine Micropaleontology*, *50*, 319-338.
- Thunell, R. C., W. B. Curry, and S. Honjo (1983), Seasonal variation in the flux of planktonic foraminifera: time series sediment trap results from the Panama Basin, *EPSL*, *64*, 44-55.

---

## Author Comment (AC2) · 29 Mar 2018

**Response to Referee #2:**

**Ref.: Ms. No. bg-2017-429**

**Title: Modeling seasonal and vertical habitats of planktonic foraminifera on a global scale**

We would like to thank the reviewer for the constructive comments and suggestions, which will help us to greatly improve our manuscript. Based on the comments of all four reviewers we will prepare a new version of our manuscript as outlined below.

However, during the review process, we discovered an error in the underlying ocean model. Unfortunately, the ocean circulation is not correctly represented in the used coarse resolution (i.e., ~3º) model configuration. For a correct representation of the ocean and to yield scientifically consistent results, we had to perform a new model run with a higher horizontal resolution (i.e., ~1º) on a supercomputing system. This model run takes ca. 5 weeks and is currently in the final production phase. At a first glance, the new results will not differ that much from our previous results as the representation of the upper ocean, where the analyzed foraminiferal species live, was actually reasonably well simulated in the coarse resolution model configuration compared to, e.g., the World Ocean Atlas 2013. We expect that the distribution of only a few species might be affected, when using the higher resolution model configuration with a more realistic representation of the ocean physics. Since we have not yet obtained the final results, we were not always able to provide detailed answers to your comments and had to keep our responses rather general.

Please find, in the following, the original comments in black and our responses in light blue; the indicated page and line numbers refer to the previously submitted manuscript.

Referee #2 comments:

The authors use existing sediment trap and plankton tow data to add seasonal and depth habitat information to the PLAFOM2.0 model. The authors then compare model results to modern data, concluding that they find a reasonable agreement between simulated and observed results for species-specific flux timing and depth habitat. The manuscript is well written, and the discussion of global trends in depth habitat is fantastic and alone an important contribution to the literature. Moreover, in light of an increasing understanding of the consequences of foraminifera habitat tracking for proxy data interpretation, the development of such a modeling tool is potentially quite useful.

The manuscript is successful in modeling modern depth preferences from unfortunately sparse observational data. While the model seems to reproduce broad trends (spinose species in nearsurface waters) and earlier-when-warmer seasonality in some environments, figures 6-7 and the supplemental figures often show a strikingly poor fit between modeled and observed timing and depth preferences at specific sites. As the authors point out, the model tends to underestimate both amplitude of seasonal changes and potentially depth stratification. The authors should consider explicitly discussing why the model might be insensitive in replicating observed variability and how this would be likely to effect modeling of different climate inputs.

This is a good point and we will extend the discussion in this regard especially by bearing in mind that the coarse 3° ocean model is not fully able to represent the ocean's physics properly. Apart from the uncertainty in the observational data (see section 4.2), it is due to the model complexity not trivial to determine which model component (i.e., POP2, BEC or PLAFOM2.0) contributes to what extent to the model-data-mismatch. Determining this would require a suite of sensitivity experiments with each model component. Whilst we agree that these would be useful – and will consider this for future work – we think that the model as it is already presents a useful contribution to improve the interpretation of foraminifera-based proxy records.

Nevertheless, we will expand the discussion on the model uncertainty in section 4.2 after evaluation of the new model run. We will specifically address the dependence of the results on the individual model components. The inferred importance of temperature and food availability (provided by POP2 and BEC, respectively) on the distribution of foraminifera implies that each model component is important for an accurate representation of foraminifera distribution. Hence, we expect the higher resolution ocean model to provide a more realistic representation of ocean physics, which will cascade through the model hierarchy leading to an improved overall model skill. Nevertheless, sub-grid processes and known POP2 and BEC model issues (see, e.g., Danabasoglu et al., 2012, 2014; Moore et al., 2013) will remain. These will contribute to the model-data mismatch, but will not provide information/constraints on the planktonic foraminifera model per se.

When the authors discuss relative abundance of species, are they referring to relative abundance with respect to just modeled species or all foraminifera? Is this consistent throughout? It might be worth clarifying this point.

When we are discussing species relative abundances for the core-top data, we always refer to relative abundances with respect to only the five modeled species. We mention this in section 2.5.1 (page 7, line 17) and also in the caption of Figure 2. To avoid confusion, we will repeat this point also in section 4.1.1 in the beginning of the first paragraph:

*"Note that the relative abundances for the core-top data have been calculated with respect to just the*

*five modeled species and not to the whole faunal assemblage."*

Why have the authors chosen not to include sediment trap based habitat depth based assessments?
Since sediment traps provide export flux rates, which are not modeled here, and thus do not provide information about depth habitat, a sediment trap based depth habitat assessment is simply not possible. However, there exist calcification depth estimates based on chemical properties of foraminifera from sediment traps, but calcification depth is not identical to habitat depth. Therefore, we only use plankton tow data for a meaningful depth habitat assessment.

p8/l23 (and throughout) – Do the authors really mean differences in biomass as opposed to species abundances? If so, is the biomass different in different species and how is this accounted for? And how does this metric compare to species abundances, as presumably used in the modern data to which the model is compared?
PLAFOM2.0 calculates the foraminiferal abundance of each species via carbon biomass to be consistent with the ecosystem model (see section 2.3 in the manuscript and Fraile et al., 2008). In the manuscript we prefer to use this unit, rather than foraminifera abundance, since conversion to abundance requires, as the reviewer rightly points out, another step.
However, this conversion of biomass to abundance is only of importance for the comparison of the modeled and observed assemblages. For the global comparison with the core-top data, we are not interested in assessing absolute abundances and, therefore, calculate species' relative abundances. For this comparison, however, we will now account for the different sizes of each species by using a mean size for each species based on the results of Schmidt et al. (2004) and will recalculate the modeled relative abundances accordingly. This allows for a sound comparison with the core-top data, which will, i.a., likely be evident in the newly introduced and considered Bray-Curtis similarity measure. We will add this similarity analysis to the manuscript (i.e., to sections 3.1 and 4.1.1) to provide a thorough model-data-comparison.
We would like to emphasize that the patterns of vertical and/or seasonal abundance are independent of the amount of carbon per shell (as long as there is no significant and systematic size variability). This allows us to directly compare modeled and observed data.

p9/l18 (and throughout this section) – I'm not sure it makes sense for "maximum production" to be "year-round." Could you clarify?
That is a very good point. Here, we actually wanted to say that uniform and/or constant species

fluxes occur year-round, thus no seasonal peak is evident in the species production. We will change the wording throughout this section accordingly.

section 3.3 – might be helpful to define what you mean by "surface" and "subsurface" as these are pretty general terms but are being used as if the authors have a fairly specific depth range in mind. Thank you for pointing this out. We will provide more precise depth ranges throughout section 3.3 and will avoid especially the general term "subsurface". The surface is in general defined from 0 to 10m water depth, which corresponds to the first vertical layer of the used model configuration.

p12/l30 –"prefer thriving" -> "thrive"
Done.

p12/l35 – delete "largely"
Done.

p14/l4 – delete "among each other"
Done.

p14/l11 – delete "preferably"
Done.

p14/l31 – "cold to transitional" compares a temperature to a zonation
We will change "transitional" to "temperate" to be consistent in the wording.

p15/l22 – a -> the
Done.

p17/l2 – might be better to describe these as short time series as compared to plankton tows which really are "snapshots"
We agree and will now describe sediment trap time series as short time series rather than snapshots:
*"[...] span at most a few years and, hence, represent short time series that [...] plankton tow samples represent snapshots (of one particular day) [...]"*

p17/l18 – or genotypes or phenotypes?

We agree that genotype is a more suitable term in this regard and will change the wording accordingly.

p17/l26 "a few"?

Done.

Figure 6 is extremely difficult to read given the mix of opacity and multiple symbols and colors. Is there a better way to present this data?

We agree and will try to find a better solution to present the data when evaluating the new results.

Figures 6 and 7 (a-c) suggest a quite poor fit of modeled data to sediment trap observations. i.e. 7c shows the model completing missing the flux timing of bulloides in JGOFS34. The authors include an overview or why there might be some data-model mismatch, but I think a wider discussion of why and how this could impact or limit interpretation of model results is warranted.

Please refer to our response to your first comment, where you address the same issues.

References:

- Danabasoglu, G., S. C. Bates, B. P. Briegleb, S. R. Jayne, M. Jochum, W. G. Large, S. Peakcock, S. G. Yeager (2012), The CCSM4 Ocean Component, *Journal of Climate*, *25*, 1361-1389.
- Danabasoglu, G., S. G. Yeager et al. (2014), North Atlantic simulations in Coordinated Ocean-ice Reference Experiments phase II (CORE-II). Part I: Mean states, *Ocean Modelling*, *73*, 76–107.
- Fraile, I., M. Schulz, S. Mulitza, and M. Kucera (2008), Predicting the global distribution of planktonic foraminifera using a dynamic ecosystem model, *Biogesciences*, *5*, 891-911.
- Moore, J. K., K. Lindsay, S. C. Doney, M. C. Long, K. Misumi (2013), Marine Ecosystem Dynamics and Biogeochemical Cycling in the Community Earth System Model [CESM1(BGC)]: Comparison of the 1990s with the 2090s under the RCP4.5 and RCP8.5 Scenarios, *Journal of Climate*, *26*, 9291-9312.
- Schmidt, D. N., S. Renaud, J. Bollmann, R. Schiebel, and H. R. Thierstein (2004), Size distribution of Holocene planktic foraminifer assemblages: biogeography, ecology and adaptation, *Marine Micropaleontology*, *50*, 319-338.

---

## Author Comment (AC3) · 29 Mar 2018

**Response to Referee #3:**

**Ref.: Ms. No. bg-2017-429**

**Title: Modeling seasonal and vertical habitats of planktonic foraminifera on a global scale**

We would like to thank reviewer Jelle Bijma for his constructive comments and suggestions, which will help us to greatly improve our manuscript. Based on the comments of all four reviewers we will prepare a new version of our manuscript as outlined below.

However, during the review process, we discovered an error in the underlying ocean model. Unfortunately, the ocean circulation is not correctly represented in the used coarse resolution (i.e., ~3°) model configuration. For a correct representation of the ocean and to yield scientifically consistent results, we had to perform a new model run with a higher horizontal resolution (i.e., ~1°) on a supercomputing system. This model run takes ca. 5 weeks and is currently in the final production phase. At a first glance, the new results will not differ that much from our previous results as the representation of the upper ocean, where the analyzed foraminiferal species live, was actually reasonably well simulated in the coarse resolution model configuration compared to, e.g., the World Ocean Atlas 2013. We expect that the distribution of only a few species might be affected, when using the higher resolution model configuration with a more realistic representation of the ocean physics. Since we have not yet obtained the final results, we were not always able to provide detailed answers to your comments and had to keep our responses rather general.

Please find, in the following, the original comments in black and our responses in light blue; the indicated page and line numbers refer to the previously submitted manuscript.

Referee #3 comments:

Scientific significance: Excellent

The manuscript by Kretschmer et al. represents a substantial contribution to scientific progress within the scope of Biogeosciences. It is the latest one in a series of "foram-flux modelling" papers from the Bremen group. In 2006, Zaric et al. Developed the first empirical model that described globally the fluxes of planktonic foraminifera at species level in dependence of sea-surface temperature, mixed-layer depth and export production. Over the years, the foram model itself, its parameterization, and its implementation and coupling to other models has evolved (e.g. Fraile et al., 2008; 2009; Kretschmer et al., 2016). The aim of all of these papers has always been to project the effect of changing environmental conditions on species distributional patterns in time and space. The current paper adds a vertical dimension to the existing foram model by applying the previously used spatial parameterization of biomass as a function of temperature, light, nutrition, and competition on depth-resolved parameter fields.

Scientific quality: good

The scientific approach and methods are valid. The results are discussed appropriately but the discussion lacks a critical analysis of the model-data comparison beyond the caveats mentioned in section 4.2 "Comparison with local observations".

Please refer to our response to reviewer 2 for proposed additions to the discussion regarding this point.

Even though the model-data-comparison revealed several discrepancies and is subject to caveats, the model produces nonetheless seasonal and vertical abundance patterns that are consistent with our current understanding and which emerge without any explicit parameterization of abundance in time and space. These patterns emerge from the model itself.

In addition, each model component (i.e., POP2, BEC, PLAFOM2.0) of the used model configuration consists of a rather complex model structure itself and rendering sensitivity experiments will be very time-consuming, expensive and non-trivial. We find that for a first try we obtain very good results.

The authors write on p. 17 line 22-23: "This vertical migration of planktonic foraminifera during their ontogeny cannot be reproduced by PLAFOM2.0 as the model parameterizations do not include the individual species' life cycles.". It is quite understandable that implementing true reproduction cycles of cohorts of foraminifera, including "real" population dynamics and ontogenetic migration is beyond the present manuscript. Hence, the model does not calculate absolute or relative numbers of a certain species within a certain ontogenetic size class based on reproductive success and size specific growth- and mortality-rates, but rather calculates changes in species specific carbon concentration (in mmol C m−3), which can be converted to numbers afterwards.

There is nothing wrong with this approach but it means that the parameterization of PLAFOM2.0 is based on practical "sum" or "composite" parameters. These are then used to tune the model outcome to the overall data. For instance, growth of all species is approximated using a modified form of Michaelis-Menton kinetics in dependence of species specific food availability and temperature sensitivity (Fraile et al., 2008). To account for the light dependence with depth, influencing the growth of only symbiont bearing foraminifera, the authors included a "photosynthetic growth rate". They use "......a similar approach as Doney et al. (1996) and Geider et al. (1998), who determined phytoplankton growth rates by available light and nutrients..... (p.5 line 15-17)". Such a parameterization is normally used for phytoplankton, that has orders of magnitude higher densities and cell division rates that respond very fast (within a day) and directly

to light and nutrients. The symbiont bearing forams in this manuscript obey a (semi) lunar reproduction cycle and occur in densities that are very much lower, such that a "phytoplankton" kind of response cannot be expected. The authors use it as an additional tuning parameter for symbiont bearing forams next to food preference and temperature to develop species specific depth (light/nutrient) habitat preferences. Although it is a valid approach, the authors should clearly state that it is artificial.

Here, we applied a similar approach as Doney et al. (1996) and Geider et al. (1998) as a first approximation to account for a photosynthetic growth rate for the symbiont-bearing species. We are aware that a phytoplankton kind of response to light is not transferable one to one to the response of planktonic foraminifera. We will make this more clear in the manuscript and we will state that this approach is a first approximation, and in that way it should be considered as rather artificial. Nevertheless, we also think that this is a valid approach, given that the photosynthetic growth rate accounts in numerical terms most likely only for the smallest proportion of the total growth.

Growth is balanced by mortality, which is not a formulation for "real" mortality but another tuning parameter: "we adjusted parts of the mortality rate equation to improve the model accuracy (p. 5 line 8-9).".

Overall, there are many factors that allow tuning, e.g. "p% represents the fraction of photosynthesis contributing to growth (p.5 line 31)". Interestingly, the authors have a higher p% for T. sacculifer (0.4) than for G. ruber (0.3), where I would have done it the other way around (see my comments on these species further below).

Here, we followed Lombard et al. (2011), who also used a somewhat higher $p_{\%}$ for *T. sacculifer* (0.40) than for *G. ruber* (0.37). We performed a few short preliminary test runs using different $p_{\%}$-values but obtained the best results on a first glance by using the given parameter values. We were not able to perform a suite of sensitivity experiments with changing the $p_{\%}$-values due to the long runtime of the used complex model configuration.

Another tuning factor is the temperature dependence of the predation term: "......we followed Moore et al. (2004) and adjusted the temperature dependence of the predation term (MLpred in mmolCm$-3$s$-1$) (p.6 line 3-4). Also "....we included a stronger competitive behavior of G. bulloides by adjusting the free parameters in the competition term. (p.6 line 10-11). Having collected planktonic foraminifera by SCUBA diving for many, many years and looking at average typical blue water densities of ca. 10 specimens per m3 per species, and 3 dominant species in an assemblage, it is hard to believe that they compete with each other for resources as each of

them occupies a space of only a few mm3 and they are stationary in the water column.

A good point indeed. Whether or not planktonic foraminifera compete directly is a field of active research. However, we would like to point out that even though foraminifera occur at very low densities and may never directly meet, they are still likely to compete for scarce resources. It is therefore reasonable to include a competition term in the model.

Certain boundary conditions also correct model misfits, e.g. "...zero fluxes have been replaced by half of the observed minimum flux. (p.7 line 25-26)".

All of these parameters were introduced to allow a good fit between model output and data but maybe not for the right reason. As such, we do not know how realistic this parameterization represents real planktonic foraminiferal population dynamics which is more complex (including lunar based reproduction cycles, ontogenetic migration, etc.).

This is true, but nevertheless we are able to simulate the seasonal and vertical habitat of the five considered foraminiferal species remotely realistic using our approach. However, for a more realistic representation of planktonic foraminiferal population dynamics, PLAFOM2.0 needs to be extended by, e.g., considering the ontogenetic migration, reproduction cycles as well as additional foraminiferal species. Thus, PLAFOM2.0 will become more complex and more parameters have to be introduced. In addition, using, e.g., reanalysis data as forcing instead of a climatological forcing could also lead to a more realistic representation of the modern foraminiferal population dynamics when considering a point-by-point comparison with present-day data.

However, even if our understanding of foraminiferal population dynamics will be largely improved in the future due to, e.g., more laboratory experiments, and if we are able to properly translate those complex processes into model code, we will still only be able to provide an approximation of the real dynamics.

Winter mixing, thermocline shoaling and annual irradiation changes are probably important parameters controlling foram population dynamics just as certain density layers may be important for gamete fusion in real foram life. I'm not sure how well these features are implemented in the models.

This is a very good point and all those processes you mentioned likely affect the dynamics of the foraminiferal population. Here we used an ocean-ice-only model configuration and applied a climatological forcing to obtain our results. Hence, there is no explicit interaction between the ocean and the atmosphere and additionally an inter-annual variability of the forcing variables can be excluded. In addition, the lower the resolution of the ocean model the less well represented are

processes such as winter mixing, thermocline shoaling, and upwelling. Since we are now going to present results of a 1° ocean model simulation, most of these processes will likely be better represented than in the previously used 3° simulation, not only because of a more realistic representation of the ocean physics, but also due to the higher resolution, which could likely improve our model results. However, some small-scale processes, oceanic fronts, river runoff areas, and coastal upwelling regions might most likely still not be well represented. In order to analyze inter-annual variability of the foraminiferal population and to investigate how annual radiation changes influence the population dynamics, the model system should be forced with reanalysis data rather than climatologies. Furthermore, using a fully coupled model configuration initialized from reanalysis data could also provide information on how annual changes in the atmosphere feed back on the foraminiferal population dynamics. This, however, was beyond the scope of this study. Here, we actually aimed for an approach that is as simple and general as possible, such that we specifically avoided an explicit parameterization of depth. This way our approach is also easier to follow and we can more easily ensure the reproducibility of our study.

The bottom line is that, even though I appreciate the model and the manuscript a lot, I would like to see a discussion on these issues and if possible a statistical verification of the model performance. The description of the results and the discussion on modeled geographical ranges, seasonal and vertical distribution, as well as on the modeled seasonal variability of depth habitat, lacks a statistical treatment of the data. How good is the model performance and how sensitive is it to each of the model parameters?

Here, we did not perform a sensitivity study in regard of the different model parameters, first, because the runtime of this new model configuration is too long (with a model throughput of ~11-20 simulated years/day for the 3° model configuration and/or a model throughput of ~9.5 simulated years/day for the 1° model configuration depending on the machine capacities) to yield scientifically reasonable results and, second, because Fraile et al. (2008), who introduced PLAFOM (which is the base of PLAFOM2.0), already performed a sensitivity study of the free parameters. Fraile et al. (2008) modified the values chosen for the foraminifera module and quantified the sensitivity by calculating the change in the root mean square error between each sensitivity experiment and the standard run. They found that none of the parameters led to a uniform change for all species and that not surprisingly the parameter controlling the temperature tolerance range (i.e., $\sigma$) seems to be the most sensitive parameter (see Table 3 of Fraile et al., 2008). Since PLAFOM2.0 is in its base form identical to PLAFOM, we did not feel the need to perform another sensitivity analysis and also due to the high computational costs. However we will briefly discuss the sensitivity analysis of Fraile et al. (2008) in section 2 to assure that our applied approach is valid and that our results are in general reliable.

I would appreciate a more quantitative treatment of the model performance instead of statements like "The predicted global distribution patterns of the five considered planktonic foraminiferal species are in good agreement with the core-top data (Figure 2) (p. 11 line 14-15)?

To perform a more quantitative model-data-comparison and to provide some measure of confidence, we will now calculate the Bray-Curtis index of similarity between the model and the core-top data. For this calculation, we will account for the different sizes of each species by using a mean size for each species based on the results of Schmidt et al. (2004) and recalculate the modeled relative abundances accordingly. We will add this analysis to sections 3.1 and 4.1.1 to provide a more thorough model-data-comparison.

The discussion on the global distribution patterns is mostly related to temperature. What about the other parameters: food, nutrients, productivity, light, etc.?

Our results indicate that the habitat variability and the foraminiferal distributions are primarily driven by temperature and for the colder water species (*N. pachyderma*, *N. incompta*, *G. bulloides*) also by food supply. This was also shown by Fraile et al. (2008) and Kretschmer et al. (2016). Fraile et al. (2008) demonstrated that the foraminiferal distribution patterns respond most sensitively to changes in the temperature tolerance ranges of the individual species, indicating the strong temperature dependence of the foraminiferal population dynamics. Therefore, we mainly relate our results to temperature, but also discuss the food dependency extensively (see sections 4.1.3 and 4.2); the other parameters, however, seem to be less effective.

How does it compare to the "Longhurst Biogeographical Provinces". He partitioned the world oceans into provinces ("Ecological Geography of the Sea") based on the prevailing physical factors as a regulator of phytoplankton distribution, including temperature, photic depth, mixed layer depth etc. (e.g. Longhurst 1995; 1998).

The underlying parameterizations used in PLAFOM itself are based on the parameterizations used in the ecosystem model of Moore et al. (2002a) and do not include a spatial parameterization. Since Longhurst's partitioning of the ocean is more or less only descriptive, a comparison with our model results is in our understanding not appropriate. In addition, to properly compare the simulated global distribution patterns with Longhurst (1995, 1998), we would have to take into account the characteristics of each biogeographical province in the model parameterizations, which would most likely result in an overfitting.

Having "fixed" model parameters simulates so called "habitat tracking" of the forams through the seasons (but also on timescales of climate change or on glacial/interglacial cycles). This is a very important aspect to verify and would call for a section/paragraph by itself (see also Rebotim et al.,

2017). For instance, on p15 line 23-25 you write "Rebotim et al. (2017) identified an annual cycle in the habitat of T. sacculifer and N. incompta in the subtropical eastern North Atlantic. Both species appear to descend in the water column from winter to spring and reach their deepest habitat in spring to summer before ascending again to a shallower depth towards winter (Rebotim et al., 2017).". How does this fit the "habitat tracking" picture? The authors could probably use observations on G. ruber and T. sacculifer for that as well. I may be wrong but I always thought that G. ruber lives closer to the surface than T. sacculifer (see also table 3 in Rebotim et al., 2017)? From laboratory experiments I know that T. sacculifer can handle living prey such as copepods much better than G. ruber while the latter seems to rely more on symbiont carbon, i.e. shows a more "autotrophic" lifestyle. Is it possible to see this in the data based on a more rigorous model-data comparison?

Our results reveal that outside their preferred habitat, where they naturally have to face a changing environment, the seasonal occurrence of both *G. ruber* (white) and *T. sacculifer* is limited to the warm surface layer, whereas in the low latitudes both species exhibit a weak seasonal cycle in their depth habitat (see Figure 5 of the manuscript). This indicates that both species adapt to changing environmental conditions by adjusting their habitat to local circumstances, which is consistent with the concept of habitat tracking. We will add this to section 4.1.3.

In addition, we are not able to derive dietary preferences from the model, as those are prescribed for the underlying model parameterizations. In the model parameterizations, we do not distinguish between the heterotrophic and/or autotrophic lifestyle of the species, just the parameters determining the preference for a food source differ slightly among the species (see Table 1 in Fraile et al., 2008). Additionally, those parameters introduced to account for the light sensitivity of *G. ruber* (white) and *T. sacculifer* with depth differ also among them (see Table 1 of the manuscript). So by prescribing light sensitivity and food preferences a similar depth ranking compared to observations already emerges from the model. Nevertheless, for a more rigorous model-data-comparison a sensitivity study regarding the species-specific food preferences should be performed. We will bear this in mind for a future model development.

The results of the point-by-point comparative analysis for each site and species as provided in the Supplement (Figures S3 and S4) are very helpful but also show that the model is far from perfect and sometimes there is a complete mismatch. I would have appreciated a sensitivity study to determine the hierarchy of factors for the different species controlling the shell export fluxes regional and seasonal (including e.g. bimodal patterns) as well as the vertical distribution (including ALD). This would probably be a paper by itself but in my view a very important one.

This is a real good and true point and we also think that such a sensitivity study would improve PLAFOM2.0. Therefore, we agree that such a study would be very important and should be

considered in the future. However, due to the high computational costs it is at present not feasible to perform this analysis. In addition, such a study would require using observational data with realistic year to year variability as forcing, but also for the model validation, which would in turn require a sensitivity study for each sediment trap/plankton tow by itself.

Based on the sensitivity analysis of Fraile et al. (2008) and also on our own results it seems that temperature has the strongest influence on the foraminiferal distribution regarding both the seasonal and vertical habitat. In particular, the distribution of each individual foraminiferal species seems to react most sensitively to changes in the individual temperature tolerance ranges (see Fraile et al., 2008). However, to further assess the sensitivity of the model to the chosen parameters especially in regard to the vertical distribution of the foraminiferal species a thorough sensitivity analysis should be performed in an independent study, which we will bear in mind for the future. Nevertheless, even after a further tuning based on such a sensitivity analysis the model will be far from perfect and discrepancies between the model data and the observations may always be present, as the caveats mentioned in section 4.2 will still be valid.

Presentation quality: good/fair

Although the scientific results and conclusions are presented in a relatively clear and well-structured way it is not easy to grasp why the model underestimates e.g peak amplitude. What would happen if growth in the equation is increased or mortality is decreased? I sometimes wondered why the authors didn't play more with the model or used statistical techniques to quantify data-model mismatch (this is the reason for the "fair" mark).

As already mentioned, due to the long runtime of the model and, hence, the high computational costs we were not able to perform a thorough sensitivity analysis and just performed some very preliminary and short test runs to evaluate the model performance. In addition, since PLAFOM2.0 is based on PLAFOM, which has been tested and validated thoroughly (e.g., Fraile et al., 2008; Kretschmer et al., 2016), and since our aim was to demonstrate the applicability and the usability of PLAFOM to simulate the vertical distribution of individual foraminiferal species when combined with a complex 3D model configuration (such as CESM1.2(BGC)) without explicitly parameterizing the vertical dimension, we on purpose decided to not test what would happen if we change the given parameter setting. However, to better quantify the model-data-mismatch, we will calculate the Bray-Curtis index of similarity between the model and the core-top data. This way, we can provide some measure of confidence regarding the general model performance. In addition, we will also extend the discussion regarding the model-data-comparison by also considering potential mismatches due to the ocean model. Nevertheless, we will not entirely be able to unequivocally differentiate between the different model components (i.e., POP2, BEC or PLAFOM2.0) and their individual share likely leading to the model-data-mismatch.

The number and quality of figures/tables is good and the supplementary material is very appropriate. The English language is very good.

Thank you!

Minor corrections:

On page 2 line 18-20: "…………the lunar cycle and/or the structure of the water column), which influence the species-specific depth habitats (including their mean living depth and vertical migration) (e.g., Fairbanks and Wiebe, 1980; Fairbanks et al., 1982; Schiebel et al., 2001; Simstich et al., 2003; Field, 2004; Salmon et al., 2015; Rebotim et al., 2017), the only attempt to model the vertical habitat is by Lombard et al. (2011).", and on page 17 line 20-23: "Several studies from different areas also showed that the main habitat depth of some species increases from the surface to deeper water layers during shell growth (Peeters and Brummer, 2002; Field, 2004; Iwasaki et al., 2017). Although I appreciate all the references that you list for ontogenetic migration and lunar cycle, there are only a few papers that specifically deal with very detailed population dynamics, lunar cyclicity and ontogenetic migration of planktonic forams that could/should be mentioned here (it was one of the first topics I studied when starting to work on planktonic foraminifera): Bijma et al., 1990; Bijma, 1991; Bijma and Hemleben, 1994; Bijma et al., 1994; Hemleben and Bijma, 1994; Schiebel et al., 1997. In my opinion, these references would fit best on p. 19 line 32-34: "……..and by explicitly parameterizing the ontogeny of each individual planktonic foraminifera, thus, by considering the changes in the species' life cycles with depth, could considerably improve the model.".

Thank you for pointing this out. We will add the mentioned references accordingly.

P. 9 line 27-30: "Although seasonal changes in the modeled foraminiferal peak fluxes with temperature are evident, all five species exhibit an almost constant peak amplitude (i.e., the maximum concentration divided by the annual mean) in their preferred habitat, which is, i.a., limited by temperature. Outside their preferred living conditions the peak amplitudes increase for most of the species considerably (Figure 3).". It has not become clear to me what it means when "peak amplitude" is large or small in terms of real population dynamics ("bloom"?) and what it means in terms of model performance?

The maximum seasonal abundance or flux in itself is not a very useful parameter that can be compared among different regions/studies. Population dynamics can be much better (if not only) described in terms of deviations from the mean conditions, for instance, it would be impossible to recognize a bloom event in the absence of knowledge about the mean conditions. Moreover, any seasonal or vertical weighting of the proxy signal – and constraining this was the main motivation to develop the model – varies as a function of the relative departure from the mean. To be more

clear about this, we will rewrite this paragraph on page 9 in section 3.2 and will provide a better explanation.

P. 14 line 26-28: "This would explain why the highest modeled concentrations of T. sacculifer occur at shallower depths compared to G. ruber (white) (see Figures 4d-e and 5d-e).". Strictly speaking this doesn't explain it because this is what you put into the model in the first place (see my comments above).

Actually, this is a perfect example how the habitat emerges from the model. We only prescribe the light sensitivity and still obtain the right depth ranking. Throughout the model code, we specifically did not specify the depth ranking. We will, however, rewrite this sentence to avoid confusion:

*"This is to some degree also indicated in our results, as the highest modeled concentrations of T. sacculifer occur at shallower depths compared to G. ruber (white) (see Figures 4d-e and 5d-e)."*

P. 16 line 18: "G. bulloides, however, is found year-round close to the surface along the.....". Write the genus name full at the beginning of a sentence.

Done and applied throughout the manuscript.

References:

- Bijma, J., Erez, J. and Hemleben, C. (1990) Lunar and semi-lunar reproductive cycles in some spinose planktonic foraminifers. Journal of Foraminiferal Research 20, 117-127.
- Bijma, J. (1991) Lunar pulses of carbonate output by spinose planktonic Foraminifera, in: Reid, P.C., Turley, C.M., Burkill, P.H. (Eds.), Protozoa and Their Role in Marine Processes. NATO ASI Series G: Ecological Sciences. Elsevier, Plymouth, pp. 353-354.
- Bijma, J. and Hemleben, C. (1994) Population dynamics of the planktic foraminifer Globigerinoides sacculifer (Brady) from the central red sea. Deep-sea research part I: oceanographic research papers 41, 485-510.
- Bijma, J., Hemleben, C. and Wellnitz, K. (1994) Lunar-influenced carbonate flux of the planktic foraminifer Globigerinoides sacculifer (Brady) from the central red sea. Deep-sea research part I: oceanographic research papers 41, 511-530.
- Hemleben, C. and Bijma, J. (1994) Foraminiferal population dynamics and stable carbon isotopes., in: Zahn, R., Pedersen, T.F., Kaminski, M., Labeyrie, L. (Eds.), Carbon Cycling in the Glacial Ocean: Constraints on the Ocean's Role in Global Change. Elsevier, Fellhorst, pp. 145-166.
- Longhurst, A. (1995) Seasonal cycles of pelagic production and consumption. Progress in Oceanography 36, 77-167.

- Longhurst, A. (1998) Ecological Geography of the Sea ACADEMIC PRESS

- Schiebel, R., Bijma, J. and Hemleben, C. (1997) Population dynamics of the planktic foraminifer Globigerina bulloides from the North Atlantic. Deep Sea Research 44, 1701-1713.

- Fraile, I., M. Schulz, S. Mulitza, and M. Kucera (2008), Predicting the global distribution of planktonic foraminifera using a dynamic ecosystem model, *Biogeosciences*, *5*, 891-911.

- Kretschmer, K., M. Kucera, and M. Schulz (2016), Modeling the distribution and seasonality of *Neogloboquadrina pachyderma* in the North Atlantic Ocean during Heinrich Stadial 1, *Paleoceanography*, *31*, 986-1010.

- Lombard, F., L. Labeyrie, E. Michel, L. Bopp, E. Cortijo, S. Retailleau, H. Howa, and F. Jorissen (2011), Modelling planktic foraminifer growth and distribution using an ecophysiological multi-species approach, *Biogeosciences*, *8*, 853-873.

- Moore, J. K., S. C. Doney, J. A. Kleypas, D. M. Glover, I. Y. Fung (2002a), An intermediate complexity marine ecosystem model for the global domain, *DSR II*, *49*, 403-462.

- Schmidt, D. N., S. Renaud, J. Bollmann, R. Schiebel, and H. R. Thierstein (2004), Size distribution of Holocene planktic foraminifer assemblages: biogeography, ecology and adaptation, *Marine Micropaleontology*, *50*, 319-338.

---

## Author Comment (AC4) · 29 Mar 2018

**Response to Referee #4:**

Ref.: Ms. No. bg-2017-429

**Title: Modeling seasonal and vertical habitats of planktonic foraminifera on a global scale**

We would like to thank the reviewer for the constructive comments and suggestions, which will help us to greatly improve our manuscript. Based on the comments of all four reviewers we will prepare a new version of our manuscript as outlined below.

However, during the review process, we discovered an error in the underlying ocean model. Unfortunately, the ocean circulation is not correctly represented in the used coarse resolution (i.e.,  $\sim$ 3°) model configuration. For a correct representation of the ocean and to yield scientifically consistent results, we had to perform a new model run with a higher horizontal resolution (i.e.,  $\sim$ 1°) on a supercomputing system. This model run takes ca. 5 weeks and is currently in the final production phase. At a first glance, the new results will not differ that much from our previous results as the representation of the upper ocean, where the analyzed foraminiferal species live, was actually reasonably well simulated in the coarse resolution model configuration compared to, e.g., the World Ocean Atlas 2013. We expect that the distribution of only a few species might be affected, when using the higher resolution model configuration with a more realistic representation of the ocean physics. Since we have not yet obtained the final results, we were not always able to provide detailed answers to your comments and had to keep our responses rather general.

Please find, in the following, the original comments in black and our responses in light blue; the indicated page and line numbers refer to the previously submitted manuscript.

**Referee #4 comments:**

This paper builds upon preexisting work modeling planktonic foram distributions in the global oceans via a coupling to CESM's ocean model. The goal is to better understand how the vertical distribution of foraminifera species varies seasonally and throughout larger climatic changes in the ocean. The paper is generally well written, clear, and broadly does a fine job demonstrating the usefulness of the model. It is also very thorough in its examination of the model's performance against available data. The methods seem robust and I can recommend that with some minor revisions (mostly grammar and clarity) the paper be published in Biogeosciences.

I must acknowledge that I am not an expert on the biogeochemistry of planktonic forams in any way and hope the other reviewers can address the methods and parameterizations employed in this

1

paper in particular. I can instead comment on the benefit of this work and the need for such proxy system models for the robust interpretation of paleoceanographic records via the use of PLAFOM2.0 + CESM1.2. To that end, my first major comment is that the authors can focus more in the introduction and conclusion on the body of literature developing forward models, or proxy system models, for understanding paleoclimate proxies and introduce this work as a part of this group of literature. A major effort has been underway to build proxy system models, link them with GCMs, and make these models publicly available, and this paper is absolutely in this category and should make as much clear.

See for example:

- Dee, S., et al. "PRYSM: An open-source framework for PRoxY System Modeling, with applications to oxygen-isotope systems." Journal of Advances in Modeling Earth Systems 7.3 (2015): 1220-1247.
- Evans, Michael N., et al. "Applications of proxy system modeling in high resolution paleoclimatology." *Quaternary Science Reviews* 76 (2013): 16-28.
- Schmidt, Gavin A. "Forward modeling of carbonate proxy data from planktonic foraminifera using oxygen isotope tracers in a global ocean model." *Paleoceanography* 14.4 (1999): 482-497.

We agree that over the last decades proxy system/formation models have become more and more important for understanding paleoclimate proxies and that PLAFOM2.0 belongs to a series of different proxy system/formation models. We will briefly introduce PLAFOM2.0 as part of this large group of proxy system models in section 2.1 (page 3, line 24):

"Thus, PLAFOM2.0, as belonging to a suite of proxy system models (e.g., Pollard and Schulz, 1994; Schmidt, 1999; Fraile et al., 2008; Evans et al., 2013; Dee et al., 2015; Völpel et al., 2017), might add to the improvement of interpreting paleoclimate reconstructions."

You might also consider mentioning (in the intro or discussion) the potential for PLAFOM to assist in data assimilation exercises for periods extending back further than the last millennium, for example. A number of papers look at the impacts of using process-based models in the DA framework and this is another application of your model. See work of Hugues Goosse's lab (e.g. Goosse, Hugues, et al. "Reconstructing surface temperature changes over the past 600 years using climate model simulations with data assimilation." *Journal of Geophysical Research: Atmospheres* 115.D9 (2010)), as well as:

• Steiger, Nathan J., et al. "Assimilation of time-averaged pseudoproxies for climate

2

reconstruction." Journal of Climate 27.1 (2014): 426-441.

- Dee, Sylvia G., et al. "On the utility of proxy system models for estimating climate states over the common era." *Journal of Advances in Modeling Earth Systems* 8.3 (2016):
- Hakim, Gregory J., et al. "The last millennium climate reanalysis project: Framework and first results." *Journal of Geophysical Research: Atmospheres* 121.12 (2016): 6745-6764

Thank you for pointing this out. We agree that PLAFOM2.0 has the potential to be used in the data assimilation framework and we will add a statement in this regard to section 2.1:

"In addition, PLAFOM2.0 has the potential to be used in the paleoclimate data assimilation framework, which provides a promising technique to estimate past climates (see, e.g., Dee et al., 2016; Goosse et al., 2010; Hakim et al., 2016; Steiger et al., 2014)."

In Section 4, it would be nice if the authors could provide a more quantitative data-model comparison technique—you identify areas where the model does not well simulate the observations and Figure 2 summarizes this to some extent, but perhaps you could include an additional table or figure or even compute something like the RMSE for each oceanic province? Or the mean RMSE for each species over all of the locations where core-top data exist?

Please refer to our response to a similar comment by reviewer #1 regarding a more quantitative model-data-comparison.

Finally, in the discussion, you assert (correctly) that your new model is a powerful tool for separating the independent influences of habitat and climate on foram reconstructions. I think this paper would be greatly strengthened by a demonstration of this. Can you take a well-known and vetted reconstruction and apply this model in a meaningful way to reassess the climatic interpretation? I think this would show the power of forward modeling in this field to make more robust assessments of uncertainties in oceanic climate changes... And I think having this demonstration would add weight to the assertions you make in your Discussion section.

This is a good point and we agree that such a demonstration would add to a better understanding of climate change. In a next study, we plan on performing a model run with, e.g., Last Glacial Maximum climate conditions to test the applicability of our modeling approach. Here, in this study we simply wanted to test if the existing planktonic foraminifera model is able to reproduce species-specific habitats when combined with a model configuration that resolves the vertical.

Minor / Line by Line comments: (Page-Line)

2-10 awkward paragraph break, consider revising

We agree and will delete the paragraph break.

2-13 comma after perspective,

Done.

2-20 Have you investigated/reviewed Schmidt et al., 1998, 1999? These papers I believe address vertical migration of foram species in the water column—worth checking/citing if appropriate.

- Schmidt, Gavin A. "Oxygen-18 variations in a global ocean model." *GRL* 25.8 (1998): 1201-1204.
- Schmidt, Gavin A. "Forward modeling of carbonate proxy data from planktonic foraminifera using oxygen isotope tracers in a global ocean model." *Paleoceanography* 14.4 (1999): 482-497.

Thank you for referring to those two studies. In both studies, Schmidt does unfortunately not address the vertical migration of foraminifera. Schmidt (1998, 1999) investigates the distribution of oxygen isotopes in seawater and subsequently calculates equilibrium calcite values based on different temperature equations. This, however, is beyond the scope of our present study and, hence, citing those studies is not appropriate.

2-26 need comma after behavior. Done.

Check for needed commas and small grammatical errors throughout text.

3-6 comma after estimate, Done.

3-13 this phrase is awkward, revise ("with the biogeochemical model being enabled")
We will revise the phrase as follows:
"[...] as an off-line module into the ocean component of the Community Earth System Model, version
1.2.2 (CESM1.2; Hurrell et al., 2013), with active ocean biogeochemistry (which is denoted as

**CESM1.2(BGC) configuration)."**

3-15 change "aimed for" to 'aimed to' Done.

3-16 change "at geologic timescales" to "ON geologic timescales" Done.

Check for similar awkward language throughout. Done.

3-23 comma after configuration, Done.

3-30 no paragraph break. Done.

4-9 what do you mean by 'data models' for the atmosphere, etc.? Are you not using the fully coupled simulations and using some kind of statistical representation of the other components?

The CESM data models are "non-active" model components that read external data, modify that data (e.g., interpolate that data in time and space), and subsequently return the final data fields to the coupler. In this study, we did not perform a fully coupled simulation. Here we analyze an ocean-ice-only simulation with active ocean biogeochemistry coupled to data models for the atmosphere, land, and river routing. Since "data model" is a common term in the CESM model community, we also used it to be consistent with other publications using the CESM1.2(BGC) configuration. However, for a better understanding, we will revise the sentence as follows:

"Here we performed an ocean-ice-only simulation with active ocean biogeochemistry, whereby the ocean model is coupled to both the sea ice model and data models for the atmosphere, land, and river routing, which provide the respective input data for the considered simulation."

**Heading 2.4 consider changing this to "Coupled GCM Setup" ?**

Since our results are based on an ocean-ice-only simulation, a heading change to "Coupled GCM setup" is not appropriate. Nevertheless, we will change heading 2.4 to "Model simulation", which is

**more accurately describing this section.**

7-15 missing space before new sentence. Done.

8-21 comma after 'life cycle,' Done.

Throughout section 3, be extremely clear about whether you are referring to observations vs. the model simulation of foram distributions/abundances etc. The reader gets a bit lost in the data-model comparison here unless that's super clear.

In section 3, we actually just describe model results and do not provide a model-data-comparison. To be more concise and clear about that, we will revise this section accordingly. In addition, we will also revise section 4 to be more clear about when we refer to observations and/or model output.

16-29 no comma after 'data' Done.

**16-30 this is a run-on sentence-consider shortening/rewriting**

We will revise this run-on sentence by splitting it into two parts:

"The emergence of seasonal and vertical habitat patterns consistent with observational data provides important support for our modeling approach. Nevertheless, a more detailed comparison with observations is warranted to gain further insight into the model behavior."

I appreciate the thorough discussion of the model – data comparison limitations on page 17. Thank you!

**Figure 5 has some strange cropping issues along top margin.**

Thank you for pointing this out. We will check for this and will adjust Figure 5 accordingly after obtaining our final results.

**References:**

• Fraile, I., M. Schulz, S. Mulitza, and M. Kucera (2008), Predicting the global distribution of

planktonic foraminifera using a dynamic ecosystem model, *Biogeosciences*, 5, 891-911.

- Pollard, D. and M. Schulz (1994), A model for the potential locations of Triassic evaporite basins driven by paleoclimatic GCM simulations, *Global and Planetary Change*, *9*, 233-249.
- Schmidt, D. N., S. Renaud, J. Bollmann, R. Schiebel, and H. R. Thierstein (2004), Size distribution of Holocene planktic foraminifer assemblages: biogeography, ecology and adaptation, *Marine Micropaleontology*, 50, 319-338.
- Völpel, R., A. Paul, A. Krandick, S. Mulitza, and M. Schulz (2017), Stable water isotopes in the MITgcm, *Geosci. Model Dev.*, *10*, 3125-3144.

---

## Author Response (AR1)

**Ref.:  Ms. No. bg-2017-429**

**Title: Modeling seasonal and vertical habitats of planktonic foraminifera on a global scale**

Dear Lennart,

First of all we would like to thank the reviewers for their constructive comments and suggestions, which greatly helped to improve our manuscript "Modeling seasonal and vertical habitats of planktonic foraminifera on a global scale".

In response to the reviewers' suggestions and due to an incorrect representation of the ocean circulation in the formerly used coarse resolution (i.e., ~3º) model configuration, we have made the following major changes to the manuscript:

- As we informed you earlier, we have noticed an inconsistency in the ocean state as represented in the originally used 3º physical model resolution setup. To resolve this inconsistency, we performed a new simulation of foraminifera distribution based on a physical model run with a 1º horizontal resolution, which yielded a more realistic ocean state. The spin-up was run for 300 model years, which we consider sufficient to ensure that at least the upper ocean, where the investigated foraminiferal species live, was in equilibrium. As expected, the evaluation of the thus obtained foraminifera distribution did not change the main findings of the paper. However, we note an improvement in the agreement between the model data and the observations regarding the horizontal, vertical and seasonal distribution of the analyzed planktonic foraminifera species, which we ascribe to the better representation of the upper ocean in the new physical model setup.

- In response to the comments of the referees, we carried out a more quantitative model-data-comparison by calculating the Bray-Curtis index of similarity between the model data and the core-top data and we extended the discussion regarding the model-data-mismatch by also considering limiting factors regarding the underlying complex model configuration.

We feel that thanks to the reviewers' suggestions and comments we were able to produce a more robust manuscript, which includes a broader critical analysis of the model-data-comparison and provides a thorough analysis of the overall model performance.

We append here the point-by-point responses to each review as well as the revised version of the manuscript (with the changes highlighted in red). All comments provided by the reviewers were taken into consideration and included in the revised version. We hope that our revised manuscript meets the criteria for publication in Biogeosciences.

Kind regards,

Kerstin Kretschmer (on behalf of all co-authors)

**Response to Referee #1:**

We would like to thank reviewer Inge van Dijk for her constructive comments and suggestions, which helped us to greatly improve our manuscript. Please find, in the following, the original comments in black and our responses in light blue; the indicated page and line numbers refer to the revised manuscript.

Referee #1 comments:

I have carefully read the manuscript 'Modeling seasonal and vertical habitats of planktonic foraminifera on a global scale' by Kretschmer and coauthors, which presents a model to predict global concentrations of five species of planktonic foraminifera and their depth habitat. This model could aid paleoclimatologists to correct for habitat depth when using shells of planktonic foraminifera to reconstruct ocean conditions. I need to remark that I have no experience using PLAFOM, or any practical experience with either the BEC model or CESM1.2(BGC) configuration. Therefore, my comments are rather general and an experienced user should review e.g. the use of model parameters and choice of configuration. I only have a couple of remarks that mainly focus on the usability and applicability of the model to reconstruct past depth habitats.

General comments

In general the authors should avoid certain 'model jargon', if they want to convince the broad foraminiferal society to use and apply this model. It is sometimes difficult to follow which steps are taken and assumptions were made to test or simulate certain scenarios (e.g. page 6, lines 23-25).

Thank you for pointing this out. We will change parts of the method section, also according to the higher resolution model configuration, and will include more or delete redundant information, when appropriate, for a better understanding. However, to ensure reproducibility of our study, we cannot avoid using a certain 'model jargon' to explain the applied modeling approach and the used model setup. We already tried to use as little model jargon as possible and provided in all conscience a comprehensible model description.

Even though habitat tracking is very important when using shells of planktonic foraminifera to reconstruct ocean conditions, it is still (more?) crucial to pinpoint the actual calcification depth within the depth habitat, since this is where the calcite is formed. Even though the model can reasonably well predict (globally) the vertical distribution, this does not mean that at this specific depth the environmental signal was 'logged' into the shell. Please include somewhere a couple of sentences on the reconstructed depth habitat compared to the actual calcification depth. Could this be the next step for PLAFOM3.0?

This is a valid point and in a next step, we would like to combine PLAFOM2.0 with a module, which specifically takes this into account and calculates species-specific isotope compositions of the modeled foraminiferal species, such that we could directly infer information about the calcification depth of each species. However, without any information about the species-specific habitats, it is difficult to provide a statement regarding the calcification depths of the individual foraminiferal species. Therefore, we at first intended to simulate realistic species-specific habitat depths and next we plan on obtaining realistic calcification depths. We included a paragraph in section 4.1.3 (p. 18, lines 18-27) regarding a comparison of the reconstructed depth habitat with the actual calcification depth of the individual species:

*"We find that the modeled depth habitats of the five considered foraminiferal species are in agreement with the relative ranking of their apparent calcification depths, but the inferred absolute values of calcification depth are often deeper or show a broader range of depths (e.g., Carstens and Wefer, 1992; Kohfeld et al., 1996; Ortiz et al., 1996; Bauch et al., 1997; Schiebel et al., 1997; Ganssen and Kroon, 2000; Peeters and Brummer, 2002; Anand et al., 2003; Simstich et al., 2003; Nyland et al., 2006; Jonkers et al., 2010, 2013; van Raden et al., 2011). This is not surprising, because PLAFOM2.0 does not model species' ontogeny and cannot capture processes related to ontogenetic depth migration (e.g., Fairbanks et al., 1980;*

*Duplessy et al., 1981). The same limitation applies to estimates of living depth derived from plankton tow data, which often appears to deviate from apparent calcification depths (e.g., Duplessy et al., 1981; Rebotim et al., 2017). Nevertheless, as a first essential step in understanding the variability in calcification depths, PLAFOM2.0 provides a powerful tool that can aid the interpretation of proxy records."*

Section 2.3.1. What about other ocean parameters that vary over geological timescales which might influence growth rates? Like $[PO_4^{3-}]$ (Aldridge et al., 2012, BG) on SNW or the effect of carbonate chemistry on calcification rates? For instance Lombard et al., 2010 found lower growth rates of several species with lowered $[CO_3^{2-}]$ conditions and Davis et al., 2017 (Sci. Rep.) observed lower calcification rates with decreasing pH. Why are these parameters not taken into account in the model? Are these effect minor compared to temperature and food availability?

This is a valid point again, but we are not attempting to model species-specific growth rates (as opposed to Lombard et al., 2011). Rather we aim to more directly estimate foraminifera abundance, which can be compared to the sediment record more directly. The relationship between growth rate and abundance is far from straightforward (cf. Lombard et al., 2011) and we are not aware of studies that have investigated the effect of those parameters on the abundance of planktonic foraminifera. We are aware that other ocean parameters might influence species-specific growth rates. The aim of this study, however, was to test if the existing planktonic foraminifera model is able to reproduce species-specific habitats when combined with a model configuration that resolves the vertical. One has to bear in mind that a model is only a simplification of reality and including more parameters would likely introduce more degrees of freedom and could lead to more model uncertainty and could additionally increase the computational costs. However, for a future model development it is worth considering those parameters. Here it is beyond the scope of this study to include more parameters to determine growth rates.

Section 2.5.2. and 2.5.3. The authors use the sediment trap/plankton tow samples to test the accuracy of the model in predicting seasonality & depth habitats. However, the amount of data used for this comparison is not covering the total range of oceanic settings, since big parts of the ocean are underrepresented. Is it possible to extend this database by adding other published sediment trap data? This way you can show your model can predict depth habitat in a wider range of ocean conditions, which will make it more robust for application in deep time. Just some quick suggestions: Mediterranean Sea: Mallo et al., 2017 BG; SW Atlantic: Venancio et al., 2016 Marine Micropaleontology; Mozambique channel: Steinhardt et al., 2014 Marine Micropaleontology; Panama basin: Thunell et al., 1983 EPSL; Indian Ocean: Guptha et al., 1997 JFR.

The reviewer rightly points out that our data compilation is not comprehensive. However, we pursued the strategy to acquire sediment trap and plankton tow data at more or less the same region to guarantee a consistent model-data-comparison throughout the manuscript when analyzing species-specific seasonal and vertical habitat patterns (see Figure 1b). We agree that this prerequisite limits the number of studies that can be used to evaluate the model, but the underlying data base covers all provinces and provides good estimates of the different species-specific habitats and their variability on a global scale that is sufficient to show the strength and weaknesses of our model.

Figure 2. Is it possible to add an 'offset map', in which you correlate e.g. the coretop data with the model data, to see where the model exactly over-/underestimates the data? This way you would be able to perform some (correlation) statistics, and this would clearly show the areas where the model did not predict the correct distribution. I understand you are trying to capture the global signal (as stated several times in the manuscript), but paleooceanographers are more interested in specific areas when correction for e.g. depth habitat, and these are often also in more complicated oceanic settings (for

example coastal/upwelling/river run off areas).

We included an additional map in Figure 2 that provides a more thorough comparison between modeled and observed assemblages. Therefore, we calculated the Bray-Curtis index of similarity between the model data and the core-top data, such that we provide a measure of confidence. Note for the calculation, we accounted for the different sizes of each species by using a relative size for each species based on the results of Schmidt et al. (2004) and recalculated the modeled relative abundances accordingly. We added this analysis to the manuscript (i.e., to section 3.1, p. 9, lines 11-15) to provide a thorough model-data-comparison. Nevertheless, the used model configuration consisting of three different models (i.e., POP2, BEC, PLAFOM2.0) could hamper a thorough statistical analysis as it is not unequivocally possible to differentiate which component might actually lead to a possible over-/underestimation of the data. Even the now used higher model resolution could likely lead to misrepresentations of small-scale processes, oceanic fronts, river runoff areas, and coastal upwelling regions, and could, thus, account for the model-data-mismatch. In addition, it is not possible to correlate the core-top data with the model data directly, because PLAFOM2.0 calculates foraminiferal concentrations via carbon biomass (i.e., in mmol C/m$^3$) and the core-top samples provide foraminiferal concentrations via number of specimens.

*"For a direct comparison of the observed (i.e., the core-top data) and modeled foraminiferal community composition the Bray-Curtis index of similarity was used. The comparison reveals generally a good fit between the simulated and sedimentary assemblage composition with median Bray-Curtis similarity of ~ 68%. The fit is particularly good in the high latitudes and in the tropics (Bray-Curtis similarity > 80%) and only a few regions (off South America and southern Africa, in the equatorial and North Pacific, and in the eastern North Atlantic) reveal a poorer agreement with similarities of < 50% (Figure 2a)."*

Page 11, line 27-31 and page 12, line 20-21. The authors state that part of the mismatch between the model and coretop data might stem from different genotypes having varying ecological preferences, and therefore their own unique model parameters. If so, does did not create a major bias for the whole model, especially when reconstructing depth habitats in deep time? For geological samples it is not possible to distinguish between genotypes, and therefore certain species might respond different in terms of depth habitat than the model will predict? Also, could it be that certain ecological preferences have changed over time? Can the authors predict how far in geological time you could still use this model to obtain reliable data on global distribution and depth habitat?

The reviewer points out two important considerations: i) cryptic species with different ecological preferences and ii) the question of stationarity. We would argue that both hold for all attempts to use planktonic foraminifera to reconstruct the past ocean. The assumption of stationarity of any proxy is fundamental to all paleoclimate reconstructions. The model can of course only be used for the time that the species have been present and for as long as we have indications that their ecology remained constant (cf. Huber et al., 2000 for *N. pachyderma*). The primary intended use of the model is to apply it to climate conditions covering the Last Glacial Maximum and/or the last couple of glacial-interglacial cycles, but not to deep time, when different species existed or extant species may have had different ecological preferences.

With respect to cryptic species the reviewer is right to point out that this forms an important caveat. However, as the reviewer also mentions, it is often impossible to distinguish between cryptic species in the fossil record, so this caveat applies to any reconstruction using planktonic foraminifera. This is exactly the reason why ecological preferences of cryptic species need to be resolved, so that reconstructions and modeling efforts can be improved. To clarify this point, we added this issue to the end of section 4.1.1 (p. 14, lines 16-23):

*"[...] Likely an even larger part of the discrepancies between the model and core-top data stems from the underlying model parameterizations applied on a global scale, which do not distinguish between distinct genotypes of the different species with potentially varying ecological preferences. Theoretically, this problem could be solved by parameterizing all known*

*genotypes individually and approximating the total morphospecies abundance as the sum of its constituent genotypes. This would allow a comparison with sediment data, but not a diagnosis, since the sediment data provide no information on which genotypes are contained in the assemblages. Interestingly, the generally fair fit between the model and observations suggests that ecological differences between cryptic species are likely limited and that the model provides a useful first-order approximation of global species distribution."*

Minor comments
Page 2, line 18, 32; Page 6, line 16; page 11, line 23: Some problem with bracketing, e.g. double bracketing etc.
We checked for the double bracketing and, where possible, we will delete the unnecessary brackets. However, for some cases (i.e., p. 6, line 19; p. 12, line 26) we will not change the bracketing as this would potentially cause a misunderstanding with the referencing.

Page 6, line 24: quasi-steady
Done.

Page 7, line 15: space missing between '(Figure 1a).' and 'We'
Done.

Page 8, line 5 and page 11, line 17: Arctic Circle
Done.

Page 12, line 10-14. Can you explain the underestimation of the model in scenarios were assemblages are dominated by two species?
Here, we actually meant that the model is not able to capture the full extent of the observed relative abundances in certain areas where a dominance of some species is actually expected.

Page 12, line 21: change or remove 'see'
Done.

References:
- Huber, R., H. Meggers, K.-H. Baumann, M. E. Raymo, and R. Henrich (2000), Shell size variation of the planktonic foraminifer *Neogloboquadrina pachyderma* sin. in the Norwegian-Greenland Sea during the last 1.3 Myrs: implications for paleoceanographic reconstructions, *Palaeogeography, Palaeoclimatology, Palaeoecology*, *160*, 193-212.
- Lombard, F., L. Labeyrie, E. Michel, L. Bopp, E. Cortijo, S. Retailleau, H. Howa, and F. Jorissen (2011), Modeling planktic foraminifer growth and distribution using an ecophysiological multi-species approach, *Biogeosciences*, *8*, 853-873.
- Schmidt, D. N., S. Renaud, J. Bollmann, R. Schiebel, and H. R. Thierstein (2004), Size distribution of Holocene planktic foraminifer assemblages: biogeography, ecology and adaptation, *Marine Micropaleontology*, *50*, 319-338.
- Thunell, R. C., W. B. Curry, and S. Honjo (1983), Seasonal variation in the flux of planktonic foraminifera: time series sediment trap results from the Panama Basin, *EPSL*, *64*, 44-55.

**Response to Referee #2:**

We would like to thank the reviewer for the constructive comments and suggestions, which helped us to greatly improve our manuscript. Please find, in the following, the original comments in black and our responses in light blue; the indicated page and line numbers refer to the revised manuscript.

Referee #2 comments:

The authors use existing sediment trap and plankton tow data to add seasonal and depth habitat information to the PLAFOM2.0 model. The authors then compare model results to modern data, concluding that they find a reasonable agreement between simulated and observed results for species-specific flux timing and depth habitat. The manuscript is well written, and the discussion of global trends in depth habitat is fantastic and alone an important contribution to the literature. Moreover, in light of an increasing understanding of the consequences of foraminifera habitat tracking for proxy data interpretation, the development of such a modeling tool is potentially quite useful.

The manuscript is successful in modeling modern depth preferences from unfortunately sparse observational data. While the model seems to reproduce broad trends (spinose species in near-surface waters) and earlier-when-warmer seasonality in some environments, figures 6-7 and the supplemental figures often show a strikingly poor fit between modeled and observed timing and depth preferences at specific sites. As the authors point out, the model tends to underestimate both amplitude of seasonal changes and potentially depth stratification. The authors should consider explicitly discussing why the model might be insensitive in replicating observed variability and how this would be likely to effect modeling of different climate inputs.

This is a good point and we extended the discussion in this regard especially by bearing in mind that the coarse 3° ocean model is not fully able to represent the ocean's physics properly. Apart from the uncertainty in the observational data (see section 4.2), it is due to the model complexity not trivial to determine which model component (i.e., POP2, BEC or PLAFOM2.0) contributes to what extent to the model-data-mismatch. Determining this would require a suite of sensitivity experiments with each model component. Whilst we agree that these would be useful – and will consider this for future work – we think that the model as it is already presents a useful contribution to improve the interpretation of foraminifera-based proxy records.

Nevertheless, we expanded the discussion on the model uncertainty in section 4.2 (p. 19-20, lines 23-5). We specifically addressed the dependence of the results on the individual model components. The inferred importance of temperature and food availability (provided by POP2 and BEC, respectively) on the distribution of foraminifera implies that each model component is important for an accurate representation of foraminifera distribution. Hence, as expected the higher resolution ocean model provides a more realistic representation of ocean physics, which cascades through the model hierarchy leading to an improved overall model skill. Nevertheless, sub-grid processes and known POP2 and BEC model issues (see, e.g., Danabasoglu et al., 2012, 2014; Moore et al., 2013) remain. These contributes to the model-data mismatch, but will not provide information/constraints on the planktonic foraminifera model per se.

*"The underlying complex model configuration consists of three major model components (i.e., the POP2 ocean model, the BEC ecosystem model, and PLAFOM2.0), which follow a certain model hierarchy by interacting differently with each other. Both the BEC model and PLAFOM2.0 run within POP2 (see Moore et al., 2013; Lindsay et al., 2014; this study), which provides the temperature distribution used to determine, i.a., the phytoplankton, zooplankton, and/or foraminifera carbon concentrations. It was shown that POP2 exhibits several temperature biases (e.g., Danabasoglu et al., 2012, 2014). These include large warm SST biases originating in the coastal upwelling regions of North and South America and of South Africa, colder-than-observed subthermocline waters in the equatorial Pacific as well as cold temperature biases of up to 7◦C in the*

*North Atlantic emerging throughout the water column (see Figure S5 and Danabasoglu et al., 2012, 2014). These temperature biases influence the foraminiferal distributions directly and indirectly by affecting the distributions of their food sources in the BEC model. In addition, the BEC model also exhibits several biases, such as higher-than-observed (lower-than-observed) surface nutrient and chlorophyll concentrations at low (high) latitudes (Moore et al., 2013), implying potential misrepresentations of the modeled phytoplankton and zooplankton distributions, likely influencing the foraminiferal carbon concentrations. The inferred importance of temperature and food availability (estimated by POP2 and/or the BEC model) in PLAFOM (see Fraile et al., 2008; Kretschmer et al., 2016), on the distribution of planktonic foraminifera implies that each model component is important for an accurate representation of the foraminifera distribution. Therefore, it is difficult to unequivocally differentiate between the different model components of the CESM1.2(BGC+PLA) model configuration and their individual share likely leading to the model-data-mismatch."*

When the authors discuss relative abundance of species, are they referring to relative abundance with respect to just modeled species or all foraminifera? Is this consistent throughout? It might be worth clarifying this point.

When we are discussing species relative abundances for the core-top data, we always refer to relative abundances with respect to only the five modeled species. We mention this in section 2.5.1 (page 7, line 25-26) and also in the caption of Figure 2.

Why have the authors chosen not to include sediment trap based habitat depth based assessments?

Since sediment traps provide export flux rates, which are not modeled here, and thus do not provide information about depth habitat, a sediment trap based depth habitat assessment is simply not possible. However, there exist calcification depth estimates based on chemical properties of foraminifera from sediment traps, but calcification depth is not identical to habitat depth. Therefore, we only use plankton tow data for a meaningful depth habitat assessment.

p8/l23 (and throughout) – Do the authors really mean differences in biomass as opposed to species abundances? If so, is the biomass different in different species and how is this accounted for? And how does this metric compare to species abundances, as presumably used in the modern data to which the model is compared?

PLAFOM2.0 calculates the foraminiferal abundance of each species via carbon biomass to be consistent with the ecosystem model (see section 2.3 in the manuscript and Fraile et al., 2008). In the manuscript we prefer to use this unit, rather than foraminifera abundance, since conversion to abundance requires, as the reviewer rightly points out, another step.

However, this conversion of biomass to abundance is only of importance for the comparison of the modeled and observed assemblages. For the global comparison with the core-top data, we are not interested in assessing absolute abundances and, therefore, calculate species' relative abundances. For this comparison, however, we now account for the different sizes of each species by using a relative size for each species based on the results of Schmidt et al. (2004) and recalculated the modeled relative abundances accordingly. This allowed for a sound comparison with the core-top data, which is evident in the newly introduced and considered Bray-Curtis similarity measure. We added this similarity analysis to the manuscript (i.e., to section 3.1, p. 9, lines 11-15) to provide a thorough model-data-comparison.

We would like to emphasize that the patterns of vertical and/or seasonal abundance are independent of the amount of carbon per shell (as long as there is no significant and systematic size variability). This allows us to directly compare modeled and observed data.

p9/l18 (and throughout this section) – I'm not sure it makes sense for "maximum production" to be "year-round." Could you

clarify?

That is a very good point. Here, we actually wanted to say that uniform and/or constant species fluxes occur year-round, thus no seasonal peak is evident in the species production. We changed the wording throughout this section accordingly.

section 3.3 – might be helpful to define what you mean by "surface" and "subsurface" as these are pretty general terms but are being used as if the authors have a fairly specific depth range in mind.

Thank you for pointing this out. We now provide more precise depth ranges throughout section 3.3 and especially avoided the general term "subsurface". The surface is in general defined from 0 to 10m water depth, which corresponds to the first vertical layer of the used model configuration.

p12/l30 –"prefer thriving" -> "thrive"

Done.

p12/l35 – delete "largely"

Done.

p14/l4 – delete "among each other"

Done.

p14/l11 – delete "preferably"

Done.

p14/l31 – "cold to transitional" compares a temperature to a zonation

We will changed "transitional" to "temperate" to be consistent in the wording.

p15/l22 – a -> the

Done.

p17/l2 – might be better to describe these as short time series as compared to plankton tows which really are "snapshots"

We agree and describe sediment trap time series now as short time series rather than snapshots (p.19, lines 2-4):

"[...] span at most a few years and hence represent short time series that are potentially aliased/biased by inter-annual, seasonal, and/or monthly variability. Similarly, plankton tow samples represent snapshots (of one particular day) [...]."

p17/l18 – or genotypes or phenotypes?

We agree that genotype is a more suitable term in this regard and changed the wording accordingly.

p17/l26 "a few"?

Done.

Figure 6 is extremely difficult to read given the mix of opacity and multiple symbols and colors. Is there a better way to present this data?

We agree, but could not find a better solution to present the data.

Figures 6 and 7 (a-c) suggest a quite poor fit of modeled data to sediment trap observations. i.e. 7c shows the model completing missing the flux timing of bulloides in JGOFS34. The authors include an overview or why there might be some data-model mismatch, but I think a wider discussion of why and how this could impact or limit interpretation of model results is warranted.

Please refer to our response to your first comment, where you address the same issues.

**Response to Referee #3:**

We would like to thank reviewer Jelle Bijma for his constructive comments and suggestions, which helped us to greatly improve our manuscript. Please find, in the following, the original comments in black and our responses in light blue; the indicated page and line numbers refer to the revised manuscript.

Referee #3 comments:

Scientific significance: Excellent

The manuscript by Kretschmer et al. represents a substantial contribution to scientific progress within the scope of Biogeosciences. It is the latest one in a series of "foram-flux modelling" papers from the Bremen group. In 2006, Zaric et al. Developed the first empirical model that described globally the fluxes of planktonic foraminifera at species level in dependence of sea-surface temperature, mixed-layer depth and export production. Over the years, the foram model itself, its parameterization, and its implementation and coupling to other models has evolved (e.g. Fraile et al., 2008; 2009; Kretschmer et al., 2016). The aim of all of these papers has always been to project the effect of changing environmental conditions on species distributional patterns in time and space. The current paper adds a vertical dimension to the existing foram model by applying the previously used spatial parameterization of biomass as a function of temperature, light, nutrition, and competition on depth-resolved parameter fields.

Scientific quality: good

The scientific approach and methods are valid. The results are discussed appropriately but the discussion lacks a critical analysis of the model-data comparison beyond the caveats mentioned in section 4.2 "Comparison with local observations".

Please refer to our response to reviewer 2 for proposed additions to the discussion regarding this point.

Even though the model-data-comparison revealed several discrepancies and is subject to caveats, the model produces nonetheless seasonal and vertical abundance patterns that are consistent with our current understanding and which emerge without any explicit parameterization of abundance in time and space. These patterns emerge from the model itself.

In addition, each model component (i.e., POP2, BEC, PLAFOM2.0) of the used model configuration consists of a rather complex model structure itself and rendering sensitivity experiments will be very time-consuming, expensive and non-trivial. We find that for a first try we obtain very good results.

The authors write on p. 17 line 22-23: "This vertical migration of planktonic foraminifera during their ontogeny cannot be reproduced by PLAFOM2.0 as the model parameterizations do not include the individual species' life cycles.". It is quite understandable that implementing true reproduction cycles of cohorts of foraminifera, including "real" population dynamics and ontogenetic migration is beyond the present manuscript. Hence, the model does not calculate absolute or relative numbers of a certain species within a certain ontogenetic size class based on reproductive success and size specific growth- and mortality-rates, but rather calculates changes in species specific carbon concentration (in mmol C m$^{-3}$), which can be converted to numbers afterwards.

There is nothing wrong with this approach but it means that the parameterization of PLAFOM2.0 is based on practical "sum" or "composite" parameters. These are then used to tune the model outcome to the overall data. For instance, growth of all species is approximated using a modified form of Michaelis-Menton kinetics in dependence of species specific food availability and temperature sensitivity (Fraile et al., 2008). To account for the light dependence with depth, influencing the growth of only symbiont bearing foraminifera, the authors included a "photosynthetic growth rate". They use "......a similar approach as Doney et al. (1996) and Geider et al. (1998), who determined phytoplankton growth rates by available light and

nutrients..... (p.5 line 15-17)". Such a parameterization is normally used for phytoplankton, that has orders of magnitude higher densities and cell division rates that respond very fast (within a day) and directly to light and nutrients. The symbiont bearing forams in this manuscript obey a (semi) lunar reproduction cycle and occur in densities that are very much lower, such that a "phytoplankton" kind of response cannot be expected. The authors use it as an additional tuning parameter for symbiont bearing forams next to food preference and temperature to develop species specific depth (light/nutrient) habitat preferences. Although it is a valid approach, the authors should clearly state that it is artificial.

Here, we applied a similar approach as Doney et al. (1996) and Geider et al. (1998) as a first approximation to account for a photosynthetic growth rate for the symbiont-bearing species. We are aware that a phytoplankton kind of response to light is not transferable one to one to the response of planktonic foraminifera. We made this more clear in the manuscript (p.5, lines 15-18) and we state that this approach is a first approximation, and in that way it should be considered as rather artificial. Nevertheless, we also think that this is a valid approach, given that the photosynthetic growth rate accounts in numerical terms most likely only for the smallest proportion of the total growth.

Growth is balanced by mortality, which is not a formulation for "real" mortality but another tuning parameter: "we adjusted parts of the mortality rate equation to improve the model accuracy (p. 5 line 8-9).".

Overall, there are many factors that allow tuning, e.g. "p% represents the fraction of photosynthesis contributing to growth (p.5 line 31)". Interestingly, the authors have a higher p% for T. sacculifer (0.4) than for G. ruber (0.3), where I would have done it the other way around (see my comments on these species further below).

Here, we followed Lombard et al. (2011), who also used a somewhat higher $p_\%$ for *T. sacculifer* (0.40) than for *G. ruber* (0.37). We performed a few short preliminary test runs using different $p_\%$-values but obtained the best results on a first glance by using the given parameter values. We were not able to perform a suite of sensitivity experiments with changing the $p_\%$-values due to the long runtime of the used complex model configuration.

Another tuning factor is the temperature dependence of the predation term: "......we followed Moore et al. (2004) and adjusted the temperature dependence of the predation term (MLpred in mmolCm$-3$s$-1$) (p.6 line 3-4). Also "....we included a stronger competitive behavior of G. bulloides by adjusting the free parameters in the competition term. (p.6 line 10-11). Having collected planktonic foraminifera by SCUBA diving for many, many years and looking at average typical blue water densities of ca. 10 specimens per m3 per species, and 3 dominant species in an assemblage, it is hard to believe that they compete with each other for resources as each of them occupies a space of only a few mm3 and they are stationary in the water column.

A good point indeed. Whether or not planktonic foraminifera compete directly is a field of active research. However, we would like to point out that even though foraminifera occur at very low densities and may never directly meet, they are still likely to compete for scarce resources. It is therefore reasonable to include a competition term in the model.

Certain boundary conditions also correct model misfits, e.g. "...zero fluxes have been replaced by half of the observed minimum flux. (p.7 line 25-26)".

All of these parameters were introduced to allow a good fit between model output and data but maybe not for the right reason. As such, we do not know how realistic this parameterization represents real planktonic foraminiferal population dynamics which is more complex (including lunar based reproduction cycles, ontogenetic migration, etc.).

This is true, but nevertheless we are able to simulate the seasonal and vertical habitat of the five considered foraminiferal

species remotely realistic using our approach. However, for a more realistic representation of planktonic foraminiferal population dynamics, PLAFOM2.0 needs to be extended by, e.g., considering the ontogenetic migration, reproduction cycles as well as additional foraminiferal species. Thus, PLAFOM2.0 will become more complex and more parameters have to be introduced. In addition, using, e.g., reanalysis data as forcing instead of a climatological forcing could also lead to a more realistic representation of the modern foraminiferal population dynamics when considering a point-by-point comparison with present-day data.

However, even if our understanding of foraminiferal population dynamics will be largely improved in the future due to, e.g., more laboratory experiments, and if we are able to properly translate those complex processes into model code, we will still only be able to provide an approximation of the real dynamics.

Winter mixing, thermocline shoaling and annual irradiation changes are probably important parameters controlling foram population dynamics just as certain density layers may be important for gamete fusion in real foram life. I'm not sure how well these features are implemented in the models.

This is a very good point and all those processes you mentioned likely affect the dynamics of the foraminiferal population. Here we used an ocean-ice-only model configuration and applied a climatological forcing to obtain our results. Hence, there is no explicit interaction between the ocean and the atmosphere and additionally an inter-annual variability of the forcing variables can be excluded. In addition, the lower the resolution of the ocean model the less well represented are processes such as winter mixing, thermocline shoaling, and upwelling. Since we now present results of a 1° ocean model simulation, most of these processes are likely better represented than in the previously used 3° simulation, not only because of a more realistic representation of the ocean physics, but also due to the higher resolution, which likely improved our model results. However, some small-scale processes, oceanic fronts, river runoff areas, and coastal upwelling regions are still not well represented. In order to analyze inter-annual variability of the foraminiferal population and to investigate how annual radiation changes influence the population dynamics, the model system should be forced with reanalysis data rather than climatologies. Furthermore, using a fully coupled model configuration initialized from reanalysis data could also provide information on how annual changes in the atmosphere feed back on the foraminiferal population dynamics. This, however, was beyond the scope of this study. Here, we actually aimed for an approach that is as simple and general as possible, such that we specifically avoided an explicit parameterization of depth. This way our approach is also easier to follow and we can more easily ensure the reproducibility of our study.

The bottom line is that, even though I appreciate the model and the manuscript a lot, I would like to see a discussion on these issues and if possible a statistical verification of the model performance. The description of the results and the discussion on modeled geographical ranges, seasonal and vertical distribution, as well as on the modeled seasonal variability of depth habitat, lacks a statistical treatment of the data. How good is the model performance and how sensitive is it to each of the model parameters?

Here, we did not perform a sensitivity study in regard of the different model parameters, first, because the runtime of this new model configuration is too long (with a model throughput of ~11-20 simulated years/day for the 3° model configuration and/or a model throughput of ~9.5 simulated years/day for the 1° model configuration depending on the machine capacities) to yield scientifically reasonable results and, second, because Fraile et al. (2008), who introduced PLAFOM (which is the base of PLAFOM2.0), already performed a sensitivity study of the free parameters. Fraile et al. (2008) modified the values chosen for the foraminifera module and quantified the sensitivity by calculating the change in the root mean square error between each sensitivity experiment and the standard run. They found that none of the parameters led to a uniform change for all species and that not surprisingly the parameter controlling the temperature tolerance range (i.e., $\sigma$)

seems to be the most sensitive parameter (see Table 3 of Fraile et al., 2008). Since PLAFOM2.0 is in its base form identical to PLAFOM, we did not feel the need to perform another sensitivity analysis and also due to the high computational costs. We added the following statement (p. 6, lines 22-24):

*"A parameter sensitivity assessment for PLAFOM was carried out by Fraile et al. (2008) and since PLAFOM2.0 is based on the same underlying formulation, we consider an extensive new sensitivity assessment not essential at this stage."*

I would appreciate a more quantitative treatment of the model performance instead of statements like "The predicted global distribution patterns of the five considered planktonic foraminiferal species are in good agreement with the core-top data (Figure 2) (p. 11 line 14-15)?

To perform a more quantitative model-data-comparison and to provide some measure of confidence, we now calculated the Bray-Curtis index of similarity between the model and the core-top data. For this calculation, we account for the different sizes of each species by using a relative size for each species based on the results of Schmidt et al. (2004) and recalculated the modeled relative abundances accordingly. We added this analysis to section 3.1 (p. 9, lines 11-15) to provide a more thorough model-data-comparison.

The discussion on the global distribution patterns is mostly related to temperature. What about the other parameters: food, nutrients, productivity, light, etc.?

Our results indicate that the habitat variability and the foraminiferal distributions are primarily driven by temperature and for the colder water species (*N. pachyderma*, *N. incompta*, *G. bulloides*) also by food supply. This was also shown by Fraile et al. (2008) and Kretschmer et al. (2016). Fraile et al. (2008) demonstrated that the foraminiferal distribution patterns respond most sensitively to changes in the temperature tolerance ranges of the individual species, indicating the strong temperature dependence of the foraminiferal population dynamics. Therefore, we mainly relate our results to temperature, but also discuss the food dependency extensively (see sections 4.1.3 and 4.2); the other parameters, however, seem to be less effective.

How does it compare to the "Longhurst Biogeographical Provinces". He partitioned the world oceans into provinces ("Ecological Geography of the Sea") based on the prevailing physical factors as a regulator of phytoplankton distribution, including temperature, photic depth, mixed layer depth etc. (e.g. Longhurst 1995; 1998).

The underlying parameterizations used in PLAFOM itself are based on the parameterizations used in the ecosystem model of Moore et al. (2002a) and do not include a spatial parameterization. Since Longhurst's partitioning of the ocean is more or less only descriptive, a comparison with our model results is in our understanding not appropriate. In addition, to properly compare the simulated global distribution patterns with Longhurst (1995, 1998), we would have to take into account the characteristics of each biogeographical province in the model parameterizations, which would most likely result in an overfitting.

Having "fixed" model parameters simulates so called "habitat tracking" of the forams through the seasons (but also on timescales of climate change or on glacial/interglacial cycles). This is a very important aspect to verify and would call for a section/paragraph by itself (see also Rebotim et al., 2017). For instance, on p15 line 23-25 you write "Rebotim et al. (2017) identified an annual cycle in the habitat of T. sacculifer and N. incompta in the subtropical eastern North Atlantic. Both species appear to descend in the water column from winter to spring and reach their deepest habitat in spring to summer before ascending again to a shallower depth towards winter (Rebotim et al., 2017).". How does this fit the "habitat tracking" picture? The authors could probably use observations on G. ruber and T. sacculifer for that as well. I may be wrong but I always thought that G. ruber lives closer to the surface than T. sacculifer (see also table 3 in Rebotim et al., 2017)? From

laboratory experiments I know that T. sacculifer can handle living prey such as copepods much better than G. ruber while the latter seems to rely more on symbiont carbon, i.e. shows a more "autotrophic" lifestyle. Is it possible to see this in the data based on a more rigorous model-data comparison?

Our results reveal that outside their preferred habitat, where they naturally have to face a changing environment, the seasonal occurrence of both *G. ruber* (white) and *T. sacculifer* is limited to the warm surface layer, whereas in the low latitudes both species exhibit a weak seasonal cycle in their depth habitat (see Figure 5 of the manuscript). This indicates that both species adapt to changing environmental conditions by adjusting their habitat to local circumstances, which is consistent with the concept of habitat tracking. We added this to section 4.1.3 (p. 18, lines 7-17).

In addition, we are not able to derive dietary preferences from the model, as those are prescribed for the underlying model parameterizations. In the model parameterizations, we do not distinguish between the heterotrophic and/or autotrophic lifestyle of the species, just the parameters determining the preference for a food source differ slightly among the species (see Table 1 in Fraile et al., 2008). Additionally, those parameters introduced to account for the light sensitivity of *G. ruber* (white) and *T. sacculifer* with depth differ also among them (see Table 1 of the manuscript). So by prescribing light sensitivity and food preferences a similar depth ranking compared to observations already emerges from the model. Nevertheless, for a more rigorous model-data-comparison a sensitivity study regarding the species-specific food preferences should be performed. We will bear this in mind for a future model development.

The results of the point-by-point comparative analysis for each site and species as provided in the Supplement (Figures S3 and S4) are very helpful but also show that the model is far from perfect and sometimes there is a complete mismatch. I would have appreciated a sensitivity study to determine the hierarchy of factors for the different species controlling the shell export fluxes regional and seasonal (including e.g. bimodal patterns) as well as the vertical distribution (including ALD). This would probably be a paper by itself but in my view a very important one.

This is a real good and true point and we also think that such a sensitivity study would improve PLAFOM2.0. Therefore, we agree that such a study would be very important and should be considered in the future. However, due to the high computational costs it is at present not feasible to perform this analysis. In addition, such a study would require using observational data with realistic year to year variability as forcing, but also for the model validation, which would in turn require a sensitivity study for each sediment trap/plankton tow by itself.

Based on the sensitivity analysis of Fraile et al. (2008) and also on our own results it seems that temperature has the strongest influence on the foraminiferal distribution regarding both the seasonal and vertical habitat. In particular, the distribution of each individual foraminiferal species seems to react most sensitively to changes in the individual temperature tolerance ranges (see Fraile et al., 2008). However, to further assess the sensitivity of the model to the chosen parameters especially in regard to the vertical distribution of the foraminiferal species a thorough sensitivity analysis should be performed in an independent study, which we will bear in mind for the future. Nevertheless, even after a further tuning based on such a sensitivity analysis the model will be far from perfect and discrepancies between the model data and the observations may always be present, as the caveats mentioned in section 4.2 will still be valid.

Presentation quality: good/fair

Although the scientific results and conclusions are presented in a relatively clear and well-structured way it is not easy to grasp why the model underestimates e.g peak amplitude. What would happen if growth in the equation is increased or mortality is decreased? I sometimes wondered why the authors didn't play more with the model or used statistical techniques to quantify data-model mismatch (this is the reason for the "fair" mark).

As already mentioned, due to the long runtime of the model and, hence, the high computational costs we were not able to

perform a thorough sensitivity analysis and just performed some very preliminary and short test runs to evaluate the model performance. In addition, since PLAFOM2.0 is based on PLAFOM, which has been tested and validated thoroughly (e.g., Fraile et al., 2008; Kretschmer et al., 2016), and since our aim was to demonstrate the applicability and the usability of PLAFOM to simulate the vertical distribution of individual foraminiferal species when combined with a complex 3D model configuration (such as CESM1.2(BGC)) without explicitly parameterizing the vertical dimension, we on purpose decided to not test what would happen if we change the given parameter setting. However, to better quantify the model-data-mismatch, we calculated the Bray-Curtis index of similarity between the model and the core-top data. This way, we can provide some measure of confidence regarding the general model performance. In addition, we also extended the discussion regarding the model-data-comparison by also considering potential mismatches due to the ocean model (p. 19-20, lines 23-5). Nevertheless, we are not entirely able to unequivocally differentiate between the different model components (i.e., POP2, BEC or PLAFOM2.0) and their individual share likely leading to the model-data-mismatch.

The number and quality of figures/tables is good and the supplementary material is very appropriate. The English language is very good.

Thank you!

Minor corrections:

On page 2 line 18-20: ".............the lunar cycle and/or the structure of the water column), which influence the species-specific depth habitats (including their mean living depth and vertical migration) (e.g., Fairbanks and Wiebe, 1980; Fairbanks et al., 1982; Schiebel et al., 2001; Simstich et al., 2003; Field, 2004; Salmon et al., 2015; Rebotim et al., 2017), the only attempt to model the vertical habitat is by Lombard et al. (2011).", and on page 17 line 20-23: "Several studies from different areas also showed that the main habitat depth of some species increases from the surface to deeper water layers during shell growth (Peeters and Brummer, 2002; Field, 2004; Iwasaki et al., 2017). Although I appreciate all the references that you list for ontogenetic migration and lunar cycle, there are only a few papers that specifically deal with very detailed population dynamics, lunar cyclicity and ontogenetic migration of planktonic forams that could/should be mentioned here (it was one of the first topics I studied when starting to work on planktonic foraminifera): Bijma et al., 1990; Bijma, 1991; Bijma and Hemleben, 1994; Bijma et al., 1994; Hemleben and Bijma, 1994; Schiebel et al., 1997. In my opinion, these references would fit best on p. 19 line 32-34: "........and by explicitly parameterizing the ontogeny of each individual planktonic foraminifera, thus, by considering the changes in the species' life cycles with depth, could considerably improve the model.".

Thank you for pointing this out. We added the mentioned references accordingly (see p. 22, lines 17-18).

P. 9 line 27-30: "Although seasonal changes in the modeled foraminiferal peak fluxes with temperature are evident, all five species exhibit an almost constant peak amplitude (i.e., the maximum concentration divided by the annual mean) in their preferred habitat, which is, i.a., limited by temperature. Outside their preferred living conditions the peak amplitudes increase for most of the species considerably (Figure 3).". It has not become clear to me what it means when "peak amplitude" is large or small in terms of real population dynamics ("bloom"?) and what it means in terms of model performance?

The maximum seasonal abundance or flux in itself is not a very useful parameter that can be compared among different regions/studies. Population dynamics can be much better (if not only) described in terms of deviations from the mean conditions, for instance, it would be impossible to recognize a bloom event in the absence of knowledge about the mean conditions. Moreover, any seasonal or vertical weighting of the proxy signal – and constraining this was the main motivation to develop the model – varies as a function of the relative departure from the mean. To be more clear about this, we rewrote

this paragraph and provided a better explanation (p. 10, lines 21-26):

*"To allow for a global comparison of the modeled and observed flux seasonality, we standardized peak amplitudes for each foraminiferal species, i.e., the species' maximum concentration divided by its annual mean. This reveals that the timing of the modeled foraminiferal peak abundances varies with temperature, but all five species exhibit an almost constant peak amplitude in their preferred thermal habitat. Outside their preferred living conditions, modeled peak amplitudes considerably increase for most of the species(Figure 3), thus, the species experience a strong deviation from their annual mean living conditions and likely occur only at times when the ambient conditions are (close to) their optima."*

P. 14 line 26-28: "This would explain why the highest modeled concentrations of T. sacculifer occur at shallower depths compared to G. ruber (white) (see Figures 4d-e and 5d-e).". Strictly speaking this doesn't explain it because this is what you put into the model in the first place (see my comments above).

Actually, this is a perfect example how the habitat emerges from the model. We only prescribe the light sensitivity and still obtain the right depth ranking. Throughout the model code, we specifically did not specify the depth ranking. We rewrote this sentence to avoid confusion (p. 16, lines 8-11):

*"This is to some degree also indicated in our results, as on average the highest modeled concentrations of T. sacculifer occur at shallower depths compared to G. ruber (white) (see Figures 4d-e and 5d-e). However, at some locations both model and observations show the reverse (see Figure S4 and, e.g., Rippert et al., 2016; Rebotim et al., 2017), indicating that this depth ranking is not globally valid. "*

P. 16 line 18: "G. bulloides, however, is found year-round close to the surface along the.....". Write the genus name full at the beginning of a sentence.

Done and applied throughout the manuscript.

**Response to Referee #4:**

We would like to thank the reviewer for the constructive comments and suggestions, which helped us to greatly improve our manuscript. Please find, in the following, the original comments in black and our responses in light blue; the indicated page and line numbers refer to the revised manuscript.

Referee #4 comments:

This paper builds upon preexisting work modeling planktonic foram distributions in the global oceans via a coupling to CESM's ocean model. The goal is to better understand how the vertical distribution of foraminifera species varies seasonally and throughout larger climatic changes in the ocean. The paper is generally well written, clear, and broadly does a fine job demonstrating the usefulness of the model. It is also very thorough in its examination of the model's performance against available data. The methods seem robust and I can recommend that with some minor revisions (mostly grammar and clarity) the paper be published in Biogeosciences.

I must acknowledge that I am not an expert on the biogeochemistry of planktonic forams in any way and hope the other reviewers can address the methods and parameterizations employed in this paper in particular. I can instead comment on the benefit of this work and the need for such proxy system models for the robust interpretation of paleoceanographic records via the use of PLAFOM2.0 + CESM1.2. To that end, my first major comment is that the authors can focus more in the introduction and conclusion on the body of literature developing forward models, or proxy system models, for understanding paleoclimate proxies and introduce this work as a part of this group of literature. A major effort has been underway to build proxy system models, link them with GCMs, and make these models publicly available, and this paper is absolutely in this category and should make as much clear.

See for example:

- Dee, S., et al. "PRYSM: An open-source framework for PRoxY System Modeling, with applications to oxygen-isotope systems." Journal of Advances in Modeling Earth Systems 7.3 (2015): 1220-1247.
- Evans, Michael N., et al. "Applications of proxy system modeling in high resolution paleoclimatology." *Quaternary Science Reviews* 76 (2013): 16-28.
- Schmidt, Gavin A. "Forward modeling of carbonate proxy data from planktonic foraminifera using oxygen isotope tracers in a global ocean model." *Paleoceanography* 14.4 (1999): 482-497.

We agree that over the last decades proxy system/formation models have become more and more important for understanding paleoclimate proxies and that PLAFOM2.0 belongs to a series of different proxy system/formation models. We briefly introduced PLAFOM2.0 as part of this large group of proxy system models in section 2.1 (page 3, lines 22-24):

*"Thus, PLAFOM2.0, as belonging to a suite of proxy system models (e.g., Pollard and Schulz, 1994; Schmidt, 1999; Fraile et al., 2008; Evans et al., 2013; Dee et al., 2015; Völpel et al., 2017), will aid the interpretation of paleoclimate reconstructions."*

You might also consider mentioning (in the intro or discussion) the potential for PLAFOM to assist in data assimilation exercises for periods extending back further than the last millennium, for example. A number of papers look at the impacts of using process-based models in the DA framework and this is another application of your model. See work of Hugues Goosse's lab (e.g. Goosse, Hugues, et al. "Reconstructing surface temperature changes over the past 600 years using climate model simulations with data assimilation." *Journal of Geophysical Research: Atmospheres* 115.D9 (2010)), as well as:

- Steiger, Nathan J., et al. "Assimilation of time-averaged pseudoproxies for climate reconstruction." *Journal of Climate* 27.1 (2014): 426-441.

- Dee, Sylvia G., et al. "On the utility of proxy system models for estimating climate states over the common era." *Journal of Advances in Modeling Earth Systems* 8.3 (2016):
- Hakim, Gregory J., et al. "The last millennium climate reanalysis project: Framework and first results." *Journal of Geophysical Research: Atmospheres* 121.12 (2016): 6745-6764

Thank you for pointing this out. We agree that PLAFOM2.0 has the potential to be used in the data assimilation framework and we added a statement in this regard to section 2.1 (p. 3, lines 24-26):

*"In addition, PLAFOM2.0 has the potential to be used in a 25 paleoclimate data assimilation framework (see, e.g., Goosse et al., 2010; Steiger et al., 2014; Dee et al., 2016; Hakim et al., 2016)."*

In Section 4, it would be nice if the authors could provide a more quantitative data-model comparison technique—you identify areas where the model does not well simulate the observations and Figure 2 summarizes this to some extent, but perhaps you could include an additional table or figure or even compute something like the RMSE for each oceanic province? Or the mean RMSE for each species over all of the locations where core-top data exist?

Please refer to our response to a similar comment by reviewer #1 regarding a more quantitative model-data-comparison.

Finally, in the discussion, you assert (correctly) that your new model is a powerful tool for separating the independent influences of habitat and climate on foram reconstructions. I think this paper would be greatly strengthened by a demonstration of this. Can you take a well-known and vetted reconstruction and apply this model in a meaningful way to reassess the climatic interpretation? I think this would show the power of forward modeling in this field to make more robust assessments of uncertainties in oceanic climate changes... And I think having this demonstration would add weight to the assertions you make in your Discussion section.

This is a good point and we agree that such a demonstration would add to a better understanding of climate change. In a next study, we plan on performing a model run with, e.g., Last Glacial Maximum climate conditions to test the applicability of our modeling approach. Here, in this study we simply wanted to test if the existing planktonic foraminifera model is able to reproduce species-specific habitats when combined with a model configuration that resolves the vertical.

Minor / Line by Line comments: (Page-Line)

2-10 awkard paragraph break, consider revising

We agree and deleted the paragraph break.

2-13 comma after perspective,

Done.

2-20 Have you investigated/reviewed Schmidt et al., 1998, 1999? These papers I believe address vertical migration of foram species in the water column—worth checking/citing if appropriate.

- Schmidt, Gavin A. "Oxygen-18 variations in a global ocean model." *GRL* 25.8 (1998): 1201-1204.
- Schmidt, Gavin A. "Forward modeling of carbonate proxy data from planktonic foraminifera using oxygen isotope tracers in a global ocean model." *Paleoceanography* 14.4 (1999): 482-497.

Thank you for referring to those two studies. In both studies, Schmidt does unfortunately not address the vertical migration of foraminifera. Schmidt (1998, 1999) investigates the distribution of oxygen isotopes in seawater and subsequently calculates equilibrium calcite values based on different temperature equations. This, however, is beyond the scope of our

present study and, hence, citing those studies is not appropriate.

2-26 need comma after behavior.

Done.

Check for needed commas and small grammatical errors throughout text.

Done.

3-6 comma after estimate,

Done.

3-13 this phrase is awkward, revise ("with the biogeochemical model being enabled")

We revised the phrase as follows (p. 3, lines 9-11):

*"[…] as an off-line module into the ocean component of the Community Earth System Model, version 1.2.2 (CESM1.2; Hurrell et al., 2013), with active ocean biogeochemistry (which is denoted as CESM1.2(BGC) configuration)."*

3-15 change "aimed for" to 'aimed to'

Done.

3-16 change "at geologic timescales" to "ON geologic timescales"

Done.

Check for similar awkward language throughout.

Done.

3-23 comma after configuration,

Done.

3-30 no paragraph break.

Done.

4-9 what do you mean by 'data models' for the atmosphere, etc.? Are you not using the fully coupled simulations and using some kind of statistical representation of the other components?

The CESM data models are "non-active" model components that read external data, modify that data (e.g., interpolate that data in time and space), and subsequently return the final data fields to the coupler. In this study, we did not perform a fully coupled simulation. Here we analyze an ocean-ice-only simulation with active ocean biogeochemistry coupled to data models for the atmosphere, land, and river routing. Since "data model" is a common term in the CESM model community, we also used it to be consistent with other publications using the CESM1.2(BGC) configuration. However, for a better understanding, we revised the sentence as follows (p.4, lines 10-12):

*"Here we performed an ocean-ice-only simulation with active ocean biogeochemistry, whereby the ocean model is coupled to both the sea ice model and data models for the atmosphere, land, and river routing, which provide the required input data for the simulation."*

Heading 2.4 consider changing this to "Coupled GCM Setup" ?

Since our results are based on an ocean-ice-only simulation, a heading change to "Coupled GCM setup" is not appropriate. Nevertheless, we changed heading 2.4 to "Model simulation", which is more accurately describing this section.

7-15 missing space before new sentence.

Done.

8-21 comma after 'life cycle,'

Done.

Throughout section 3, be extremely clear about whether you are referring to observations vs. the model simulation of foram distributions/abundances etc. The reader gets a bit lost in the data-model comparison here unless that's super clear.

In section 3, we actually just describe model results and do not provide a model-data-comparison. To be more concise and clear about that, we revised this section accordingly. In addition, we also revised section 4 to be more clear about when we refer to observations and/or model output.

16-29 no comma after 'data'

Done.

16-30 this is a run-on sentence—consider shortening/rewriting

We revised this run-on sentence by splitting it into two parts (p. 18, lines 29-31):

*"The emergence of seasonal and vertical habitat patterns consistent with observational data provides important support for our modeling approach. Nevertheless, a more detailed comparison with observations is warranted to gain further insight into the model behavior."*

I appreciate the thorough discussion of the model – data comparison limitations on page 17.

Thank you!

Figure 5 has some strange cropping issues along top margin.

Thank you for pointing this out. We checked for this and adjusted Figure 5 accordingly.

[revised manuscript text omitted]

**Site GS2**

[Figure]

**Site OG5**

[Figure]

**Site NB6/7**

[Figure]

[Figure]

**Site PAC50**

[Figure]

[Figure]

**Site PAPA**

[Figure]

[Figure]

**Site SA**

[Figure]

[Figure]

**Site KNOT**

[Figure]

**Site WCT6**

[Figure]

**Site WCT2**

[Figure]

[Figure]

**Site WCT7**

[Figure]

**Site WCT1**

[Figure]

[Figure]

[Figure]

**Site SBB**

[Figure]

**Site SPB**

[Figure]

**Site JGOFS34**

[Figure]

[Figure]

[Figure]

**Site BATS**

[Figure]

**Site WAST**

[Figure]

**Site EA1**

[Figure]

**Site EA2**

[Figure]

**Site EA3**

[Figure]

[Figure]

[Figure]

**Site EA4**

[Figure]

[Figure]

[Figure]

**Site WA1**

**Site NCR**

[Figure]

**Site SCR**

[Figure]

**Site CP**

[Figure]

**Site WS34**

[Figure]

**Figure S4.**

**Station 93-36**

[Figure]

**Station PS78-25**

[Figure]

**Station PS78-44**

[Figure]

**Station PS78-75**

[Figure]

**Station PS55-025**

*N. pachyderma*

[Figure]

**Station PS55-043**

*N. pachyderma*

**Station PS55-063**

*N. pachyderma*          *G. bulloides*

**Station MN116**

[Figure]

**Station MN2**

[Figure]

**Station MN323**

[Figure]

**Station MN314**

[Figure]

**Station PAPA**

[Figure]

**Station 101**

[Figure]

**Station 79**

[Figure]

**Station KNOT**

[Figure]

**Station #B**

[Figure]

**Station #b**

[Figure]

**Station #A**

[Figure]

**Station #E**

[Figure]

**Station POS383-165**

[Figure]

**Station POS383-175**

[Figure]

**Station POS247-1389**

[Figure]

**Station MOC1-38**

[Figure]

**Station MOC1-28**

[Figure]

**Station MOC1-23**

[Figure]

**Station 310**

[Figure]

**Station 920**

[Figure]

**Station 313**

[Figure]

**Station 917**

[Figure]

**Station MOC63**

[Figure]

**Station MOC65**

[Figure]

**Station MOC12**

[Figure]

**Station MOC66**

[Figure]

**Station MOC15**

[Figure]

**Station MOC69**

[Figure]

**Station MOC20**

[Figure]

**Station MOC71**

[Figure]

**Station MOC72**

[Figure]

**Station SO225-21-3**

[Figure]

**Station TNO57-16**

[Figure]

**Station TNO57-13**

[Figure]

**Station AN98-O**

[Figure]

**Station AN99-O**

[Figure]

**Station AN00-O**

[Figure]

**Station AN01-O**

[Figure]

[Figure]

**Table S1.**

[revised manuscript text omitted]

[a] The nearest model grid point for site SBB fell onto land. Therefore, we used the nearest model grid point in the ocean to perform a consistent model-data-comparison.

**Table S3b.**

| Province | Site | Latitude (°N) | Longitude (°E) | *N. pachyderma* Trap | PLAFOM2.0 | *N. incompta* Trap | PLAFOM2.0 | *G. bulloides* Trap | PLAFOM2.0 | *G. ruber* (white) Trap | PLAFOM2.0 | *T. sacculifer* Trap | PLAFOM2.0 |
|---|---|---|---|---|---|---|---|---|---|---|---|---|---|
| Polar | GS2 | 75.00 | 0.00 | 0.78 |  0.26 | - | - | - | - | - | - | - | - |
| Polar | OG5 | 72.40 | -7.70 | 0.64 |  0.19 | - | - | - | - | - | - | - | - |
| Polar | NB6/7 | 69.69 | -0.47 | 0.80 |  0.42 | 0.95 |  0.57 | - | - | - | - | - | - |
| Sub-polar | PAC50 | 50.01 | 165.03 | 0.75 |  0.20 | 0.77 |  0.36 | 0.66 |  0.36 | - | - | - | - |
| Sub-polar | PAPA | 50.00 | -145.00 | 1.07 |  0.28 | 1.20 | 0.04 | 1.10 |  0.22 | - | - | - | - |
| Sub-polar | SA | 49.00 | -174.00 | 0.95 |  0.29 | - | - | 0.94 |  0.21 | - | - | - | - |
| Transitional | KNOT | 43.97 | 155.06 | 0.69 |  0.36 | 0.79 |  0.16 | 0.66 |  0.33 | 0.76 |  0.70 | - | - |
| Transitional | WCT6 | 42.00 | 155.34 | 0.42 |  0.33 | 0.47 |  0.15 | 0.62 |  0.25 | 0.73 |  0.59 | - | - |
| Transitional | WCT2 | 39.00 | 147.00 | - | - | 0.64 |  0.17 | 0.69 |  0.22 | 0.74 |  0.44 | - | - |
| Transitional | WCT7 | 36.68 | 154.94 | - | - | - | - | 0.55 |  0.19 | - | - | 0.57 |  0.34 |
| Subtropics | SBB[a] | 34.23 | -120.03 | - | - | 0.68 |  0.04 | 0.67 | 0.06 | 0.87 |  0.19 | - | - |
| Subtropics | JGOFS34 | 34.00 | -21.00 | - | - | 0.85 |  0.05 | 0.76 |  0.07 | - | - | 0.69 |  0.14 |
| Subtropics | SPB | 33.55 | -118.50 | - | - | 0.73 |  0.04 | 0.85 | 0.06 | 0.77 |  0.18 | - | - |
| Subtropics | L1 | 33.00 | -22.00 | - | - | 1.28 |  0.05 | 0.91 |  0.06 | 0.70 |  0.09 | 0.59 |  0.13 |
| Subtropics | BATS | 32.08 | -64.25 | - | - | - | - | 0.72 | 0.05 | 0.37 |  0.07 | 0.96 |  0.12 |
| Subtropics | WCT1 | 25.00 | 136.99 | - | - | 0.42 |  0.06 | - | - | 0.77 |  0.10 | 0.88 |  0.07 |
| Tropics | WAST | 16.32 | 60.47 | - | - | - | - | 0.77 |  0.02 | 0.70 |  0.14 | 0.66 |  0.02 |
| Tropics | EA1 | 3.17 | -11.25 | - | - | - | - | 0.48 |  0.03 | 0.36 |  0.15 | 0.47 |  0.07 |
| Tropics | EA2 | 1.78 | -11.25 | - | - | - | - | 0.52 | 0.02 | 0.33 |  0.13 | 0.59 |  0.03 |
| Tropics | EA3 | 0.08 | -10.77 | - | - | - | - | 0.81 |  0.02 | 0.57 |  0.10 | 0.47 |  0.05 |
| Tropics | EA4 | -2.19 | -10.09 | - | - | - | - | 0.83 |  0.02 | 0.60 |  0.10 | 0.50 |  0.03 |
| Tropics | WA1 | -4.00 | -25.57 | - | - | - | - | - | - | 0.67 |  0.15 | 0.58 |  0.06 |
| Transitional | NCR | -42.70 | 178.63 | 0.91 |  0.47 | 0.78 |  0.11 | 0.73 |  0.16 | 0.77 |  0.46 | 0.85 | 0.54 |
| Transitional | SCR | -44.62 | 178.62 | 0.87 |  0.14 | 0.57 |  0.08 | 0.89 |  0.16 | - | - | - | - |
| Sub-polar | CP | -52.62 | 174.15 | 1.29 | 0.28 | 1.12 |  0.08 | 1.10 |  0.11 | - | - | - | - |
| Polar | WS34 | -64.90 | -2.60 | 1.12 |  0.09 | - | - | - | - | - | - | - | - |

[a] The nearest model grid point for site SBB fell onto land. Therefore, we used the nearest model grid point in the ocean to perform a consistent model-data-comparison.

**Table S4.**

| Province | Site | Latitude (°N) | Longitude (°E) | N. pachyderma Tow[a] (m) | N. pachyderma PLAFOM2.0[a] (m) | N. incompta Tow[a] (m) | N. incompta PLAFOM2.0[a] (m) | G. bulloides Tow[a] (m) | G. bulloides PLAFOM2.0[a] (m) | G. ruber (white) Tow[a] (m) | G. ruber (white) PLAFOM2.0[a] (m) | T. sacculifer Tow[a] (m) | T. sacculifer PLAFOM2.0[a] (m) |
|---|---|---|---|---|---|---|---|---|---|---|---|---|---|
| Polar | 93-36 | 80.36 | -10.14 | 85±35 | 55± 35 | - | - | - | - | - | - | - | - |
| | PS78-25 | 78.83 |  7.00 |  85± 55 |  50± 35 | - | - | - | - | - | - | - | - |
| | PS78-44 | 78.83 |  0.08 |  80± 40 |  45± 35 | - | - | - | - | - | - | - | - |
| | PS78-75 | 78.83 | -3.92 | 70±40 |  45± 35 | - | - | - | - | - | - | - | - |
| | PS55-025 | 75.00 | -10.58 | 90±70 |  45±35 | - | - | - | - | - | - | - | - |
| | PS55-043 | 75.00 | 0.36 | 60±40 |  45± 30 | - | - | - | - | - | - | - | - |
| | PS55-063 | 75.00 | 10.65 | 85±65 |  45± 30 | - | - | 55±25 | 25±15 | - | - | - | - |
| | MN116 | 75.00 | -7.31 | 150±40 |  45± 30 | - | - | - | - | - | - | - | - |
| | MN2 | 70.00 | 3.40 | 170±215 |  60± 50 | - | - | - | - | - | - | - | - |
| | MN323 | 69.69 | 0.47 | 140±155 | 55± 45 | - | - | - | - | - | - | - | - |
| | MN314 | 67.54 | 5.58 | 125±60 | 65± 55 | - | - | - | - | - | - | - | - |
| Sub-polar | PAPA | 49.98 | -144.97 | 230±30 | 80±35 | 105±85 |  60±35 | 60±45 | 70±40 | - | - | - | - |
| | 101 | 47.00 | -174.95 | 95±50 |  60± 40 | 140±40 | 55±35 | 65±30 |  50± 35 | - | - | - | - |
| | 79 | 46.98 | 166.73 | 110±55 |  55± 40 | 150±50 |  25± 15 | 70±35 |  30± 25 | - | - | - | - |
| Transitional | KNOT | 44.08 | 154.98 | 90±45 |  70± 45 | 75±55 |  20± 10 | 45±30 |  30± 25 | - | - | - | - |
| | #B[b] | 41.57 | 141.90 | 125±40 |  80± 40 | 105±45 |  70± 45 | 100±55 | 75±45 | 122±40 | 5±5 | - | - |
| | #b | 41.15 | 143.38 | 85±40 |  80± 40 | 35±25 |  70± 45 | 40±30 | 75±45 | - | - | - | - |
| | MOC1-38 | 38.92 | -67.90 | - | - | - | - | 65±55 |  75± 40 | 30±20 |  35± 20 | 35±20 |  40± 20 |
| | #A | 36.02 | 141.78 | - | - | 25±20 |  65± 35 | 25±25 |  65±35 | 20±20 |  25± 20 | 25±20 |  30± 20 |
| Subtropics | POS383-165 | 34.00 | -22.00 | - | - | 85±60 |  80±45 | 170±70 |  75± 40 | 65±25 |  30± 25 | 185±85 | 50±30 |
| | MOC1-28 | 33.91 | -71.78 | - | - | - | - | 80±35 |  50± 30 | 60±35 | 50±30 | - | - |
| | POS383-175 | 33.15 | -22.00 | - | - | 85±55 |  80± 40 | 95±50 |  75± 40 | 65±25 |  35± 25 | 190±65 | 50±30 |
| | POS247-1389 | 33.08 | -22.00 | - | - | 30±0 |  80±45 | 30±0 |  80±45 | 55±25 |  50±30 | 40±35 | 55±30 |
| | MOC1-23 | 32.73 | -71.16 | - | - | - | - | 140±0 |  70± 35 | 115±25 | 45±25 | 95±50 |  50±30 |
| | #E | 32.17 | 133.88 | - | - | 60±30 |  80± 40 | 70±45 | - | 30±20 | - | 45±40 |  35± 25 |
| Tropics | 920 | 16.09 | 52.70 | - | - | - | - | 65±65 |  70± 40 | 40±45 | 40±20 | 20±20 | 45±25 |
| | 310 | 16.02 | 52.73 | - | - | - | - | 30±35 |  50±35 | 180±100 |  10± 5 | 180±85 |  15± 10 |
| | 313 | 15.91 | 53.02 | - | - | - | - | 70±80 |  50±35 | 30±35 |  10± 5 | 30±30 |  10± 5 |
| | 917 | 15.89 | 52.97 | - | - | - | - | 90±65 |  70± 40 | 75±50 | 40±20 | 20±15 | 45±25 |
| | MOC63 | 2.92 | -140.20 | - | - | - | - | 15±10 | 95±50 | 20±15 | 45±25 | 25±15 |  45±25 |
| | MOC65 | 2.05 | -141.49 | - | - | - | - | 25±15 | 90±50 | 35±25 |  40± 20 | 25±15 | 40±25 |
| | MOC12 | 2.01 | -139.88 | - | - | - | - | 65±25 |  80±45 | 45±25 | 45±25 | 45±25 |  40± 25 |
| | MOC66 | 1.13 | -140.01 | - | - | - | - | 55±25 | 85±45 | 45±20 | 35±20 | 25±15 | 35±20 |
| | MOC15 | 0.00 | -140.07 | - | - | - | - | - | - | 20±10 | 35±20 | 25±15 | 35±20 |
| | MOC69 | -1.05 | -139.97 | - | - | - | - | 25±15 |  80±45 | 25±15 |  20±15 | 25±15 |  40±20 |
| | MOC20 | -2.02 | -140.16 | - | - | - | - | - | - | 35±15 |  45±25 | 40±20 | 40±25 |
| | MOC71 | -2.33 | -140.32 | - | - | - | - | 45±25 |  90±45 | 35±25 | 45±25 | 35±25 |  45±25 |
| | SO225-21-3 | -3.05 | -165.06 | - | - | - | - | 145±90 | 55±30 | 65±35 |  55± 30 | 75±45 |  55±30 |
| | MOC72 | -3.21 | -140.25 | - | - | - | - | 40±20 | 90±50 | 35±15 | 50±25 | 35±20 |  50±25 |
| Sub-polar | TNO57-16 | -50.12 | 5.75 | 70±10 |  60±35 | 70±10 | 25±15 | 80±5 | 30±15 | - | - | - | - |
| | TNO57-13 | -53.18 | 5.13 | 85±60 |  55±35 | - | - |  |  | - | - | - | - |
| Polar | AN98-O | -63.25 | 177.25 | 55±30 | 60±35 | - | - | - | - | - | - | - | - |
| | AN99-O | -63.40 | 178.05 | 25±15 |  60±40 | - | - | - | - | - | - | - | - |
| | AN01-O | -63.43 | 178.10 | 120±0 |  60±40 | - | - | - | - | - | - | - | - |
| | AN00-O | -63.53 | 178.38 | 95±50 |  60±40 | - | - | - | - | - | - | - | - |

[a] ALD±VD (in m) of the planktonic foraminiferal species calculated after Rebotim et al. (2017) for the plankton tow samples and for PLAFOM2.0 (obtained at the nearest model grid points of the given plankton tow locations). Note that the values have been rounded to the nearest 5 m.
[b] The nearest model grid point for site #B fell onto the shelf. Therefore, we used the nearest model grid point in the open ocean to perform a consistent model-data-comparison.

---

## Author Response (AR2)

**Ref.: Ms. No. bg-2017-429**

**Title: Modeling seasonal and vertical habitats of planktonic foraminifera on a global scale**

Dear Lennart,

We would like to thank you for reviewing our revised manuscript. We have prepared a corrected version of our manuscript with your suggestions of improvement taken into consideration. Please find, in the following, your original comments in black and our responses in light blue; the indicated page and line numbers refer to the corrected manuscript.

We hope that now everything is correct and understandable and that our corrected manuscript is worthy of being published in Biogeosciences.

Kind regards,

Kerstin Kretschmer (on behalf of all co-authors)
* * *
Comments to the Author:

Dear Dr. Kretschmer and co-workers,

I had a thorough look at the replies to the reviewers, as well as the new version of your manuscript, which I now think is fit for publication in Biogeosciences. I came upon a few (which is surprising given the length of your manuscript) typos, which I copied below.

Sincerely,

Lennart

Abstract

line 2: `seasonality´ is a bit weird here. What is `the seasonality´ of a species? Could you rephrase this?

We agree and rephrased the sentence (p.1, line 2):

*"The seasonal and vertical habitats are not constant […]."*

lines 3-5: this sentence is a bit difficult to follow: why does one need independent fossil evidence? I think I understand the this, but could you still be a bit more specific?

Thank you for pointing this out. We rephrased the sentence as follows (p. 1, lines 3-6):

*"However, detecting the effect of changing vertical and seasonal habitat on foraminifera proxies requires independent evidence for either habitat or climate change. In practice, this renders accounting for habitat tracking from fossil evidence almost impossible. An alternative method that could […]."*

line 23: `indicates´ should be `indicating´

Here `indicates´ is related to `The emergence´ and is, thus, the grammatically correct form. However, to avoid confusion we rephrased the sentence as follows (p. 1, lines 22-23):

*"The emergence in PLAFOM2.0 of species-specific vertical habitats, which are consistent with observations, indicates that the population [...]."*

Introduction

page 2, line 23: insert `that´ before `it´

Done (see p. 2, line 24).

Results

page 9, line 6: `provide´

Done (see p. 9, line 6).

page 11, line 25: remove `in the model´

Done (see p. 11, line 25).

page 11, line 28: try to avoid `deeper depths´

Thanks for pointing this out. We now avoid using `deeper depths´ throughout the manuscript and changed the wording accordingly.

p. 11, line 28: *"[...] N. pachyderma is generally found between 50 and 100m water depth for almost [...]."*

[revised manuscript text omitted]